# Conditional Gradient Methods with Standard LMO for Stochastic Simple Bilevel Optimization

**Khanh-Hung Giang-Tran**
Cornell University
tg452@cornell.edu

**Soroosh Shafiee**
Cornell University
shafiee@cornell.edu

**Nam Ho-Nguyen**
The University of Sydney
nam.ho-nguyen@sydney.edu.au

## Abstract

We propose efficient methods for solving stochastic simple bilevel optimization problems with convex inner levels, where the goal is to minimize an outer stochastic objective function subject to the solution set of an inner stochastic optimization problem. Existing methods often rely on costly projection or linear optimization oracles over complex sets, limiting their scalability. To overcome this, we propose an iteratively regularized conditional gradient approach that leverages linear optimization oracles exclusively over the base feasible set. Our proposed methods employ a vanishing regularization sequence that progressively emphasizes the inner problem while biasing towards desirable minimal outer objective solutions. In the one-sample stochastic setting and under standard convexity assumptions, we establish non-asymptotic convergence rates of $\tilde{O}(t^{-1/4})$ for both the outer and inner objectives. In the finite-sum setting with a mini-batch scheme, the corresponding rates become $\tilde{O}(t^{-1/2})$. When the outer objective is nonconvex, we prove non-asymptotic convergence rates of $\tilde{O}(t^{-1/7})$ for both the outer and inner objectives in the one-sample stochastic setting, and $\tilde{O}(t^{-1/4})$ in the finite-sum setting. Experimental results on over-parametrized regression and dictionary learning tasks demonstrate the practical advantages of our approach over existing methods, confirming our theoretical findings.

## 1 Introduction

We consider *stochastic simple bilevel optimization problems*, where the goal is to minimize an outer stochastic objective function subject to the solution set of an inner stochastic convex optimization problem. Formally, the problem is defined as

$$F_{\text{opt}} := \min_{x \in \mathbf{R}^d} \{F(x) := \mathbb{E}[f(x, \theta)] : x \in X_{\text{opt}}\}, \text{ where } X_{\text{opt}} := \arg\min_{z \in X} \{G(z) := \mathbb{E}[g(z, \xi)]\}.$$
(1)

Here $X \subset \mathbb{R}^d$ is a compact convex set, and both $F$ and $G$ are smooth with $G$ is additionally convex, making the optimization problem (1) a possibly nonconvex problem with convex domain. This framework has broad applicability, including hyper-parameter optimization [16, 17, 39], meta-learning [17, 42], reinforcement learning [32] and game theory [50]. In particular, problem (1) arises in learning applications where $G$ is the prediction error of a model $x$ on data, and $F$ is an auxiliary objective (e.g., regularization, or a validation data set). The dependence on data is through the expectation operators and the random variables $\theta, \xi$. The bilevel formulation provides a *tuning-free* alternative to the usual regularization approach where we optimize $\sigma F(x) + G(x)$ for a carefully tuned parameter $\sigma$.

Despite its wide applications in machine learning there are three primary challenges in solving (1). First, we do not have an explicit representation of the optimal set $X_{\text{opt}}$ in general, preventing us

39th Conference on Neural Information Processing Systems (NeurIPS 2025).

from using some common operations in optimization such as projection onto or linear optimization over $X_{\mathrm{opt}}$. An alternative is to reformulate (1) using its *value function formulation*

$$\min_{x \in X} \quad F(x) \quad \text{s.t.} \quad G(x) \leq G_{\mathrm{opt}} := \min_{z \in X} G(z). \tag{2}$$

However, this leads to our second challenge. Problem (2) is inherently ill-conditioned: by definition of $G_{\mathrm{opt}}$, there exists no $x \in X$ such that $G(x) < G_{\mathrm{opt}}$, hence (2) does not satisfy Slater's condition. As a result, the Lagrangian dual of (2) may not be solvable, complicating the use of standard primal-dual methods. In addition, we do not know $G_{\mathrm{opt}}$ a priori, so an alternative is to approximate it with $\bar{G} \leq G_{\mathrm{opt}} + \epsilon_g$, and consider the constraint $G(x) \leq \bar{G}$ instead. However, this approach does not solve the actual bilevel problem (1), and may still introduce numerical instability. The third challenge comes from the stochastic nature of the objectives. Since $F$ and $G$ are defined as expectations, their exact computation may be intractable when dealing with large-scale datasets. In single-level optimization, this is typically addressed through sampling-based methods that operate on mini-batches drawn from the distributions of $\theta$ and $\xi$ instead of the actual distributions themselves. However, such stochastic approximations introduce noise into the optimization process, necessitating new techniques to control the noise and ensure convergence.

## 1.1 Related works

**Simple bilevel optimization problems with nonconvex outer objectives** This setting has been studied by Jiang et al. [29] in the deterministic case and by Cao et al. [4] in the stochastic case. These works propose conditional gradient methods that require performing linear optimization over intersections of the base domain $X$ with a halfspace. In contrast, our approach removes this requirement by relying solely on linear optimization over $X$. We note that we do not address the more general case with both nonconvex inner and outer objectives considered in [5, 21, 24]. In the remainder of this section, we will focus on simple bilevel problems with convex inner and outer objectives.

**Iterative regularization methods.** Simple bilevel optimization extends Tikhonov regularization [52] beyond $\ell_2$ penalties by considering a blended objective $\sigma F(x) + G(x)$, where $\sigma > 0$ modulates the trade-off between inner and outer objectives. As $\sigma \to 0$, the problem converges to a bilevel form that prioritizes the inner objective. Friedlander and Tseng [18] established fundamental conditions ensuring solution existence for $\sigma > 0$, forming the basis for *iterative regularization*, where $\sigma$ is gradually reduced. This idea was first implemented by Cabot [3] in the unconstrained case $X = \mathbf{R}^d$ using a positive decreasing sequence $\sigma_t$, and later extended to general convex constraints by Dutta and Pandit [14]. While these methods use proximal steps, which can be computationally expensive, Solodov [49] proposed a more efficient gradient-based alternative. Helou and Simões [23] further generalized the framework to non-smooth settings via an $\epsilon$-subgradient method. These works show asymptotic convergence but lack convergence rates. Amini and Yousefian [1] analyzed an iterative regularized projected gradient method and established a convergence rate of $O(t^{-(1/2-b)})$ for the inner objective, assuming non-smooth $F$ and $G$, compact $X$, and strong convexity of $f$. Kaushik and Yousefian [30] later removed the strong convexity assumption and proved rates of $O(t^{-b})$ and $O(t^{-(1/2-b)})$ for the inner and outer objectives, respectively, for any $b \in (0, 1/2)$. Malitsky [35] studied an accelerated variant based on Tseng's method [53], achieving an $o(t^{-1})$ rate for the inner problem. More recently, Merchav et al. [37] proposed an accelerated proximal scheme with a $O(\log(t+1)/t)$ rate under smoothness assumption and $O(t^{-2})$ and $O(1/t^{2-\gamma})$ for inner and outer objectives, respectively, under a Hölderian error bound condition with modulus $\gamma \in (1, 2)$. Our work also follows an iterative regularization approach, but avoids expensive projections or proximal operations by leveraging linear optimization oracles.

**Projection-free methods.** Several important applications have domains $X$ where projection-based operations are inefficient, yet linear optimization over $X$ is easier. To address the limitations of projection-based methods, recent work has focused on projection-free bilevel optimization via linear optimization oracles. Jiang et al. [29] approximated $X_{\mathrm{opt}}$ by replacing the constraint $G(x) \leq G_{\mathrm{opt}}$ with a linear approximation, which removed the need for Slater's condition, and applying the conditional gradient method, refining the approximation at each step. They established $O(t^{-1})$ convergence rates for both objectives under smoothness and compactness assumptions in case $f$ is convex. Under non-convex setting, they claimed the convergence rates of $O(t^{-1/2})$ to the stationary point of the problem. However, their method requires a pre-specified tolerance parameter $\epsilon_g$, and only

guarantees $(\epsilon_g/2)$-infeasibility. Doron and Shtern [12] proposed an alternative approach based on sublevel sets of the outer function $F$. Their method performs conditional gradient updates over sets of the form $X \times \{x : F(x) \leq \alpha\}$, with a surrogate $\hat{G}_t$ that is updated iteratively. They achieved rates of $O(t^{-1})$ for the inner and $O(t^{-1/2})$ for the outer objective under composite structure in $G$ and an error bound on $F$. While linear optimization over $X$ may be efficient, this is not always true for sets like $X \cap H$, where $H$ is a halfspace, or $\{F(x) \leq \alpha\}$ unless $F$ has a special structure. To address this, Giang-Tran et al. [20] introduced a conditional gradient-based iterative regularization scheme that only requires linear optimization over $X$, albeit with slower rates: $O(t^{-p})$ and $O(t^{-(1-p)})$ for the inner and outer problems, respectively, for any $p \in (0, 1)$. We extend this framework to the stochastic setting, maintaining the same oracle-based reliance on linear optimization over $X$.

**Stochastic methods.** Stochastic bilevel optimization remains less explored. Jalilzadeh et al. [27] developed an iterative regularization-based stochastic extragradient algorithm, requiring $O(1/\epsilon_f^4)$ and $O(1/\epsilon_g^4)$ stochastic gradient queries to achieve $\epsilon$-optimality for both objectives. Cao et al. [4] extended the projection-free framework of Jiang et al. [29] to the stochastic setting. Their methods achieve $\tilde{O}(t^{-1})$ convergence rate when the noise distributions have finite support, and $\tilde{O}(t^{-1/2})$ convergence rates under sub-Gaussian noise. However, their method requires linear optimization over intersections of $X$ with a halfspace. In contrast, our work relaxes this requirement by relying solely on linear optimization over the base set $X$.

**Other methods and bilevel problem classes.** Several alternative approaches exist for simple bilevel optimization, including sequential averaging schemes [36, 44, 46], sublevel set-based methods [2, 6, 21, 58], accelerated algorithms [6, 7, 45, 54] and primal-dual strategies [47]. These methods are less related to our framework and do not address the stochastic setting. More general bilevel problems have seen considerable attention, particularly in Stackelberg game formulations where the upper and lower levels model leader-follower dynamics [8, 10, 19, 28, 31, 33, 34, 56]. Recent extensions include contextual bilevel optimization with exogenous variables [25, 51, 55] and problems with functional constraints at the lower level [30, 40]. Our focus in this paper, however, is restricted to the stochastic simple bilevel problems.

## 1.2 Contributions

We present two iterative regularization methods for solving problem (1) in both stochastic and finite-sum settings. The key idea is to introduce a decreasing regularization sequence $\{\sigma_t\}_{t \geq 1}$, and at each iteration $t$, perform a gradient-based update on the composite objective $\sigma_t F + G$. As $\sigma_t$ decreases, the algorithm gradually shifts focus from $F$ to $G$, thereby steering $x_t$ toward points in $X_{\text{opt}}$. At the same time the $\sigma_t F$ term encourages $x_t$ towards a point in $X_{\text{opt}}$ that minimizes $F$. In the stochastic formulation (1), computing exact gradients of the upper and lower objectives is computationally expensive. To address this, we employ sample-based gradient estimators. Specifically, we use the STOchastic Recursive Momentum (STORM) estimator [9] in the one-sample stochastic setting, and the Stochastic Path-Integrated Differential EstimatoR (SPIDER) [15, 38] in the finite-sum setting.

The key contributions of this paper are summarized below.

◇ **One-sample stochastic setting**: We develop a projection-free method for stochastic simple bilevel optimization that under convexity of $F$ on the base set $X$ achieves high-probability convergence rates of $O(t^{-1/4}\sqrt{\log(dt^2/\delta)})$ for the outer-level problem and $O(t^{-1/4}\sqrt{\log(d/\delta)})$ for the inner-level problem, with probability at least $1 - \delta$. Without convexity of $F$, the method enjoys the non-asymptotic convergence rates of $O(t^{-\frac{1}{7}}\log(dt^2/\delta))$ for both the outer- and inner-level objectives. Compared to the state-of-the-art approach by Cao et al. [4, Algorithm 1], which achieves the $O(t^{-1/2}\sqrt{\log(td/\delta)})$ rate under convex settings and $O(t^{-1/3}\sqrt{\log(td/\delta)})$ under nonconvex settings for both the outer and inner problems, our method offers several key advantages. First, it eliminates the need for optimization over intersections of the base domain with a halfspace, requiring only linear optimization over the original constraint set. Second, it removes the dependency on their pre-specified tolerance parameter $\epsilon_g$, and the associated computationally expensive initialization procedure to find an $\epsilon_g$-optimal starting point. Third, our algorithm enjoys *anytime* guarantees. Overall, the proposed algorithm is a simple, single-loop procedure that is significantly easier to implement than that of [4].

◇ **Finite-sum setting**: For problems with $n$ component functions, we establish convergence rates of $O(t^{-1/2}\sqrt{\log(t^2/\delta)})$ (outer) and $O(t^{-1/2}\sqrt{\log(1/\delta)})$ (inner) with probability $1 - \delta$. Without convexity of $F$, the method enjoys the non-asymptotic convergence rates of $O(t^{-\frac{1}{4}}\log(dt^2/\delta))$ for both the outer- and inner-level objectives. Although Cao et al. [4, Algorithm 2] achieves improved $O(t^{-1}\log(t)\sqrt{\log(t/\delta)})$ rates under convex settings and $O(t^{-1/2}\sqrt{\log(t/\delta)})$ rates under nonconvex settings for both objectives, it again relies on more computationally demanding halfspace-intersection oracles, requires an additional initialization step, depends on a fixed step-size, and does not offer anytime guarantees.

◇ **Numerical validation**: We demonstrate the practical efficiency of our methods on over-parametrized regression and dictionary learning tasks, showing significant speedups over existing projection-based and projection-free approaches. The experiments validate both the convergence rates and computational advantages of our framework.

The supplementary material is organized in the appendices as follows. Appendix A establishes the preparatory lemmas used throughout our analysis. Appendices B and C contain the detailed proofs for the results presented in Section 2 and Section 3, respectively. Appendix D provides additional material, including analyses of the convergence rate coefficients as well as details of the numerical implementations in Section 4.

## 2 Algorithm and convergence results for the one-sample stochastic setting

In this section, we introduce the Iteratively Regularized Stochastic Conditional Gradient (`IR-SCG`) method, summarized in Algorithm 1. This approach is a conditional gradient method that leverages the `STORM` estimator to progressively reduce noise variance via momentum-based updates, assuming sub-Gaussian noise. While the use of the `STORM` estimator in projection-free methods was first explored in [60], our `IR-SCG` algorithm offers several key advantages. Notably, unlike the method of Cao et al. [4], `IR-SCG` is an anytime algorithm that does not require careful initialization, preset step sizes, or knowledge of the total number of iterations $T$ in advance.

---

**Algorithm 1:** Iteratively Regularized Stochastic Conditional Gradient (`IR-SCG`) Algorithm.

**Data:** Parameters $\{\alpha_t\}_{t\geq 0} \subseteq [0,1], \{\sigma_t\}_{t\geq 0} \subseteq \mathbb{R}_{++}$.
**Result:** sequences $\{x_t, z_t\}_{t\geq 1}$.
Initialize $x_0 \in X$;
**for** $t = 0, 1, 2, \ldots$ **do**
    **if** $t = 0$ **then**
        Compute

$$\widehat{\nabla F}_t := \nabla_x f(x_t, \theta_t), \quad \widehat{\nabla G}_t := \nabla_x g(x_t, \xi_t)$$

    **else**
        Compute

$$\widehat{\nabla F}_t := (1 - \alpha_t)\widehat{\nabla F}_{t-1} + \nabla_x f(x_t, \theta_t) - (1 - \alpha_t)\nabla_x f(x_{t-1}, \theta_t)$$
$$\widehat{\nabla G}_t := (1 - \alpha_t)\widehat{\nabla G}_{t-1} + \nabla_x g(x_t, \xi_t) - (1 - \alpha_t)\nabla_x g(x_{t-1}, \xi_t)$$

    Compute

$$v_t \in \arg\min_{v \in X}\left\{\left(\sigma_t\widehat{\nabla F}_t + \widehat{\nabla G}_t\right)^\top v\right\}$$
$$x_{t+1} := x_t + \alpha_t(v_t - x_t)$$
$$S_{t+1} := (t+2)(t+1)\sigma_{t+1} + \sum_{i \in [t+1]}(i+1)i(\sigma_{i-1} - \sigma_i)$$
$$z_{t+1} := \frac{(t+2)(t+1)\sigma_{t+1}x_{t+1} + \sum_{i \in [t+1]}(i+1)i(\sigma_{i-1} - \sigma_i)x_i}{S_{t+1}}$$

---

To establish convergence rates for Algorithm 1, we impose the following standard assumptions on the simple bilevel problem (1). In the following, all norms refer to the Euclidean norm.

**Assumption 1.** *The following hold.*

(a) *$X \subseteq \mathbb{R}^d$ is convex and compact with diameter $D < \infty$, i.e., $\|x - y\| \leq D$ for any $x, y \in X$.*

(b) *Functions $F, G$ are convex over $X$ and continuously differentiable on an open neighborhood of $X$.*

(c) *For any $\theta$, $f(\cdot, \theta)$ is $L_f$-smooth on an open neighborhood of $X$, i.e., it is continuously differentiable and its derivative is $L_f$-Lipschitz:*

$$\|\nabla_x f(x, \theta) - \nabla_x f(y, \theta)\| \leq L_f \|x - y\|,$$

*for any $x, y \in X$.*

(d) *For any $\xi$, $g(\cdot, \xi)$ is $L_g$-smooth on an open neighborhood of $X$.*

(e) *For any $x \in X$, the stochastic gradients noise is sub-Gaussian, i.e., there exists some $\sigma_f, \sigma_g > 0$ such that*

$$\mathbb{E}\left[\exp\left(\|\nabla_x f(x, \theta) - \nabla F(x)\|^2 / \sigma_f^2\right)\right] \leq \exp(1),$$
$$\mathbb{E}\left[\exp\left(\|\nabla_x g(x, \xi) - \nabla G(x)\|^2 / \sigma_g^2\right)\right] \leq \exp(1).$$

First, we present the result for the case that $F$ is convex.

**Theorem 1.** *Let $\{z_t\}_{t \geq 1}$ be the iterates generated by Algorithm 1 with $\alpha_t = 2/(t + 2)$ for any $t \geq 0$, and regularization parameters $\sigma_t := \varsigma(t + 1)^{-p}$ for some chosen $\varsigma > 0$, $p \in (0, 1/2)$. Under Assumption 1, for any $t \geq 1$, with probability at least $1 - \delta$, it (jointly) holds that*

$$F(z_t) - F_{\mathrm{opt}} \leq O\left(t^{-(1/2-p)}\sqrt{\log(dt^2/\delta)}\right), \quad G(z_t) - G_{\mathrm{opt}} \leq O\left(t^{-p}\sqrt{\log(d/\delta)}\right).$$

*Moreover, with probability $1$, we have*

$$\lim_{t \to \infty} F(x_t) = F_{\mathrm{opt}}, \quad \lim_{t \to \infty} G(x_t) = G_{\mathrm{opt}}.$$

Theorem 1 provides the formal convergence guarantees for the IR-SCG algorithm in the convex setting. If we set $p = 1/4$, Algorithm 1 achieves an $\epsilon$-level optimality gap for both the outer and inner problems with a sample complexity of $O(1/\epsilon^4)$.

We next extend our analysis to the more general case where $F$ is not necessarily convex. In this setting, we require a different measure for *stationarity*. We define our stationarity measure using the Frank-Wolfe gap, which for a convex function $\mathcal{F}(x) = 0$ if and only if $x$ is optimal. This provides a natural generalization for the non-convex case. Accordingly, for all $x \in X$, we define the functions

$$\mathcal{F}(x) := \max_{v \in X_{\mathrm{opt}}} \nabla F(x)^\top (x - v), \quad \mathcal{G}(x) := G(x) - G_{\mathrm{opt}}. \tag{3}$$

To avoid clutter, we present the result using the optimally tuned parameters of Algorithm 1.

**Theorem 2.** *Let $\{x_t\}_{t \geq 0}$ denote the iterates generated by Algorithm 1 with stepsizes $\alpha_t = (t+1)^{-\omega}$ and regularization parameters $\sigma_t = \varsigma(t+1)^{-p}$ for any $t \geq 0$, with $p = \frac{2}{7}$ and $\omega = \frac{6}{7}$. If Assumption 1 holds with the exception that $F$ may be nonconvex, then with probability at least $1 - \delta$, for every $t \geq 0$, it (jointly) holds that*

$$\frac{\sum_{i=0}^t \sigma_i \mathcal{F}(x_i)}{\sum_{i=0}^t \sigma_i} \leq O\left(t^{-\frac{1}{7}} \log\left(dt^2/\delta\right)\right), \quad \mathcal{G}(x_t) \leq O\left(t^{-\frac{1}{7}} \log\left(dt^2/\delta\right)\right).$$

*Moreover, with probability $1$, we have*

$$\liminf_{t \to \infty} \mathcal{F}(x_t) = 0, \quad \lim_{t \to \infty} \mathcal{G}(x_t) = 0.$$

# 3 Algorithm and convergence results for the finite-sum setting

In this section, we address the case where the expectations in (1) are finite sums over $n$ components. We introduce the Iteratively Regularized Finite-Sum Conditional Gradient (`IR-FSCG`) method, summarized in Algorithm 2, where $[n] := \{1, \ldots, n\}$. This approach is a conditional gradient method that leverages the `SPIDER` estimator, achieving variance reduction through periodic gradient recomputations with mini-batches of size $q$. While the `SPIDER` estimator has been previously integrated into projection-free methods [22, 48, 57, 59], here we show how it can be effectively adapted to our regularized setting. The primary advantage is that, unlike the method of Cao et al. [4], `IR-FSCG` preserves the desirable anytime properties of our approach while achieving a faster convergence rate that capitalizes on the finite-sum structure. Additionally, it does not require careful initialization, preset step sizes, or knowledge of the total number of iterations $T$ in advance.

To establish its convergence, we again rely on Assumption 1, noting that condition Assumption 1(e) holds automatically in this finite-sum setting. First, we present the result when $F$ is convex.

**Theorem 3.** *Suppose $F$ and $G$ in (1) are expectations over uniform distributions on $[n]$. Let $\{z_t\}_{t \geq 1}$ be the iterates generated by Algorithm 2 with $\alpha_t = \log(q)/q$ for $0 \leq t < q$, $\alpha_t = 2/(t+2)$ for any $t \geq q$, and $\sigma_t := \varsigma(\max\{t, q\} + 1)^{-p}$ for some chosen $\varsigma > 0$, $p \in (0, 1)$ and $S = q$. Under Assumption 1, with probability at least $1 - \delta$, for any $t \geq q$, it holds that*

$$F(z_t) - F_{\text{opt}} \leq O\left(c(t,q)\, t^{-(1-p)} \sqrt{\log\left(t^2/\delta\right)}\right), \quad G(z_t) - G_{\text{opt}} \leq O\left(t^{-p} \sqrt{\log\left(1/\delta\right)}\right),$$

*where $c(t, q) = \max\{1, q\log(q)/t\}$. Moreover, with probability 1, we have*

$$\lim_{t \to \infty} F(x_t) = F_{\text{opt}}, \quad \lim_{t \to \infty} G(x_t) = G_{\text{opt}}.$$

---

**Algorithm 2:** Iteratively Regularized Finite-Sum Conditional Gradient (`IR-FSCG`) Algorithm.

---

**Data:** Parameters $\{\alpha_t\}_{t \geq 0} \subseteq [0, 1]$, $\{\sigma_t\}_{t \geq 0} \subseteq \mathbb{R}_{++}$, $S, q \in [n]$.
**Result:** sequences $\{x_t, z_t\}_{t \geq 1}$.
Initialize $x_0 \in X$;
**for** $t = 0, 1, 2, \ldots$ **do**
    **if** $t = 0 \mod q$ **then**
        Compute

$$\widehat{\nabla F}_t := \nabla F(x_t), \quad \widehat{\nabla G}_t := \nabla G(x_t).$$

    **else**
        Draw $S$ new i.i.d. samples $\mathcal{S}_f = \{\theta_1, \ldots, \theta_S\}$, $\mathcal{S}_g = \{\xi_1, \ldots, \xi_S\}$;
        Compute

$$\widehat{\nabla F}_t := \widehat{\nabla F}_{t-1} + \nabla f_{\mathcal{S}_f}(x_t) - \nabla f_{\mathcal{S}_f}(x_{t-1})$$
$$\widehat{\nabla G}_t := \widehat{\nabla G}_{t-1} + \nabla g_{\mathcal{S}_g}(x_t) - \nabla g_{\mathcal{S}_g}(x_{t-1}).$$

    Compute

$$v_t \in \arg\min_{v \in X} \left\{ \left(\sigma_t \widehat{\nabla F}_t + \widehat{\nabla G}_t\right)^{\top} v \right\}$$
$$x_{t+1} := x_t + \alpha_t(v_t - x_t)$$

    **if** $t \geq q$ **then**
        Compute

$$S_{t+1} := (t+2)(t+1)\sigma_{t+1} + \sum_{i \in [t+1] \setminus [q]} (i+1)i(\sigma_{i-1} - \sigma_i)$$
$$z_{t+1} := \frac{(t+2)(t+1)\sigma_{t+1}x_{t+1} + \sum_{i \in [t+1] \setminus [q]}(i+1)i(\sigma_{i-1} - \sigma_i)x_i}{S_{t+1}}.$$

---

Theorem 3 provides the formal convergence guarantees for the `IR-FSCG` algorithm in the convex setting. Moreover, when we set $S = q = \lfloor\sqrt{n}\rfloor$ and $p = 1/2$, Algorithm 2 achieves an $\epsilon$-level optimality gap for both the outer and inner problems with a sample complexity of $O(\sqrt{n}/\epsilon^2)$. Besides, Algorithm 2 can be readily extended when the outer- and inner-level problems involve different numbers of component functions ($n_f \neq n_g$) by introducing $(S_f, q_f)$ for the outer level and $(S_g, q_g)$ for the inner level. Details are omitted for brevity.

We next extend our analysis to the more general case where $F$ is not necessarily convex, using the auxiliary function (3) to assess stationarity. To avoid clutter, we present the result using the optimally tuned parameters of Algorithm 2.

**Theorem 4.** *Let $\{x_t\}_{t\geq0}$ denote the iterates generated by Algorithm 2 with stepsizes $\alpha_t = \log(q + 1)/(q + 1)$ for any $0 \leq t \leq q$, $\alpha_t = 1/(t + 1)^\omega$ for any $t > q$ and $S = q$, and regularization parameters $\sigma_t = \varsigma(\max\{t, q + 1\} + 1)^{-p}$ with $p = \frac{1}{2}$ and $w = \frac{3}{4}$. If Assumption 1 holds with the exception that $F$ may be nonconvex, then with probability at least $1 - \delta$, for every $t \geq q$, it (jointly) holds that*

$$\frac{\sum_{i=0}^t \beta_i \mathcal{F}(x_i)}{\sum_{i=0}^t \beta_i} \leq O\left(t^{-\frac{1}{4}} \log\left(dt^2/\delta\right)\right), \quad \mathcal{G}(x_t) \leq O\left(t^{-\frac{1}{4}} \log\left(dt^2/\delta\right)\right),$$

*where*

$$\beta_i = \begin{cases} \dfrac{\sigma_0(q + 1)^\omega \log(q + 1)}{q + 1}\left(1 - \dfrac{\log(q + 1)}{q + 1}\right)^{q-i} & 0 \leq i \leq q, \\ \sigma_i & i > q. \end{cases} \tag{4}$$

*Moreover, with probability 1, we have*

$$\liminf_{t\to\infty} \mathcal{F}(x_t) = 0, \quad \lim_{t\to\infty} \mathcal{G}(x_t) = 0.$$

## 4 Numerical results

We showcase the performance of our proposed algorithms in an over-parametrized regression and a dictionary learning problems. For performance comparison, we implement four algorithms: `SBCGI` [4, Algorithm 1], `SBCGF` [4, Algorithm 2], `aR-IP-SeG` [27], and the stochastic variant of the dynamic barrier gradient descent (SDBGD) [21]. All implementation details follow those in the corresponding papers and available in Appendix D. To ensure reproducibility, all source codes are made available at https://github.com/brucegiang/CG-StoBilvl.

### 4.1 Over-parameterized regression

We first consider a simple bilevel optimization problem with a convex outer-level objective function

$$\min_{x\in\mathbf{R}^d} F(x) := \frac{1}{|\mathcal{D}_{\mathrm{val}}|}\|A_{\mathrm{val}}x - b_{\mathrm{val}}\|_2^2 \quad \text{s.t.} \quad x \in \arg\min_{\|z\|_1 \leq \delta} G(z) := \frac{1}{|\mathcal{D}_{\mathrm{tr}}|}\|A_{\mathrm{tr}}z - b_{\mathrm{tr}}\|_2^2, \tag{5}$$

where the goal is to minimize the validation loss by choosing among optimal solutions of the training loss constrained by the $\ell_1$-norm ball. We use the same training and validation datasets, $(A_{\mathrm{tr}}, b_{\mathrm{tr}})$ and $(A_{\mathrm{val}}, b_{\mathrm{val}})$, from the Wikipedia Math Essential dataset [43], as in [4]. This dataset consists of $n = 1068$ samples and $d = 730$ features.

We evaluate the performance of different algorithms across 10 experiments with random initializations and report their average performance within a 4-minute execution limit. Figure 1 shows the convergence behavior of different algorithms for the inner-level (left) and outer-level (right) optimality gaps in terms of execution time. Our proposed stochastic methods outperform existing approaches. In particular, `IR-FSCG` achieves high-accuracy solutions for the outer-level objective while maintaining strong inner-level feasibility, surpassing its mini-batch counterpart, `SBCGF`. Similarly, `IR-SCG` outperforms its single-sample counterpart, `SBCGI`. We also observe that both `SBCGI` and `SBCGF` exhibit degradation in their inner-level optimality gaps over time. This behavior aligns with the theoretical guarantees in [4], where the inner-level convergence can only be bounded by $\epsilon_g/2$ without monotonic improvement guarantees. Furthermore, neither `aR-IP-SeG` nor SDBGD makes significant progress in optimizing the outer-level objective function.

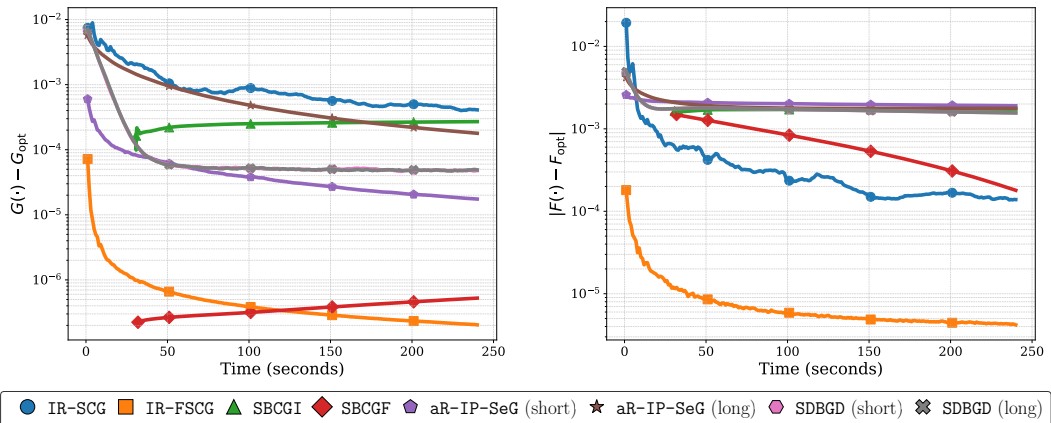

Figure 1: The inner-level optimality gap (left) and the outer-level absolute optimality gap (right) over time for different algorithms on the over-parametrized regression problem (5).

The table below also reports the average number of iterations and stochastic oracle calls over the 4-minute time limit for each algorithm. Our projection-free method completes more iterations within the fixed time budget, benefiting from a simpler linear minimization oracle compared to the projection-free approach in [4]. We also observe that SDBGD and IR-SCG perform a similar number of iterations, as the $\ell_1$-norm projection oracle has comparable computational cost to the linear minimization oracle.

| Method | Average # of iterations | Average # of oracle calls |
|---|---|---|
| IR-SCG | 1182617.0 | 2365234.0 |
| IR-FSCG | 369952.5 | 28240425.0 |
| SBCGI | 6726.9 | 7428681.9 |
| SBCGF | 6713.4 | 7935076.2 |
| aR-IP-SeG (short step) | 807700.2 | 3230800.8 |
| aR-IP-SeG (long step) | 799725.0 | 3199300.0 |
| SDBGD (short step) | 1250392.6 | 2500782.2 |
| SDBGD (long step) | 1243720.1 | 2487440.2 |

## 4.2 Dictionary learning

In the dictionary learning problem, the goal is to learn a compact representation of the input data $A = \{a_1, \ldots, a_n\} \subseteq \mathbf{R}^m$. Formally, we aim to find a dictionary $D = [d_1 \cdots d_p] \in \mathbf{R}^{m \times p}$ such that each data point $a_i$ can be approximated by a linear combination of the basis vectors in $D$. This leads to the following optimization problem

$$\min_{D,X} \quad \frac{1}{2n} \sum_{i \in [n]} \|a_i - Dx_i\|_2^2$$

$$\text{s.t.} \quad D \in \mathbf{R}^{m \times p}, \; X = [x_1 \cdots x_n] \in \mathbf{R}^{p \times n}$$
$$\|d_j\|_2 \leq 1, \quad \forall j \in [p], \quad \|x_i\|_1 \leq \delta, \quad \forall i \in [n],$$

where we refer to $X$ as the coefficient matrix. In practice, data points usually arrive sequentially and the representation evolves gradually. Hence, the dictionary must be updated sequentially as well. Assume that we already have learned a dictionary $\hat{D} \in \mathbf{R}^{m \times p}$ and the corresponding coefficient matrix $\hat{X} \in \mathbf{R}^{p \times n}$ for some data set $A$. As a new dataset $A' = \{a'_1, \ldots, a'_{n'}\} \subseteq \mathbf{R}^m$ arrives, we intend to enrich our dictionary by learning $\tilde{D} \in \mathbf{R}^{m \times q}$ ($q > p$) and the coefficient matrix $\tilde{X} \in \mathbf{R}^{q \times n'}$ for the new dataset while maintaining good performance of $\tilde{D}$ on the old dataset $A$ as well as the learned coefficient matrix $\hat{X}$.

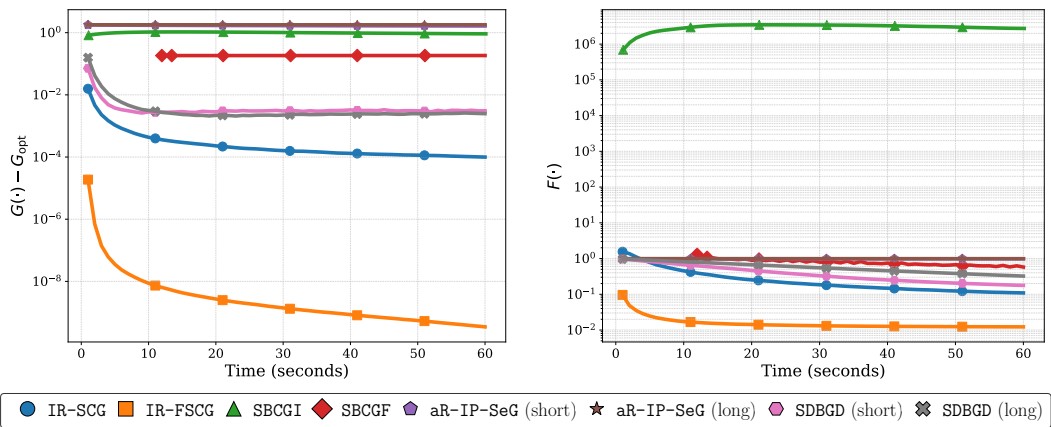

Figure 2: The inner-level optimality gap (left) and the outer-level objective function (right) over time for different algorithms on the dictionary learning problem (6).

This leads to the following simple bilevel problem with a nonconvex outer-level objective function

$$\min_{\tilde{D}, \tilde{X}} \quad F(\tilde{D}, \tilde{X}) := \frac{1}{2n'} \sum_{i \in [n']} \|a_i' - \tilde{D}\tilde{x}_i\|_2^2$$

$$\text{s.t.} \quad \tilde{D} \in \mathbf{R}^{m \times q}, \ \tilde{X} = [\tilde{x}_1 \cdots \tilde{x}_n] \in \mathbf{R}^{q \times n'}, \quad \|\tilde{x}_i\|_1 \leq \delta, \quad \forall i \in [n'], \tag{6}$$

$$\tilde{D} \in \operatorname*{arg\,min}_{\|z_j\|_2 \leq 1, \forall j \in [q]} g(Z) := \frac{1}{2n} \sum_{i \in [n]} \|a_i - Z\hat{x}_i\|_2^2,$$

where we denote $\hat{x}_i$ as the prolonged vector in $\mathbf{R}^q$ by appending zeros at the end. We consider problem (6) on a synthetic dataset with a similar setup to [4, 29].

We evaluate the performance of different algorithms across 10 experiments with random initializations and report their average performance within a 1-minute execution limit. Figure 2 shows the convergence behavior of different algorithms for the inner-level (left) and outer-level (right) problems in terms of execution time. Our proposed stochastic methods outperform existing approaches. In particular, IR-FSCG achieves better solutions for the outer-level objective while maintaining strong inner-level feasibility, surpassing its mini-batch counterpart, SBCGI. Similarly, IR-SCG outperforms its single-sample counterpart, SBCGI. We again observe that both SBCGI and SBCGF exhibit degradation in their inner-level optimality gaps over time.

The table below reports the average number of iterations and stochastic oracle calls over the 1-minute time limit for each algorithm. Our projection-free method completes more iterations within the fixed time budget, benefiting from a simpler linear minimization oracle compared to the projection-free methods in [4]. We also observe that SDBGD performs fewer iterations than IR-SCG, as it relies on a more costly projection oracle.

| Method | Average # of iterations | Average # of oracle calls |
|---|---|---|
| IR-SCG | 32736.1 | 65472.2 |
| IR-FSCG | 30532.6 | 1872851.0 |
| SBCGI | 466.3 | 7440.6 |
| SBCGF | 361.84 | 6147916.0 |
| aR-IP-SeG (short step) | 6290.0 | 25160.0 |
| aR-IP-SeG (long step) | 6278.1 | 25112.4 |
| SDBGD (short step) | 11441.3 | 22882.6 |
| SDBGD (long step) | 12239.3 | 24478.6 |

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

# A Preparatory Lemmas

Throughout all appendices, we employ the notation $\Phi_t(x) := \sigma_t F(x) + G(x)$ and $\widehat{\nabla\Phi}_t := \sigma_t \widehat{\nabla F}_t + \widehat{\nabla G}_t$ to denote the gradient estimate at step $t$. We also denote $x_{\text{opt}} \in X_{\text{opt}}$ as an optimal solution to (1). All norms refer to the Euclidean norm.

For any sequence $\{x_t\}_{t \geq 0}$ generated by either Algorithm 1 or Algorithm 2, we observe that by construction, $x_0, v_t \in X$. Hence, under convexity of $X$, $x_{t+1} \in X$ as it is a convex combination of $x_t$ and $v_t$, and $X$ is assumed to be convex. By a simple induction argument, one can easily show that the sequence $\{x_t\}_{t \geq 0} \in X$.

## A.1 Convex outer-level

**Lemma 1.** *Let $\{x_t\}_{t \geq 0}$ denote the iterates generated by Algorithm 1 or Algorithm 2 with some given stepsizes $\{\alpha_t\}_{t \geq 0}$. If Assumption 1 holds, then for every $t \geq k$, we have*

$$\Phi_t(x_{t+1}) - \Phi_t(x_{\text{opt}}) \leq (1 - \alpha_t)(\Phi_t(x_t) - \Phi_t(x_{\text{opt}})) + \frac{(L_f \sigma_t + L_g)D^2 \alpha_t^2}{2} + 2\alpha_t D \left\| \nabla\Phi_t(x_t) - \widehat{\nabla\Phi}_t \right\|,$$

*Proof of Lemma 1.* Recall that

$$x_{t+1} = x_t + \alpha_t(v_t - x_t), \quad \text{where} \quad v_t \in \arg\min_{v \in X} \left\{ \left( \widehat{\nabla\Phi}_t \right)^\top v \right\}.$$

As both functions $F$ and $G$ are assumed to be convex and smooth, it is easy to show that $\Phi$ is convex and $\sigma_t L_f + L_g$-smooth. By smoothness of $\Phi_t$ and the assumption that $\text{diam}(X) = D$, one can show that

$$\Phi_t(x_{t+1}) \leq \Phi_t(x_t) + \alpha_t \nabla\Phi_t(x_t)^\top (v_t - x_t) + \frac{(\sigma_t L_f + L_g)\alpha_t^2}{2} \|v_t - x_t\|^2$$

$$\leq \Phi_t(x_t) + \alpha_t \nabla\Phi_t(x_t)^\top (v_t - x_t) + \frac{(\sigma_t L_f + L_g)D^2 \alpha_t^2}{2}.$$

Note that for any $w_t \in X$ and any gradient estimator $\widehat{\nabla\Phi}_t$, we have

$$\nabla\Phi_t(x_t)^\top (v_t - x_t)$$

$$= (\nabla\Phi_t(x_t) - \widehat{\nabla\Phi}_t)^\top (v_t - x_t) + \widehat{\nabla\Phi}_t^\top (v_t - x_t)$$

$$\leq (\nabla\Phi_t(x_t) - \widehat{\nabla\Phi}_t)^\top (v_t - x_t) + \widehat{\nabla\Phi}_t^\top (w_t - x_t)$$

$$= (\nabla\Phi_t(x_t) - \widehat{\nabla\Phi}_t)^\top (v_t - x_t) + (\widehat{\nabla\Phi}_t - \nabla\Phi_t(x_t))^\top (w_t - x_t) + \nabla\Phi_t(x_t)^\top (w_t - x_t)$$

$$\leq 2D \|\widehat{\nabla\Phi}_t - \nabla\Phi_t(x_t)\| + \nabla\Phi_t(x_t)^\top (w_t - x_t),$$

where the first inequality follows from the definition of $v_t$, and the second inequality uses the Cauchy-Schwarz inequality, and the assumption that $\text{diam}(X) = D$. Thus, by smoothness of $\Phi_t$, for any $w_t \in X$ and any gradient estimator $\widehat{\nabla\Phi}_t$, we have

$$\Phi_t(x_{t+1}) \leq \Phi_t(x_t) + \alpha_t \nabla\Phi_t(x_t)^\top (w_t - x_t) + 2D\alpha_t \|\widehat{\nabla\Phi}_t - \nabla\Phi_t(x_t)\| + \frac{(\sigma_t L_f + L_g)D^2 \alpha_t^2}{2}. \quad (7)$$

and set $w_t = x_{\text{opt}}$. Thanks to the convexity of $\Phi_t$, one can easily show that $\nabla\Phi_t(x_t)^\top (x_{\text{opt}} - x_t) \leq \Phi_t(x_{\text{opt}}) - \Phi_t(x_t)$. Thus, we obtain

$$\Phi_t(x_{t+1}) - \Phi_t(x_{\text{opt}})$$

$$\leq \Phi_t(x_t) - \Phi_t(x_{\text{opt}}) + \alpha_t \left( \Phi_t(x_{\text{opt}}) - \Phi_t(x_t) + 2D\|\widehat{\nabla\Phi}_t - \nabla\Phi_t(x_t)\| \right) + \frac{(\sigma_t L_f + L_g)D^2 \alpha_t^2}{2}$$

$$= (1 - \alpha_t)(\Phi_t(x_{\text{opt}}) - \Phi_t(x_t)) + 2\alpha_t D\|\widehat{\nabla\Phi}_t - \nabla\Phi_t(x_t)\| + \frac{(\sigma_t L_f + L_g)D^2 \alpha_t^2}{2}.$$

The proof concludes by using the definition of the stepsizes $\alpha_t$. $\qquad\square$

**Lemma 2.** *Let $\{x_t\}_{t\geq 0}$ denote the iterates generated by Algorithm 1 or Algorithm 2 with stepsizes $\{\alpha_t\}_{t\geq 0}$ given by $\alpha_t = 2/(t+2)$ for any $t \geq k$. If Assumption 1 holds, then for each $t \geq k$, we have*

$$t(t+1)\left(\Phi_t(x_t) - \Phi_t(x_{\text{opt}})\right) + \sum_{i\in[t]\setminus[k]} (i+1)i(\sigma_{i-1} - \sigma_i)(F(x_i) - F_{\text{opt}})$$

$$\leq (k+1)k\left(\Phi_k(x_k) - \Phi_k(x_{\text{opt}})\right) + 2(\sigma_0 L_f + L_g)D^2(t-k) \quad (8)$$

$$+ \sum_{i\in[t]\setminus[k]} 4iD\|\widehat{\nabla\Phi}_{i-1} - \nabla\Phi_{i-1}(x_{i-1})\|.$$

*Proof of Lemma 2.* By definition of $\Phi_t$, we have

$$\Phi_t(x_{t+1}) - \Phi_t(x_{\text{opt}}) = \sigma_{t+1}(F(x_{t+1}) - F_{\text{opt}}) + G(x_{t+1}) - G_{\text{opt}} + (\sigma_t - \sigma_{t+1})(F(x_{t+1}) - F_{\text{opt}})$$

$$= \Phi_{t+1}(x_{t+1}) - \Phi_{t+1}(x_{\text{opt}}) + (\sigma_t - \sigma_{t+1})(F(x_{t+1}) - F_{\text{opt}}).$$

To simplify the derivations, we introduce the shorthands $h_t := \Phi_t(x_t) - \Phi_t(x_{\text{opt}})$ and

$$\eta_t := \frac{2(L_f\sigma_t + L_g)D^2}{(t+2)^2} + \frac{4D}{t+2}\|\widehat{\nabla\Phi}_t - \nabla\Phi_t(x_t)\| - (\sigma_t - \sigma_{t+1})(F(x_{t+1}) - F_{\text{opt}}).$$

Applying Lemma 1 with stepsize $\alpha_t = 2/(t+2)$ for all $t \geq k$ gives the recursion $h_{t+1} \leq \frac{t}{t+2}h_t + \eta_t$ for any $t \geq k$. This can be re-expressed as

$$(t+1)th_t \leq t(t-1)h_{t-1} + t(t+1)\eta_{t-1}$$

for any $t \geq k+1$. Thus, for $t \geq k+1$, it holds that

$$(t+1)th_t - (k+1)kh_k = \sum_{i\in[t]\setminus[k]} \left((i+1)ih_i - (i-1)ih_{i-1}\right) \leq \sum_{i\in[t]\setminus[k]} (i+1)i\eta_{i-1}.$$

By definition of $h_t$ and $\eta_t$, and using the inequalities $i/(i+1) < 1$ and $\sigma_{i-1} \leq \sigma_0$ for $i \in [t] \setminus [k]$, we arrive at

$$(t+1)t\left(\Phi_t(x_t) - \Phi_t(x_{\text{opt}})\right) - (k+1)k\left(\Phi_k(x_k) - \Phi_k(x_{\text{opt}})\right)$$

$$\leq 2(\sigma_0 L_f + L_g)D^2(t-k) - \sum_{i\in[t]\setminus[k]}(i+1)i(\sigma_{i-1} - \sigma_i)(F(x_i) - F_{\text{opt}})$$

$$+ \sum_{i\in[t]\setminus[k]} 4iD\|\widehat{\nabla\Phi}_{i-1} - \nabla\Phi_{i-1}(x_{i-1})\|,$$

which implies (8). This completes the proof. □

**Proposition 1.** *Let $\{x_t\}_{t\geq 0}$ denote the iterates generated by Algorithm 1 or Algorithm 2 with stepsizes $\{\alpha_t\}_{t\geq 0}$ given by $\alpha_t = 2/(t+2)$ for any $t \geq k$. If Assumption 1 holds, then for each $t \geq k$, we have*

$$G(x_t) - G_{\text{opt}} \leq \frac{(k+1)k\left(\Phi_k(x_k) - \Phi_k(x_{\text{opt}})\right)}{t(t+1)} + \frac{2(\sigma_0 L_f + L_g)D^2(t-k)}{t(t+1)}$$

$$+ \sum_{i\in[t]\setminus[k]} \frac{4iD\|\widehat{\nabla\Phi}_{i-1} - \nabla\Phi_{i-1}(x_{i-1})\|}{t(t+1)} \quad (9)$$

$$+ \left(F_{\text{opt}} - \min_{x\in X} F(x)\right)\left(\sigma_t + \frac{\sum_{i\in[t]}(i+1)i(\sigma_{i-1} - \sigma_i)}{t(t+1)}\right),$$

*and*

$$(t+1)t\sigma_t(F(x_t) - F_{\text{opt}}) + \sum_{i\in[t]\setminus[k]} (i+1)i(\sigma_{i-1} - \sigma_i)(F(x_i) - F_{\text{opt}})$$

$$\leq (k+1)k\left(\Phi_k(x_k) - \Phi_k(x_{\text{opt}})\right) + 2(\sigma_0 L_f + L_g)D^2(t-k) \quad (10)$$

$$+ \sum_{i\in[t]\setminus[k]} 4iD\|\widehat{\nabla\Phi}_{i-1} - \nabla\Phi_{i-1}(x_{i-1})\|.$$

*Proof of Proposition 1.* By definition of the function $\Phi_t$, the first term in (8) satisfies

$$t(t+1)\left(\Phi_t(x_t) - \Phi_t(x_{\text{opt}})\right) = t(t+1)\left(G(x_t) - G(x_{\text{opt}})\right) + t(t+1)\sigma_t\left(F(x_t) - F(x_{\text{opt}})\right). \quad (11)$$

Since $x_t \in X$, we have $\min_{x \in X} F(x) - F_{\text{opt}} \leq F(x_t) - F_{\text{opt}}$ for all $t \geq 0$. Using this bound together with the identity in (11), we can further lower bound the left-hand side of inequality (8), and arrive at the bound (9). This completes the proof of the first claim.

For the second claim, since $\{x_t\}_{t\geq 0} \in X$ is feasible in the lower-level problem, it follows directly that $G(x_t) \geq G_{\text{opt}}$. Combining this inequality with (11) and (8), we obtain (10), which concludes the proof. $\qquad\square$

### A.2  Nonconvex outer-level

When $F$ is possibly nonconvex, we start by establishing a bound, akin to Lemma 2.

**Lemma 3.** *Let $\{x_t\}_{t\geq 0}$ denote the iterates generated by Algorithm 1 or Algorithm 2 with $\{\alpha_t, \sigma_t\}_{t\geq 0}$ given by $\alpha_t = (t+1)^{-\omega}$ and $\sigma_t = \varsigma(t+1)^{-p}$ for $t \geq k$, with $0 < p \leq \omega$. If Assumption 1 holds with the exception that $F$ may be nonconvex, then for every $t \geq k$, we have*

$$(t+1)^{\omega}\mathcal{G}(x_{t+1}) - k^{\omega}\mathcal{G}(x_k) \leq 2\varsigma\sup_{z\in X}|F(z)|(t+1)^{\omega-p} + \sigma_k F(x_k) - \sum_{i=k}^{t}\sigma_i\mathcal{F}(x_i)$$

$$+ \sum_{i=k}^{t}\left(2D\left\|\widehat{\nabla\Phi}_i - \nabla\Phi_i(x_i)\right\| + \frac{(\sigma_i L_f + L_g)D^2}{2(i+1)^{\omega}}\right).$$

*Proof of Lemma 3.* Using the definition of $\Phi_t = \sigma_t F + G$, for any $t \geq k$, we have

$$\sigma_t F(x_{t+1}) + G(x_{t+1}) \leq \sigma_t F(x_t) + G(x_t) + \alpha_t \min_{v\in X}(\sigma_t \nabla F(x_t) + \nabla G(x_t))^{\top}(v - x_t)$$

$$+ 2D\alpha_t\|\widehat{\nabla\Phi}_t - \nabla\Phi_t(x_t)\| + \frac{(\sigma_t L_f + L_g)D^2\alpha_t^2}{2}$$

where the inequality follows directly from (7) and by minimizing over $w_t \in X$. We emphasize that (7) does not involve any convexity assumptions on $\Phi_t$ and relies solely on its smoothness. Furthermore, notice that for any $x \in X$,

$$\min_{v\in X}(\sigma\nabla F(x) + \nabla G(x))^{\top}(v - x) \leq \min_{v\in X_{\text{opt}}}(\sigma\nabla F(x) + \nabla G(x))^{\top}(v - x)$$

$$\leq \min_{v\in X_{\text{opt}}}\left\{\sigma\nabla F(x)^{\top}(v - x) + G(v) - G(x)\right\}$$

$$= -\sigma\mathcal{F}(x) - \mathcal{G}(x),$$

where the last equality follows from the definition of $\mathcal{F}$ and $\mathcal{G}$. Using the above bound, we may thus conclude that

$$\sigma_t F(x_{t+1}) + G(x_{t+1}) \leq \sigma_t F(x_t) + G(x_t) - \alpha_t(\sigma_t\mathcal{F}(x_t) + \mathcal{G}(x_t))$$

$$+ 2D\alpha_t\|\widehat{\nabla\Phi}_t - \nabla\Phi_t(x_t)\| + \frac{(\sigma_t L_f + L_g)D^2\alpha_t^2}{2}.$$

A simple re-arrangement and using the definition of $\mathcal{G}$ then yields the bound

$$\alpha_t(\sigma_t\mathcal{F}(x_t) + \mathcal{G}(x_t)) \leq \sigma_t(F(x_t) - F(x_{t+1})) + \mathcal{G}(x_t) - \mathcal{G}(x_{t+1})$$

$$+ 2D\alpha_t\|\widehat{\nabla\Phi}_t - \nabla\Phi_t(x_t)\| + \frac{(\sigma_t L_f + L_g)D^2\alpha_t^2}{2}. \quad (12)$$

Using the definition of $\alpha_t$ and the fact that $(t+1)^{\omega} \leq t^{\omega} - 1$ for any $t \geq 0$, we arrive at

$$(t+1)^{\omega}\mathcal{G}(x_{t+1}) \leq t^{\omega}\mathcal{G}(x_t) + (t+1)^{\omega}\sigma_t(F(x_t) - F(x_{t+1})) - \sigma_t\mathcal{F}(x_t)$$

$$+ 2D\|\widehat{\nabla\Phi}_t - \nabla\Phi_t(x_t)\| + \frac{(\sigma_t L_f + L_g)D^2}{2(t+1)^{\omega}}.$$

Unwinding this recursion back to $i = k$, we obtain

$$(t+1)^\omega \mathcal{G}(x_{t+1}) \leq k^\omega \mathcal{G}(x_k) + \sum_{i=k}^{t} \left( (i+1)^\omega \sigma_i (F(x_i) - F(x_{i+1})) \right) - \sum_{i=k}^{t} \sigma_i \mathcal{F}(x_i)$$

$$+ \sum_{i=k}^{t} \left( 2D\|\widehat{\nabla \Phi}_i - \nabla \Phi_i(x_i)\| + \frac{(\sigma_i L_f + L_g)D^2}{2(i+1)^\omega} \right).$$

Since $p \leq \omega$, we have $(i+1)^\omega \sigma_i - i^\omega \sigma_{i-1} \geq 0$. Therefore, we deduce that

$$\sum_{i=k}^{t} (i+1)^\omega \sigma_i (F(x_i) - F(x_{i+1}))$$

$$= \sum_{i=k}^{t} \left( F(x_i)((i+1)^\omega \sigma_i - i^\omega \sigma_{i-1}) \right) + \sigma_k F(x_k) - (t+1)^\omega \sigma_t F(x_{t+1})$$

$$\leq \sup_{z \in X} |F(z)| \sum_{i=k}^{t} \left( ((i+1)^\omega \sigma_i - i^\omega \sigma_{i-1}) \right) + \sigma_k F(x_k) + (t+1)^\omega \sigma_t \sup_{z \in X} |F(z)|$$

$$\leq 2\varsigma \sup_{z \in X} |F(z)|(t+1)^{\omega-p} + \sigma_k F(x_k).$$

This completes the proof. $\qquad\qquad\square$

To establish asymptotic convergence of the proposed methods under nonconvex outer-level, we rely on an observation that function $\mathcal{F}$ is Lipschitz continuous on $X$.

**Lemma 4.** *If Assumption 1 holds, the function $\mathcal{F}$ as defined in (3) is $L_f D + \max_{z \in X} \|\nabla F(z)\|$-Lipschitz continuous over $X$.*

*Proof of Lemma 4.* Given $x, y \in X$, we observe that

$$\mathcal{F}(y) = \max_{v \in X_{\mathrm{opt}}} \left\{ \nabla F(y)^\top (y - v) \right\}$$

$$= \max_{v \in X_{\mathrm{opt}}} \{ (\nabla F(y) - \nabla F(x))^\top (y - v) + \nabla F(x)^\top (y - x) + \nabla F(x)^\top (x - v) \}$$

$$= \max_{v \in X_{\mathrm{opt}}} \{ (\nabla F(y) - \nabla F(x))^\top (y - v) + \nabla F(x)^\top (y - x) + \nabla F(x)^\top (x - v) \}$$

$$\leq \max_{v \in X_{\mathrm{opt}}} \{ \|\nabla F(y) - \nabla F(x)\| \|y - v\| + \nabla F(x)^\top (y - x) + \nabla F(x)^\top (x - v) \}$$

$$\leq \max_{v \in X_{\mathrm{opt}}} \{ L_f \|x - y\| D + \nabla F(x)^\top (y - x) + \nabla F(x)^\top (x - v) \}$$

$$= L_f D \|x - y\| + \nabla F(x)^\top (y - x) + \mathcal{F}(x)$$

$$\leq \left( L_f D + \max_{z \in X} \|\nabla F(z)\| \right) \|y - x\| + \mathcal{F}(x),$$

where the first and third inequality follows from Cauchy-Schwartz inequality, while the second inequality follows from the $L_f$-smoothness of $F$.

By interchanging the role of $y$ and $x$, we deduce that

$$|\mathcal{F}(y) - \mathcal{F}(x)| \leq \left( L_f D + \max_{z \in X} \|\nabla F(z)\| \right) \|y - x\|.$$

This concludes the proof. $\qquad\qquad\square$

# B  Proofs of Section 2

## B.1  Proof of Theorem 1

We restate Theorem 1 with exact upper bounds on sub-optimality gaps.

**Theorem 1.** *Let $\{z_t\}_{t\geq 1}$ be the iterates generated by Algorithm 1 with $\alpha_t = 2/(t+2)$ for any $t \geq 0$, and regularization parameters $\{\sigma_t\}_{t\geq 0}$ given by $\sigma_t := \varsigma(t+1)^{-p}$ for some chosen $\varsigma > 0$, $p \in (0, 1/2)$. Under Assumption 1, for any $t \geq 1$, with probability at least $1 - \delta$, we have*

$$F(z_t) - F_{opt} \leq \frac{2(L_f + L_g/\varsigma)D^2}{(t+1)^{1-p}} + \frac{4c\left(4(L_f + L_g/\varsigma)D + 3(\sigma_f + \sigma_g/\varsigma)\right)}{(t+1)^{1/2-p}}\sqrt{\log\left(8dt^2/\delta\right)},$$

*and*

$$
\begin{aligned}
&G(z_t) - G_{opt} \\
&\leq \left(1 + \frac{2p}{\min\{1, 2(1-p)\}}\right)\left[(1+2p)\left(F_{opt} - \min_{x\in X}F(x)\right)\varsigma + 2(\varsigma L_f + L_g)D^2 \right.\\
&\left. + 4c\left(8\left(L_f\varsigma + L_g\right)D^2 + 3\left(\sigma_f\varsigma + \sigma_g\right)D\right)\sqrt{\frac{2}{(1-2p)e} + \log\left(8d/\delta\right)}\right](t+1)^{-p},
\end{aligned}
$$

*where $c$ is some absolute constant. Moreover, with probability 1, we have*

$$\lim_{t\to\infty} F(x_t) = F_{\mathrm{opt}}, \quad \lim_{t\to\infty} G(x_t) = G_{\mathrm{opt}}.$$

The proof proceeds by applying Proposition 1 with $k = 0$. In this case, the main random component in (9) and (10) is the term $\sum_{i\in[t]} 4iD\|\widehat{\nabla\Phi}_{i-1} - \nabla\Phi_{i-1}(x_{i-1})\|$. We first derive a probabilistic upper bound for this term, then carefully set the parameters in Algorithm 1 to establish the convergence rate for both the lower- and upper-level problems.

We begin by bounding the gradient estimator used in Algorithm 1.

**Lemma 5.** *Let $\{x_t\}_{t\geq 0}$ denote the iterates generated by Algorithm 1 with stepsize $\alpha_t = 2/(t+2)$ for any $t \geq 0$. If Assumption 1 holds, then for any $t \geq 1$, given $\delta \in (0,1)$, with probability at least $1 - \delta$, for some absolute constant $c$, it (jointly) holds that*

$$
\begin{aligned}
\|\widehat{\nabla F}_t - \nabla F(x_t)\| &\leq c\sqrt{2}\left(4L_f D + 3\sigma_f\right)\frac{\sqrt{\log(4d/\delta)}}{(t+1)^{1/2}} \\
\|\widehat{\nabla G}_t - \nabla G(x_t)\| &\leq c\sqrt{2}\left(4L_g D + 3\sigma_g\right)\frac{\sqrt{\log(4d/\delta)}}{(t+1)^{1/2}}.
\end{aligned}
\tag{13}
$$

*Proof of Lemma 5.* The proof closely follows the argument in [4, Lemma 4.1]. In particular, the bounds in [4, Lemma B.1] still hold under the modified stepsize $\alpha_t = 2/(t+2)$, instead of $1/(t+1)$. The constant inside the logarithmic term changes from 6 to 4, as we now apply a union bound over two events rather than three. Additionally, the constants of the upper bounds is doubled due to use of the modified stepsize. We omit the details for brevity. $\square$

**Proposition 2.** *Let $\{x_t\}_{t\geq 0}$ denote the iterates generated by Algorithm 1 with $\alpha_t = 2/(t+2)$ for any $t \geq 0$. If Assumption 1 holds and the parameters $\{\sigma_t\}_{t\geq 0}$ are non-increasing and positive, then given $\delta \in (0,1)$, with probability at least $1 - \delta$, for some absolute constant $c$ and any $t \geq 1$, we have*

$$
\begin{aligned}
&\sum_{i\in[t]} i\|\widehat{\nabla\Phi}_{i-1} - \nabla\Phi_{i-1}(x_{i-1})\| \\
&\leq c\left(4(\sigma_0 L_f + L_g)D + 3(\sigma_0\sigma_f + \sigma_g)\right)(t+1)^{3/2}\sqrt{\log\left(8dt^2/\delta\right)}.
\end{aligned}
$$

*Proof of Proposition 2.* Given $i \geq 1$ and $\delta \in (0,1)$, by Lemma 5, with probability at least $1 - \delta/i(i+1)$, we have

$$
\begin{aligned}
i\|\widehat{\nabla\Phi}_{i-1} - \nabla\Phi_{i-1}(x_{i-1})\| &\leq i(\sigma_i\|\widehat{\nabla F}_{i-1} - \nabla F(x_{i-1})\| + \|\widehat{\nabla G}_{i-1} - \nabla G(x_{i-1})\|) \\
&\leq i(\sigma_0\|\widehat{\nabla F}_{i-1} - \nabla F(x_{i-1})\| + \|\widehat{\nabla G}_{i-1} - \nabla G(x_{i-1})\|) \\
&\leq c\sqrt{2}\left(4(\sigma_0 L_f + L_g)D + 3(\sigma_0\sigma_f + \sigma_g)\right)\sqrt{i\log\left(4di(i+1)/\delta\right)}
\end{aligned}
$$

Thus, using the union bound, with probability at least $1 - \delta \sum_{i \geq 1} 1/i(i+1) = 1 - \delta$, for any $i \in [t]$, we have

$$\sum_{i \in [t]} i \| \widehat{\nabla \Phi}_{i-1} - \nabla \Phi_{i-1}(x_{i-1}) \| \leq \sum_{i \in [t]} c\sqrt{2} \left( 4(\sigma_0 L_f + L_g)D + 3(\sigma_0 \sigma_f + \sigma_g) \right) \sqrt{i \log\left(8dt^2/\delta\right)}$$

$$\leq c \left( 4(\sigma_0 L_f + L_g)D + 3(\sigma_0 \sigma_f + \sigma_g) \right) (t+1)^{3/2} \sqrt{\log\left(8dt^2/\delta\right)},$$

where the second inequality follows from bounding $\sum_{i \in [t]} \sqrt{i} \leq \frac{2}{3}(t+1)^{3/2}$ via the Riemann sum approximation, and observing that $2\sqrt{2} \leq 3$. $\qquad \square$

To establish the desired convergence results for Algorithm 1, we need to impose certain conditions on the regularization parameters $\{\sigma_t\}_{t \geq 0}$, stated below.

**Condition 1.** *The regularization parameters $\{\sigma_t\}_{t \geq 0}$ are non-increasing, positive, and converge to 0.*

**Condition 2.** *There exists $L \in \mathbb{R}$ such that*

$$\lim_{t \to \infty} t \left( \frac{\sigma_t}{\sigma_{t+1}} - 1 \right) = L.$$

**Condition 3.** *$(t+1)\sigma_{t+1}^2 > t\sigma_t^2$ for any $t \geq 0$, and $\log(t) = o(t\sigma_t^2)$ as $t \to \infty$.*

The following parameters will appear in our analysis and convergence rates

$$C := \sup_{t \geq 1} \left\{ \left( F_{\text{opt}} - \min_{x \in X} F(x) \right) \left( 1 + \frac{\sum_{i \in [t]}(i+1)i(\sigma_{i-1} - \sigma_i)}{(t+1)t\sigma_t} \right) + \frac{2(\sigma_0 L_f + L_g)D^2}{(t+1)\sigma_t} \right\}, \quad \text{(14a)}$$

$$\bar{C}_\delta := \sup_{t \geq 1} \left\{ \frac{8c\left(4(\sigma_0 L_f + L_g)D^2 + 3(\sigma_0 \sigma_f + \sigma_g)D\right)}{(t+1)^{1/2}\sigma_t} \sqrt{\log\left(8dt^2/\delta\right)} \right\}, \quad \text{(14b)}$$

$$C_\delta := \bar{C}_\delta + C, \quad \text{(14c)}$$

$$V := \sup_{t \geq 1} \left\{ \frac{\sum_{i \in [t]}(i+1)i(\sigma_{i-1} - \sigma_i)\sigma_i}{(t+1)t\sigma_t^2} \right\}. \quad \text{(14d)}$$

While it is not obvious a priori, Lemma 17 in Appendix D.1 guarantees that these quantities remain finite for any parameter choice satisfying Conditions 1–3. In particular, Conditions 1–3 hold when $\sigma_t = \varsigma(t+1)^{-p}$ for any $p \in (0,1)$, which is what is used in Theorems 1 and 3, and Lemma 8 provides explicit bounds for this case. We now analyze the sequence $\{x_t\}_{t \geq 0}$ in Algorithm 1 for the inner-level problem.

**Lemma 6.** *Let $\{x_t\}_{t \geq 0}$ denote the iterates generated by Algorithm 1 with $\alpha_t = 2/(t+2)$ for any $t \geq 0$, and let $C_\delta$ be defined as in (14a). If Assumption 1 and Conditions 1–3 hold, with probability at least $1 - \delta$, it holds that*

$$G(x_t) - G_{\text{opt}} \leq C_\delta \sigma_t, \quad \text{(15)}$$

*for any $t \geq 0$.*

*Proof of Lemma 6.* Combining (9) from Proposition 1 with the probabilistic bound in Proposition 2, and after some straightforward calculations, with probability at least $1 - \delta$, we have

$$G(x_t) - G_{\text{opt}} \leq \frac{8c\left(4(\sigma_0 L_f + L_g)D^2 + 3(\sigma_0 \sigma_f + \sigma_g)D\right)}{(t+1)^{1/2}} \sqrt{\log\left(8dt^2/\delta\right)}$$

$$+ \frac{2(\sigma_0 L_f + L_g)D^2}{(t+1)} + \left( F_{\text{opt}} - \min_{x \in X} F(x) \right) \left( \sigma_t + \frac{\sum_{i \in [t]}(i+1)i(\sigma_{i-1} - \sigma_i)}{(t+1)t} \right).$$

By definition of $C_\delta$, the right-hand side is at most $C_\delta \sigma_t$ for any $t \geq 1$. This completes the proof. $\quad \square$

We next analyze the convergence of the sequence $\{z_t\}_{t \geq 0}$ in Algorithm 1 for both outer- and inner-level problems.

**Lemma 7.** *Let $\{z_t\}_{t\geq 1}$ denote the iterates generated by Algorithm 1 with $\alpha_t = 2/(t+2)$ for any $t \geq 0$, and let $\bar{C}_\delta, V$ be defined as in (14). If Assumption 1 and Conditions 1–3 hold, then with probability at least $1 - \delta$, it (jointly) holds that*

$$F(z_t) - F_{\text{opt}} \leq \frac{2(\sigma_0 L_f + L_g)D^2}{(t+1)\sigma_t} + \frac{4c\left(4(\sigma_0 L_f + L_g)D + 3(\sigma_0\sigma_f + \sigma_g)\right)}{(t+1)^{1/2}\sigma_t}\sqrt{\log\left(8dt^2/\delta\right)},$$

$$G(z_t) - G_{\text{opt}} \leq C_\delta(1 + V)\sigma_t,$$

*for any $t \geq 1$.*

*Proof of Lemma 7.* Recall from Algorithm 1 that we have defined

$$\begin{cases} S_{t+1} := (t+2)(t+1)\sigma_{t+1} + \sum_{i\in[t+1]}(i+1)i(\sigma_{i-1} - \sigma_i) \\[2mm] z_{t+1} := \frac{1}{S_{t+1}}\left((t+2)(t+1)\sigma_{t+1}x_{t+1} + \sum_{i\in[t+1]}(i+1)i(\sigma_{i-1} - \sigma_i)x_i\right), \end{cases}$$

thus $z_t$ is simply a convex combination of $x_0, \ldots, x_t$ for every $t \geq 1$. Therefore, as $F$ is convex, we can apply Jensen's inequality to the left-hand side of the inequality (10) with $k = 0$, and after some tedious calculation, we arrive at

$$F(z_t) - F_{\text{opt}} \leq \frac{2(\sigma_0 L_f + L_g)D^2 t}{S_t} + \sum_{i\in[t]}\frac{4iD\|\widehat{\nabla\Phi}_{i-1} - \nabla\Phi_{i-1}(x_{i-1})\|}{S_t}.$$

Using the inequality $S_t \geq (t+1)t\sigma_t$, which holds thanks to Condition 1, and applying Proposition 2, with probability at least $1 - \delta$, we have

$$F(z_t) - F_{\text{opt}} \leq \frac{2(\sigma_0 L_f + L_g)D^2}{(t+1)\sigma_t} + \frac{8c\left(4(\sigma_0 L_f + L_g)D + 3(\sigma_0\sigma_f + \sigma_g)\right)}{(t+1)^{1/2}\sigma_t}\sqrt{\log\left(8dt^2/\delta\right)}.$$

This completes the proof of the first claim.

For the second claim, we follow the same procedure. In particular, applying Jensen's inequality with respect to the convex function $G$ to the left-hand side of (15) and using the inequality $S_t \geq (t+1)t\sigma_t$, with probability at least $1 - \delta$, we have

$$G(z_t) - G_{\text{opt}} \leq \frac{C_\delta}{t(t+1)\sigma_t}\left((t+1)t\sigma_t^2 + \sum_{i\in[t]}(i+1)i(\sigma_{i-1} - \sigma_i)\sigma_i\right).$$

The proof concludes by using the definition of $V$. $\qquad\square$

Lemma 7 establishes a convergence result in terms of the regularization parameters $\{\sigma_t\}_{t\geq 0}$. The next lemma specifies an update rule for these parameters and provides bounds on the quantities introduced in (14).

**Lemma 8.** *Consider the sequence $\sigma_t := \varsigma(t+1)^{-p}$ for $t \geq 0$ and the quantities defined in (14).*

*(i) If $p \in (0, 1)$, then $\{\sigma_t\}_{t\geq 0}$ satisfies Condition 1 and Condition 2 with $L = p$. Furthermore,*

$$C \leq (1 + 2p)\left(F_{\text{opt}} - \min_{x\in X} F(x)\right) + \frac{2(\varsigma L_f + L_g)D^2}{\varsigma}, \quad V \leq \frac{2p}{\min\{1, 2(1-p)\}}.$$

*(ii) If $p \in (0, 1/2)$, then $\{\sigma_t\}_{t\geq 0}$ satisfies Condition 3, and we have*

$$\bar{C}_\delta \leq 4c\left(8\left(L_f + \frac{L_g}{\varsigma}\right)D^2 + 3\left(\sigma_f + \frac{\sigma_g}{\varsigma}\right)D\right)\sqrt{\frac{2}{(1-2p)e} + \log\left(8d/\delta\right)}.$$

*Proof of Lemma 8.* As for assertion (i), it is trivial to see that the sequence $\{\sigma_t\}_{t\geq 0}$ satisfies Conditions 1. To validate Condition 2, observe that

$$\lim_{t\to\infty} t\left(\frac{\sigma_t}{\sigma_{t+1}} - 1\right) = \lim_{t\to\infty} t\left(\left(1 + \frac{1}{t+1}\right)^p - 1\right) = \lim_{t\to\infty}\frac{t}{t+1} \cdot \lim_{\delta\to 0}\frac{(1+\delta)^p - 1}{\delta} = p,$$

where the last equality holds as the derivative of $y^p$ at $y = 1$ equals $p$. We next establish the bound for $C$. For $t > 1$, it follows from the mean value theorem that there exists $b_t \in (t, t+1)$ such that

$$t^{-p} - (t+1)^{-p} = -pb_t^{-(p+1)}(t - t - 1) = pb_t^{-(p+1)} \le pt^{-(p+1)}.$$

Hence, we observe that

$$\sum_{i \in [t]}(i+1)i(\sigma_{i-1} - \sigma_i) \le 2\varsigma p \sum_{i \in [t]} i^{1-p} \le 2\varsigma pt^{2-p},$$

where the first inequality is due to $t + 1 \le 2t$ and the second inequality holds because $t^{1-p}$ is an increasing function in $t$. This implies

$$\frac{\sum_{i \in [t]}(i+1)i(\sigma_{i-1} - \sigma_i)}{(t+1)t\sigma_t} \le \frac{2pt^{2-p}}{t(t+1)^{1-p}} \le 2p.$$

Since $\min_{t \ge 0}(t+1)\sigma_t = \varsigma$, we obtain

$$C \le (1+2p)\left(F_{\text{opt}} - \min_{x \in X} F(x)\right) + \frac{2(\varsigma L_f + L_g)D^2}{\varsigma}.$$

We next establish the bound for $V$. Using similar arguments, we observe that

$$\sum_{i \in [t]}(i+1)i(\sigma_{i-1} - \sigma_i)\sigma_i \le 2\varsigma^2 p \sum_{i \in [t]} i^{1-2p}.$$

If $1 - 2p \ge 0$, then $t^{1-2p}$ is an increasing function in $t$ and thus

$$\sum_{i \in [t]}(i+1)i(\sigma_{i-1} - \sigma_i)\sigma_i \le 2p\varsigma^2 t^{2(1-p)}.$$

Dividing both sides by $t(t+1)\sigma_t^2$, we deduce that $V \le 2p$. When $1 - 2p < 0$, we have

$$\sum_{i \in [t]}(i+1)i(\sigma_{i-1} - \sigma_i)\sigma_i \le 2p\varsigma^2\left(1 + \frac{1}{2-2p}\left(t^{2-2p} - 1\right)\right) \le \varsigma^2 \frac{p}{1-p}t^{2(1-p)},$$

where the first inequality is due to the Riemann sum approximation $\sum_{i \in [t]} i^{1-2p} \le 1 + \int_{i=1}^{t} i^{1-2p}\mathrm{d}i$. Dividing both sides by $t(t+1)\sigma_t^2$, we deduce that $V \le p/(1-p)$, thus the claim follows.

As for assertion (ii), it is trivial to see that Condition 3 holds. We thus focus on bounding $\bar{C}_\delta$. Using the fact that $\sup_{t \ge 1} \log(t)/t = 1/e$, for any $t \ge 1$, we may conclude that

$$\frac{8c\left(4(\sigma_0 L_f + L_g)D^2 + 3(\sigma_0 \sigma_f + \sigma_g)D\right)}{(t+1)^{1/2}\sigma_t}\sqrt{\log\left(8dt^2/\delta\right)}$$

$$= 8c\left(4\left(L_f + \frac{L_g}{\varsigma}\right)D^2 + 3\left(\sigma_f + \frac{\sigma_g}{\varsigma}\right)D\right)\sqrt{\frac{1}{(t+1)^{1-2p}}\left(\frac{2}{1-2p}\log\left(t^{1-2p}\right) + \log\left(8d/\delta\right)\right)}$$

$$\le 8c\left(4\left(L_f + \frac{L_g}{\varsigma}\right)D^2 + 3\left(\sigma_f + \frac{\sigma_g}{\varsigma}\right)D\right)\sqrt{\frac{2}{(1-2p)e} + \log\left(8d/\delta\right)}.$$

The claim then follows from the definition of $\bar{C}_\delta$. $\qquad\square$

*Proof of Theorem 1.* The first claim on the non-asymptotic convergence guarantee follows directly from Lemmas 7 and 8. As for the second claim on the asymptotic convergence, we prove a more general result: even if $\sigma_t$ is not set as $\varsigma(t+1)^{-p}$, the iterates of Algorithm 1 still converge almost surely, provided the regularization sequence satisfies the conditions in Lemma 6.

Lemma 6 implies that $\lim_{t \to \infty} G(x_t) = G_{\text{opt}}$ with probability at least $1 - \delta$. As $\delta$ can be arbitrarily small, we may conclude that $\lim_{t \to \infty} G(x_t) = G_{\text{opt}}$, almost surely. This implies that any limit point of $\{x_t\}_{t \ge 0}$ is in $X_{\text{opt}}$ almost surely. Since $F$ is convex, hence lower semi-continuous over $X$, and by definition of $F_{\text{opt}}$, we have $\liminf_{t \to \infty} F(x_t) \ge F_{\text{opt}}$ almost surely.

Besides, by combining Propositions 1 and 2, we deduce that for any $\delta \in (0, 1)$, with probability at least $1 - \delta$, we have, for all $t \ge 1$,

$$F(x_t) - F_{\text{opt}} \le \frac{4c\left(4(\sigma_0 L_f + L_g)D^2 + 3(\sigma_0 \sigma_f + \sigma_g)D\right)}{(t+1)^{1/2}\sigma_t}\sqrt{\log\left(8dt^2/\delta\right)}$$

$$+ \frac{2(\sigma_0 L_f + L_g)D^2}{(t+1)\sigma_t} + \frac{\sum_{i\in[t]}(i+1)i(\sigma_{i-1}-\sigma_i)(F_{\mathrm{opt}}-F(x_i))}{t(t+1)\sigma_t}$$

$$\leq \frac{4c\left(4(\sigma_0 L_f + L_g)D^2 + 3(\sigma_0\sigma_f + \sigma_g)D\right)}{(t+1)^{1/2}\sigma_t}\sqrt{\log\left(8dt^2/\delta\right)}$$

$$+ \frac{2(\sigma_0 L_f + L_g)D^2}{(t+1)\sigma_t} + \frac{\sum_{i\in[t]}(i+1)i(\sigma_{i-1}-\sigma_i)\max\{F_{\mathrm{opt}}-F(x_i),0\}}{t(t+1)\sigma_t},$$

where the second inequality holds due to Condition 1. We claim that all terms on the right hand side converge to 0 as $t\to\infty$. This is obvious except for the last term.

Recall that almost surely $\liminf_{t\to\infty} F(x_t) \geq F_{\mathrm{opt}}$. Therefore we have $\limsup_{t\to\infty}(F_{\mathrm{opt}} - F(x_t)) \leq 0$ and hence $\lim_{t\to\infty}\max\{F_{\mathrm{opt}} - F(x_t),0\} = 0$ almost surely. Applying the Stolz–Cesàro theorem and (33) leads to

$$\lim_{t\to\infty}\frac{\sum_{i\in[t]}(i+1)i(\sigma_{i-1}-\sigma_i)\max\{F_{\mathrm{opt}}-F(x_i),0\}}{t(t+1)\sigma_t} = 0,$$

almost surely. Then each term on the right hand side of the inequality converges to 0 as $t\to\infty$. It follows that with probability at least $1-\delta$,

$$\limsup_{t\to\infty}(F(x_t) - F_{\mathrm{opt}}) \leq 0.$$

Thus, since this holds for any $\delta \in (0,1)$, it holds almost surely. In conclusion, we have shown that $\limsup_{t\to\infty}(F(x_t) - F_{\mathrm{opt}}) \leq 0$ and $\liminf_{t\to\infty}(F(x_t) - F_{\mathrm{opt}}) \geq 0$, which implies that $\lim_{t\to\infty} F(x_t) = F_{\mathrm{opt}}$ almost surely. This completes the proof. $\square$

### B.2 Proof of Theorem 2

We restate Theorem 2 with explicit upper bounds on the stationary gaps and for a more general choice of $p$ and $\omega$. Toward the end of the proof, we show that the optimal parameters are $p = 2/7$ and $\omega = 6/7$.

**Theorem 2.** *Let $\{x_t\}_{t\geq 0}$ denote the iterates generated by Algorithm 1 with stepsizes $\alpha_t = (t+1)^{-\omega}$ and regularization parameters $\sigma_t = \varsigma(t+1)^{-p}$ for any $t \geq 0$. If Assumption 1 holds with the exception that $F$ may be nonconvex, then with probability at least $1-\delta$, for every $t \geq 0$, it (jointly) holds that*

$$\frac{\sum_{i=0}^{t}(i+1)^{-p}\mathcal{F}(x_i)}{\sum_{i=0}^{t}(i+1)^{-p}}$$

$$\leq 4(1-p)\sup_{z\in X}|F(z)|(t+1)^{\omega-1} + 2(1-p)F(x_0)(t+1)^{p-1} + \frac{2(1-p)(\varsigma L_f + L_g)D^2}{\varsigma(1-\omega)}(t+1)^{p-\omega}$$

$$+ \frac{4c(1-p)\sqrt{2}(2(\varsigma L_f + L_g)D + \frac{3^{\omega}}{3^{\omega}-1}(\varsigma\sigma_f+\sigma_g))D}{\varsigma(1-\omega/2)}(t+1)^{p-\omega/2}\sqrt{\log\left(4d(t+2)^2/\delta\right)},$$

*and*

$$\mathcal{G}(x_t) \leq \varsigma\sup_{z\in X}|F(z)|\left(2t^{-p} + \frac{t^{1-p-w}}{1-p}\right) + \varsigma F(x_0)t^{-\omega} + \frac{(\varsigma L_f + L_g)D^2}{1-\omega}t^{1-2\omega}$$

$$+ \frac{2c\sqrt{2}(2(\varsigma L_f + L_g)D + \frac{3^{\omega}}{3^{\omega}-1}(\varsigma\sigma_f+\sigma_g))D}{1-\omega/2}t^{1-3\omega/2}\sqrt{\log\left(4d(t+q)^2/\delta\right)},$$

*where $c$ is some absolute constant.*

*Proof of Theorem 2.* Following the proof of [4, Lemma 4.1], for any $i \geq 0$, one can easily show that, with probability at least $1-\eta_i$, we have

$$\|\widehat{\nabla\Phi}_i - \nabla\Phi_i(x_i)\| \leq c\sqrt{2}\left(2(\sigma_i L_f + L_g)D + \frac{3^{\omega}}{3^{\omega}-1}(\sigma_i\sigma_f + \sigma_g)\right)(i+1)^{-\omega/2}\sqrt{\log\left(\frac{4d}{\eta_i}\right)}.$$

Setting $\eta_i = \frac{\delta}{(i+1)(i+2)}$, applying a union bound, and using the fact that $\sigma_i \leq \sigma_0 = \varsigma$ we have that with probability at least $1 - \delta \sum_{i=0}^{t} \frac{1}{(i+1)(i+2)} \geq 1 - \delta$, the following holds:

$$\sum_{i=0}^{t} \left( 2D\|\widehat{\nabla\Phi}_i - \nabla\Phi_i(x_i)\| + \frac{(\sigma_i L_f + L_g)D^2}{2(i+1)^\omega} \right)$$

$$\leq \sum_{i=0}^{t} \left( \frac{2\sqrt{2}c(2(\varsigma L_f + L_g)D + \frac{3^\omega}{3^\omega - 1}(\sigma_i \sigma_f + \sigma_g))D}{(i+1)^{\omega/2}} \sqrt{\log\left(\frac{4d(t+2)^2}{\delta}\right)} + \frac{(\varsigma L_f + L_g)}{2(i+1)^\omega}D^2 \right)$$

$$\leq \frac{2\sqrt{2}c(2(\varsigma L_f + L_g)D + \frac{3^\omega}{3^\omega - 1}(\varsigma\sigma_f + \sigma_g))D}{1 - \omega/2}(t+1)^{1-\omega/2}\sqrt{\log\left(\frac{4d(t+2)^2}{\delta}\right)}$$

$$+ \frac{(\varsigma L_f + L_g)D^2}{1 - \omega}(t+1)^{1-\omega}.$$

Applying Lemma 3 with $k = 0$, with probability at least $1 - \delta$, we thus have

$$\mathcal{G}(x_{t+1}) \leq 2\varsigma \sup_{z \in X}|F(z)|(t+1)^{-p} + \varsigma F(x_0) - \sum_{i=0}^{t}\sigma_i \mathcal{F}(x_i)(t+1)^{-\omega} + \frac{(\varsigma L_f + L_g)D^2}{1-\omega}(t+1)^{1-2\omega}$$

$$+ \frac{2\sqrt{2}c(2(\varsigma L_f + L_g)D + \frac{3^\omega}{3^\omega-1}(\varsigma\sigma_f + \sigma_g))D}{1-\omega/2}(t+1)^{1-3\omega/2}\sqrt{\log\left(\frac{4d(t+2)^2}{\delta}\right)} \tag{16}$$

Note that the maximum in the definition of function $\mathcal{F}$ is with respect to $X_{\text{opt}}$, not $X$. Therefore, $\mathcal{F}(x_i)$ can take both positive and negative values since $x_i \in X$. However, we can derive the following bound

$$-\sum_{i=0}^{t}\sigma_i \mathcal{F}(x_i) \leq \sup_{z \in X}|\mathcal{F}(z)|\sum_{i=0}^{t}\sigma_i \leq \varsigma \sup_{z \in X}|\mathcal{F}(z)|\frac{(t+1)^{1-p}}{1-p},$$

which further implies that

$$\mathcal{G}(x_t) \leq \varsigma \sup_{z \in X}|F(z)|\left(2t^{-p} + \frac{t^{1-p-w}}{1-p}\right) + \varsigma F(x_0)t^{-\omega} + \frac{(\varsigma L_f + L_g)D^2}{1-\omega}t^{1-2\omega}$$

$$+ \frac{2c\sqrt{2}(2(\varsigma L_f + L_g)D + \frac{3^\omega}{3^\omega - 1}(\varsigma\sigma_f + \sigma_g))D}{1-\omega/2}t^{1-3\omega/2}\sqrt{\log\left(\frac{4d(t+q)^2}{\delta}\right)}. \tag{17}$$

We next focus on the upper-level problem. Since $\mathcal{G}(x_{t+1}) \geq 0$, using (16), we have

$$\sum_{i=0}^{t}\sigma_i \mathcal{F}(x_i) \leq 2\varsigma\sup_{z \in X}|F(z)|(t+1)^{\omega-p} + \varsigma F(x_0) + \frac{(\varsigma L_f + L_g)D^2}{1-\omega}(t+1)^{1-\omega}$$

$$+ \frac{2c\sqrt{2}(2(\varsigma L_f + L_g)D + \frac{3^\omega}{3^\omega-1}(\varsigma\sigma_f + \sigma_g))D}{1-\omega/2}(t+1)^{1-\omega/2}\sqrt{\log\left(\frac{4d(t+2)^2}{\delta}\right)}.$$

Dividing both sides by $\sum_{i=0}^{t}\sigma_i$, and furthermore exploiting the bounds

$$\sum_{i=0}^{t}\sigma_i \geq \varsigma\frac{(t+2)^{1-p} - 1}{1-p} \geq \varsigma\frac{(t+1)^{1-p}}{2(1-p)},$$

we arrive at

$$\frac{\sum_{i=0}^{t}\sigma_i \mathcal{F}(x_i)}{\sum_{i=0}^{t}\sigma_i} \leq 4(1-p)\sup_{z \in X}|F(z)|(t+1)^{\omega-1} + 2(1-p)F(x_0)(t+1)^{p-1}$$

$$+ \frac{2(1-p)(\varsigma L_f + L_g)D^2}{\varsigma(1-\omega)}(t+1)^{p-\omega}$$

$$+ \frac{4c(1-p)\sqrt{2}(2(\varsigma L_f + L_g)D + \frac{3^\omega}{3^\omega-1}(\varsigma\sigma_f + \sigma_g))D}{\varsigma(1-\omega/2)}(t+1)^{p-\omega/2}\sqrt{\log\left(\frac{4d(t+2)^2}{\delta}\right)}. \tag{18}$$

Both bounds involving $\mathcal{G}$ and $\mathcal{F}$ hold with probability $\geq 1 - \delta$. Hence, the first claim on the non-asymptotic convergence guarantee follows. Moreover, by optimizing over the parameters $p$ and $\omega$, we aim to ensure that the right-hand sides of both bounds converge to 0. To achieve this, we select $p$ and $\omega$ to minimize the slowest rate with respect to $t$:

$$\min_{p, w : 0 < p \leq \omega} \max\left\{-p, 1 - p - \omega, -\omega, 1 - 2\omega, 1 - 3\omega/2, \omega - 1, p - 1, p - \omega, p - \omega/2\right\} = -\frac{1}{7}$$

which is realized by setting $p = 2/7$, $\omega = 6/7$ as required. Substituting these values yields the bound presented in the main body of the paper.

As for the second claim on asymptotic convergence, we prove a more general result: If

$$\max\left\{-p, 1 - p - \omega, -\omega, 1 - 2\omega, 1 - 3\omega/2, \omega - 1, p - 1, p - \omega, p - \omega/2\right\} < 0,$$

then it holds that

$$\liminf_{t \to \infty} \mathcal{F}(x_t) = \lim_{t \to \infty} \mathcal{G}(x_t) = 0.$$

Under this additional assumption and the fact that $\mathcal{G}(x_t) \geq 0$, we deduce that $\lim_{t \to \infty} \mathcal{G}(x_t) = 0$ from taking the limit on both sides of (17). From the continuity of $G$, we deduce that any limit point of $\{x_t\}_{t \geq 0}$ is in $X_{\mathrm{opt}}$. the continuity of $\mathcal{F}$ from Lemma 4 and the fact that $\mathcal{F}(x) \geq 0$ for any $x \in X_{\mathrm{opt}}$, we deduce that $\liminf_{t \to \infty} \mathcal{F}(x_t) \geq 0$. Recall that

$$\liminf_{t \to \infty} \mathcal{F}(x_t) = \lim_{t \to \infty} \inf_{k \geq t} \mathcal{F}(x_k),$$

and

$$\frac{\sum_{i=0}^{t} \sigma_i \mathcal{F}(x_i)}{\sum_{i=0}^{t} \sigma_i} \geq \frac{\sum_{i=0}^{t} \sigma_i \inf_{k \geq i} \mathcal{F}(x_k)}{\sum_{i=0}^{t} \sigma_i}.$$

Combining these observations and (18), we have

$$\frac{\sum_{i=0}^{t} \sigma_i \inf_{k \geq i} \mathcal{F}(x_k)}{\sum_{i=0}^{t} \sigma_i}$$
$$\leq 4(1 - p) \sup_{z \in X} |F(z)| (t + 1)^{\omega - 1} + 2(1 - p) F(x_0) (t + 1)^{p - 1}$$
$$+ \frac{2(1 - p)(\varsigma L_f + L_g) D^2}{\varsigma (1 - \omega)} (t + 1)^{p - \omega}$$
$$+ \frac{4c(1-p)\sqrt{2}(2(\varsigma L_f + L_g)D + \frac{3^\omega}{3^\omega - 1}(\varsigma \sigma_f + \sigma_g))D}{\varsigma(1 - \omega/2)} (t + 1)^{p - \omega/2} \sqrt{\log\left(\frac{4d(t + 2)^2}{\delta}\right)}.$$

By the Stolz–Cesàro theorem (since $p \in (0, 1)$, $\sum_{t \geq 0} \sigma_t = \infty$) and taking the limit on both sides of the above inequality, we obtain $\liminf_{t \to \infty} \mathcal{F}(x_t) \leq 0$. Since this holds with probability at least $1 - \delta$ for any $\delta \in (0, 1)$, it should hold almost surely. Thus, we conclude the proof. $\qquad\square$

## C  Proofs of Section 3

### C.1  Proof of Theorem 3

We restate Theorem 3 with exact upper bounds on sub-optimality gaps.

**Theorem 3.** *Suppose the expectations defining $F$ and $G$ in (1) are from uniform distributions over finite sets of size $[n]$. Let $\{z_t\}_{t \geq 1}$ be the iterates generated by Algorithm 2 with $\alpha_t = \log(q)/q$ for $0 \leq t < q$, $\alpha_t = 2/(t + 2)$ for any $t \geq q$, and $\sigma_t := \varsigma(\max\{t, q\} + 1)^{-p}$ for some chosen $\varsigma > 0$, $p \in (0, 1)$ and $S = q$. Under Assumption 1, with probability at least $1 - \delta$, for any $t \geq q$, it holds that*

$$F(z_t) - F_{opt}$$
$$\leq \frac{\max\{F(x_0) - F_{opt}, 0\} + (G(x_0) - G_{opt})/\varsigma + (L_f + L_g/\varsigma)D^2\left(\frac{1}{2} + 16\sqrt{\log\left(\frac{16t^2}{\delta}\right)}\right)\log(q)}{(q + 1)^{-1}(t + 1)^{1 - p}t}$$

$$+ \frac{2(L_f + L_g/\varsigma)D^2(t-q) + \left(8\sqrt{2}t - 4q\right)(L_f + L_g/\varsigma)D^2\sqrt{\log\left(\frac{16t^2}{\delta}\right)}}{(t+1)^{1-p}t},$$

$$G(z_t) - G_{opt}$$
$$\leq \left( (1 + 2p)\left( F_{opt} - \min_{x \in X} F(x) \right)\varsigma + 2(\varsigma L_f + L_g)D^2 + \left( \max_{x \in X} F(x) - f(x_0) \right)\varsigma \right.$$
$$+ \max_{x \in X} g(x) - g(x_0) + \frac{2}{1-p}(L_f\varsigma + L_g)D^2\left( \frac{1}{2} + 16\sqrt{\frac{2}{e(1-p)} + \log(16/\delta)} \right)$$
$$\left. +8\sqrt{2}(L_f\varsigma + L_g)D^2\sqrt{\frac{1}{e(1-p)} + \log(16/\delta)} \right)\left( 1 + \frac{2p}{\min\{1, 2(1-p)\}} \right)(t+1)^{-p}.$$

*Moreover, with probability* 1*, it holds that*

$$\lim_{t \to \infty} F(x_t) = F_{\text{opt}}, \quad \lim_{t \to \infty} G(x_t) = G_{\text{opt}}.$$

The proof proceeds by applying Proposition 1 with $k = q$. Unlike the proof of Theorem 1, here we have two main random components in (9) and (10): $(q + 1)q\left(\Phi_q(x_q) - \Phi_q(x_{\text{opt}})\right)$ and $\sum_{i \in [t] \setminus [q]} 4iD\|\widehat{\nabla\Phi}_{i-1} - \nabla\Phi_{i-1}(x_{i-1})\|$. We first derive probabilistic upper bounds for these terms, then carefully set the parameters in Algorithm 2 to establish the convergence rate for both lower- and upper-level problems. In the following we use the notation

$$F(x) = \frac{1}{n}\sum_{i \in [n]} f_i(x), \quad G(x) = \frac{1}{n}\sum_{i \in [n]} g_i(x).$$

For simplicity, we assume that $F$ and $G$ have the same size support, though our arguments are easily extended to the more general case. Also, given $\mathcal{S} \subseteq [n]$, we write

$$f_{\mathcal{S}}(x) := \frac{1}{|\mathcal{S}|}\sum_{i \in \mathcal{S}} f_i(x), \quad g_{\mathcal{S}}(x) := \frac{1}{|\mathcal{S}|}\sum_{i \in \mathcal{S}} f_i(x).$$

We begin with a probabilistic bound on the gradient estimator used in Algorithm 2.

**Lemma 9.** *Let* $\{x_t\}_{t \geq 0}$ *denote the iterates generated by Algorithm 2 with* $\alpha_t = \log(q)/q$ *for any* $0 \leq t < q$ *and* $\alpha_t = 2/(t+2)$ *for any* $t \geq q$. *If Assumption 1 holds, then for any* $t \geq 0$, *given* $\delta \in (0, 1)$, *with probability at least* $1 - \delta$, *it (jointly) holds that*

$$\left\|\widehat{\nabla F}_t - \nabla F(x_t)\right\| \leq 8L_f D\frac{\log(q)}{\sqrt{qS}}\sqrt{\log(8/\delta)},$$

$$\left\|\widehat{\nabla G}_t - \nabla G(x_t)\right\| \leq 8L_g D\frac{\log(q)}{\sqrt{qS}}\sqrt{\log(8/\delta)},$$

(19)

*provided that* $0 \leq t < q$, *and*

$$\left\|\widehat{\nabla F}_t - \nabla F(x_t)\right\| \leq 8L_f D\sqrt{\frac{(t - s_t)}{S(t+1)(s_t + 1)}}\sqrt{\log(8/\delta)}$$

$$\left\|\widehat{\nabla G}_t - \nabla G(x_t)\right\| \leq 8L_g D\sqrt{\frac{(t - s_t)}{S(t+1)(s_t + 1)}}\sqrt{\log(8/\delta)},$$

(20)

*where* $s_t := q\lfloor t/q \rfloor$, *provided that* $t > q$.

*Proof of Lemma 9.* If $t = s_t$, we have $\widehat{\nabla F}_t = \nabla F(x_t)$. Otherwise, let $\mathcal{S}_t \subset [n]$ be the index set of size $S$ chosen at iteration $t$, and for $i \in \mathcal{S}_t$ define

$$\epsilon_{t,i} := \frac{1}{S}\left( \nabla f_i(x_t) - \nabla f_i(x_{t-1}) - \nabla F(x_t) + \nabla F(x_{t-1}) \right).$$

From the update rule for $x_t$, we have $\|x_t - x_{t-1}\| = \alpha_{t-1}\|v_{t-1} - x_{t-1}\| \le \alpha_{t-1}D$ for any $t \ge 1$. As a result,

$$\|\epsilon_{t,i}\| \le \frac{1}{S}\left(\|\nabla f_i(x_t) - \nabla f_i(x_{t-1})\| + \|\nabla F(x_t) - \nabla F(x_{t-1})\|\right) \le \frac{2L_f D}{S}\alpha_{t-1},$$

for any $t \in \{s_t + 1, \ldots, s_t + q\}$ and $i \in [S_t]$. For any $t$ such that $t \ne s_t$, we have from the definition of $\widehat{\nabla F}_t$ that

$$\widehat{\nabla F}_t - \nabla F(x_t) = \widehat{\nabla F}_{t-1} - \nabla F(x_{t-1}) + \sum_{i \in [S]} \epsilon_{t,i}.$$

Thus, we deduce that

$$\left\|\widehat{\nabla F}_t - \nabla F(x_t)\right\| = \left\|\sum_{s=s_t+1}^{t}\sum_{i \in \mathcal{S}_s} \epsilon_{s,i}\right\|.$$

Since $\sum_{s=s_t+1}^{t}\sum_{i \in \mathcal{S}_s} \epsilon_{s,i}$ is a martingale with bounded increments, we apply a concentration inequality [41, Theorem 3.5] to get

$$P\left(\left\|\widehat{\nabla F}_t - \nabla F(x_t)\right\| \ge \lambda\right) \le 4\exp\left(-\frac{\lambda^2}{4\sum_{s=s_t+1}^{t}\frac{4L_f^2 D^2}{S}\alpha_{s-1}^2}\right).$$

If $1 \le t \le q$, we have

$$\sum_{s=s_t+1}^{t}\frac{4L_f^2 D^2}{S}\alpha_{s-1}^2 \le \frac{4L_f^2 D^2 q}{S}\left(\frac{\log(q)}{q}\right)^2 = 4L_f^2 D^2 \frac{\log^2(q)}{qS},$$

and thus

$$P\left(\left\|\widehat{\nabla F}_t - \nabla F(x_t)\right\| \ge \lambda\right) \le 4\exp\left(-\frac{\lambda^2 qS}{16L_f^2 D^2 \log^2(q)}\right).$$

If $t > q$, we observe that

$$\sum_{s=s_t+1}^{t}\alpha_{s-1}^2 = 4\sum_{s=s_t+1}^{t}\frac{1}{(s+1)^2} \le 4\int_{s_t}^{t}\frac{du}{(u+1)^2} = \frac{4(t-s_t)}{(t+1)(s_t+1)}.$$

Then, we have

$$P\left(\left\|\widehat{\nabla F}_t - \nabla F(x_t)\right\| \ge \lambda\right) \le 4\exp\left(-\frac{\lambda^2 S(t+1)(s_t+1)}{64L_f^2 D^2(t-s_t)}\right).$$

Arguing similarly for $\widehat{\nabla G}_t$, we have

$$P\left(\left\|\widehat{\nabla G}_t - \nabla G(x_t)\right\| \ge \lambda\right) \le 4\exp\left(-\frac{\lambda^2 qS}{16L_g^2 D^2 \log^2(q)}\right),$$

if $1 \le t \le q$ and

$$P\left(\left\|\widehat{\nabla G}_t - \nabla G(x_t)\right\| \ge \lambda\right) \le 4\exp\left(-\frac{\lambda^2 S(t+1)(s_t+1)}{64L_g^2 D^2(t-s_t)}\right),$$

if $t > q$. Given $\delta \in (0,1)$ and $1 \le t \le q$, setting the right hand side $= \delta/2$ and solving for $\lambda$, we have

$$P\left(\left\|\widehat{\nabla F}_t - \nabla F(x_t)\right\| \ge 8L_f D\frac{\log(q)}{\sqrt{qS}}\sqrt{\log(8/\delta)}\right) \le \frac{\delta}{2}$$

and

$$P\left(\left\|\widehat{\nabla G}_t - \nabla G(x_t)\right\| \ge 8L_g D\frac{\log(q)}{\sqrt{qS}}\sqrt{\log(8/\delta)}\right) \le \frac{\delta}{2}.$$

Applying union bound, we deduce that with probability at least $1 - \delta$, (19) holds. Arguing similarly for $t > q$, we also deduce that (20) holds with probability at least $1 - \delta$. $\qquad\square$

To establish convergence results for Algorithm 2, we replace Condition 3 with the following.

**Condition 4.** $\sigma_t = \sigma_0$ for $0 \leq t \leq q$, $(t+1)\sigma_{t+1} > t\sigma_t$ for any $t \geq 0$, and $\log(t) = o(t\sigma_t)$ as $t \to \infty$.

Note that $\sigma_t$ is fixed during the initialization phase, and $\sigma_t^2$ in Condition 3 is replaced by $\sigma_t^2$ in Condition 4. Unlike the proof of Theorem 1, where we set $k = 0$, the proof of Theorem 3 requires setting $k = q$. We therefore begin by analyzing the sequence $\{x_t\}_{t \geq 0}$ generated by Algorithm 2 over the first $q$ iterations.

**Lemma 10.** *Let $\{x_t\}_{t \geq 0}$ denote the iterates generated by Algorithm 2 with $\alpha_t = \log(q)/q$ for any $0 \leq t < q$ and $S = q$. If Assumption 1 and Condition 4 hold, we have then given $\delta \in (0, 1)$, with probability at least $1 - \delta \sum_{0 \leq i < q} 1/(i+1)(i+2)$, for any $t \geq q$, it holds that*

$$
\Phi_q(x_q) - \Phi_0(x_{\mathrm{opt}}) \leq \left(1 - \frac{\log(q)}{q}\right)^q (\Phi_0(x_0) - \Phi_0(x_{\mathrm{opt}}))
$$

$$
+ (L_f\sigma_0 + L_g)D^2 \left(\frac{1}{2} + 16\sqrt{\log\left(\frac{16t^2}{\delta}\right)}\right) \frac{\log(q)}{q}.
$$

*Proof of Lemma 10.* By Lemma 1, for any $0 \leq i < q$, we have

$$
\Phi_i(x_{i+1}) - \Phi_i(x_{\mathrm{opt}}) \leq (1 - \alpha_i)(\Phi_i(x_i) - \Phi_i(x_{\mathrm{opt}})) + \frac{(L_f\sigma_i + L_g)D^2\alpha_i^2}{2} \tag{21}
$$

$$
+ 2D\alpha_i\|\nabla\Phi_i(x_i) - \widehat{\nabla\Phi}_i\|.
$$

By Condition 4, we have $\alpha_0 = \cdots = \alpha_{q-1}$ and

$$
\sigma_0 = \cdots = \sigma_q \implies \Phi_0 = \cdots = \Phi_q,
$$

where the implication follows from the definition of function $\Phi_t$. Thus, we may conclude that

$$\Phi_q(x_q) - \Phi_q(x_{\mathrm{opt}})$$

$$
\leq (1 - \alpha_0)(\Phi_0(x_{q-1}) - \Phi_0(x_{\mathrm{opt}})) + \frac{(L_f\sigma_0 + L_g)D^2\alpha_0^2}{2} + 2D\alpha_0\|\nabla\Phi_{q-1}(x_{q-1}) - \widehat{\nabla\Phi}_{q-1}\|
$$

$$
\leq (1 - \alpha_0)^q (\Phi_0(x_0) - \Phi_0(x_{\mathrm{opt}})) + \sum_{i=0}^{q-1} \frac{(L_f\sigma_0 + L_g)D^2\alpha_0^2}{2}(1 - \alpha_0)^i
$$

$$
+ \sum_{i=0}^{q-1} 2D\alpha_0\|\nabla\Phi_i(x_i) - \widehat{\nabla\Phi}_i\|(1 - \alpha_0)^{q-1-i},
$$

where both inequalities are direct consequences of Condition 4 and (21). Note that

$$
\sum_{i=0}^{q-1}(1 - \alpha_0)^{q-1-i} = \sum_{i=0}^{q-1}(1 - \alpha_0)^i \leq \sum_{i=0}^{\infty}(1 - \alpha_0)^i = \frac{1}{\alpha_0},
$$

where the last equality follows from convergence of geometric series. We thus arrive at

$$
\Phi_q(x_q) - \Phi_q(x_{\mathrm{opt}}) \leq (1 - \alpha_0)^q (\Phi_0(x_0) - \Phi_0(x_{\mathrm{opt}})) + \sum_{i=0}^{q-1} \frac{(L_f\sigma_0 + L_g)D^2\alpha_0}{2}
$$

$$
+ \sum_{i=0}^{q-1} 2D\|\nabla\Phi_i(x_i) - \widehat{\nabla\Phi}_i\|. \tag{22}
$$

Note that by the definition of $\Phi_i$ and the triangle inequality, we have

$$
\|\nabla\Phi_i(x_i) - \widehat{\nabla\Phi}_i\| \leq \sigma_0\|\widehat{\nabla F}_i - \nabla F(x_i)\| + \|\widehat{\nabla G}_i - \nabla G(x_i)\|.
$$

Hence, applying Lemma 9 along with a union bound argument implies that, for any $\delta \in (0, 1)$, with probability at least $1 - \delta \sum_{0 \leq i < q} 1/(i+1)(i+2)$, we have

$$
\sum_{i=0}^{q-1}\|\nabla\Phi_0(x_i) - \widehat{\nabla\Phi}_i\| \leq \sum_{i=0}^{q-1} 8D(\sigma_0 L_f + L_g)\frac{\log(q)}{\sqrt{qS}}\sqrt{\log\left(\frac{8(i+1)(i+2)}{\delta}\right)}.
$$

Plugging the above probabilistic bound into (22), noting that $t \geq q$ and $S = q$, and using the definition of $\alpha_0$ complete the proof. $\qquad\square$

We next analyze the sequence $\{x_t\}_{t\geq 0}$ generated by Algorithm 2 after the first $q$ iterations.

**Proposition 3.** *Let $\{x_t\}_{t\geq 0}$ denote the iterates generated by Algorithm 2 with $\alpha_t = \log(q)/q$ for any $0 \leq t < q$, $\alpha_t = 2/(t+2)$ for any $t \geq q$ and $S = q$. If Assumption 1 holds and the parameters $\{\sigma_t\}_{t\geq 0}$ are non-increasing and positive, then given $\delta \in (0,1)$, with probability at least $1 - \delta \sum_{i\in[t]\setminus[q]} 1/i(i+1)$, for any $t > q$, it holds that*

$$\sum_{i\in[t]\setminus[q]} i\|\widehat{\nabla\Phi}_{i-1} - \nabla\Phi_{i-1}(x_{i-1})\| \leq \left(8\sqrt{2}t - 4q\right)(\sigma_0 L_f + L_g)D\sqrt{\log\left(\frac{16t^2}{\delta}\right)}. \qquad (23)$$

*Proof of Proposition 3.* For any $i \geq 1$, we have

$$\|\widehat{\nabla\Phi}_{i-1} - \nabla\Phi_{i-1}(x_{i-1})\| \leq (\sigma_{i-1}\|\widehat{\nabla F}_{i-1} - \nabla F(x_{i-1})\| + \|\widehat{\nabla G}_{i-1} - \nabla G(x_{i-1})\|)$$
$$\leq (\sigma_0\|\widehat{\nabla F}_{i-1} - \nabla F(x_{i-1})\| + \|\widehat{\nabla G}_{i-1} - \nabla G(x_{i-1})\|). \qquad (24)$$

Given $\delta \in (0,1)$ and $q < i \leq 2q$, by Lemma 9 and the inequality (24), with probability at least $1 - \delta/i(i+1)$, we have

$$i\|\widehat{\nabla\Phi}_{i-1} - \nabla\Phi_{i-1}(x_{i-1})\| \leq 8(\sigma_0 L_f + L_g)Di\sqrt{\frac{i-1-q}{qi(q+1)}}\sqrt{\log\left(\frac{8i(i+1)}{\delta}\right)}$$
$$\leq 8(\sigma_0 L_f + L_g)\frac{Di}{q}\sqrt{\log\left(\frac{8i(i+1)}{\delta}\right)}.$$

Hence, if $q < t \leq 2q$, with probability at least $1 - \delta\sum_{q<i\leq t} 1/i(i+1)$, we have

$$\sum_{q<i\leq t} i\|\widehat{\nabla\Phi}_{i-1} - \nabla\Phi_{i-1}(x_{i-1})\| \leq \sum_{q<i\leq t} 8(\sigma_0 L_f + L_g)\frac{Di}{q}\sqrt{\log\left(\frac{16t^2}{\delta}\right)}$$
$$\leq (\sigma_0 L_f + L_g)D(8\sqrt{2}t - 4q)\sqrt{\log\left(\frac{16t^2}{\delta}\right)},$$

where the second inequality holds as $\frac{1}{q}\sum_{q<i\leq t} i \leq \sqrt{2}t - q/2$ for all $q < t \leq 2q$. Similarly, given $\delta \in (0,1)$ and $i > 2q$, by Lemma 9 and the inequality (24), with probability at least $1 - \delta/i(i+1)$, we have

$$i\|\widehat{\nabla\Phi}_{i-1} - \nabla\Phi_{i-1}(x_{i-1})\| \leq \frac{8(\sigma_0 L_f + L_g)Di}{\sqrt{i(q\lfloor(i-1)/q\rfloor + 1)}}\sqrt{\log\left(\frac{8i(i+1)}{\delta}\right)}$$
$$\leq 8(\sigma_0 L_f + L_g)D\sqrt{\frac{i}{i-q}}\sqrt{\log\left(\frac{8i(i+1)}{\delta}\right)}$$
$$\leq 8\sqrt{2}(\sigma_0 L_f + L_g)D\sqrt{\log\left(\frac{8i(i+1)}{\delta}\right)},$$

where the second inequality follows since $q\lfloor(i-1)/q\rfloor + 1 \geq i - q$, and the third inequality follows since $i/(i-q) < 2$ when $i > 2q$. Thus, if $t > 2q$, with probability at least $1 - \delta\sum_{2q<i\leq t} 1/i(i+1) = 1 - \delta$, we have

$$\sum_{2q<i\leq t} i\|\widehat{\nabla\Phi}_{i-1} - \nabla\Phi_{i-1}(x_{i-1})\| \leq \sum_{2q<i\leq t} 8\sqrt{2}(\sigma_0 L_f + L_g)D\sqrt{\log\left(\frac{16t^2}{\delta}\right)}$$
$$= 8\sqrt{2}(\sigma_0 L_f + L_g)D(t - 2q)\sqrt{\log\left(\frac{16t^2}{\delta}\right)}.$$

For $t > 2q$, with probability at least $1 - \delta\sum_{q<i\leq t} 1/i(i+1)$, the sum over all $q < i \leq t$ is thus

$$\sum_{q<i\leq t} i\|\widehat{\nabla\Phi}_{i-1} - \nabla\Phi_{i-1}(x_{i-1})\|$$

$$\leq (\sigma_0 L_f + L_g)D(8\sqrt{2}(2q) - 4q)\sqrt{\log\left(\frac{16t^2}{\delta}\right)} + 8\sqrt{2}(\sigma_0 L_f + L_g)D(t - 2q)\sqrt{\log\left(\frac{16t^2}{\delta}\right)}$$

$$= \left(8\sqrt{2}t - (16\sqrt{2} - 16\sqrt{2} + 4)q\right)(\sigma_0 L_f + L_g)D\sqrt{\log\left(\frac{16t^2}{\delta}\right)}$$

$$= \left(8\sqrt{2}t - 4q\right)(\sigma_0 L_f + L_g)D\sqrt{\log\left(\frac{16t^2}{\delta}\right)}.$$

Therefore, regardless of whether $q < t \leq 2q$ or $t > 2q$, (23) holds. $\qquad\square$

The following parameters will appear in our analysis and convergence rates

$$C_q := \sup_{t \geq q} \left\{ \begin{array}{l} \left(F_{\mathrm{opt}} - \min_{x \in X} F(x)\right)\left(1 + \dfrac{\sum_{i \in [t]}(i+1)i(\sigma_{i-1} - \sigma_i)}{(t+1)t\sigma_t}\right) \\ + \dfrac{2(\sigma_0 L_f + L_g)D^2(t-q)}{(t+1)t\sigma_t} \end{array} \right\}, \qquad (25a)$$

$$\bar{C}_{\delta,q} := \sup_{t \geq q} \left\{ \begin{array}{l} \dfrac{(q+1)\left(\max\{\Phi_0(x_0) - \Phi_0(x_{\mathrm{opt}}), 0\} + (L_f\sigma_0 + L_g)D^2\left(\frac{1}{2} + 16\sqrt{\log\left(\frac{16t^2}{\delta}\right)}\right)\log(q)\right)}{(t+1)t\sigma_t} \\ + \dfrac{\left(8\sqrt{2}t - 4q\right)(\sigma_0 L_f + L_g)D^2\sqrt{\log\left(\frac{16t^2}{\delta}\right)}}{(t+1)t\sigma_t} \end{array} \right\}, \qquad (25b)$$

$$C_{\delta,q} := \bar{C}_{\delta,q} + C_q, \qquad (25c)$$

$$V_q := \sup_{t \geq q} \left\{ \frac{\sum_{i \in [t]\setminus[q]}(i+1)i(\sigma_{i-1} - \sigma_i)\sigma_i}{(t+1)t\sigma_t^2} \right\}. \qquad (25d)$$

Lemma 17 in Appendix D.1 guarantees that these quantities remain finite for any parameter choice satisfying Conditions 1–2 and 4. In particular, these conditions hold when $\sigma_t = \varsigma(t+1)^p$ for any $p \in (0, 1)$, which is what is used in Theorem 3. We now analyze the sequence $\{x_t\}_{t \geq 0}$ in Algorithm 2 for the inner-level problem.

**Lemma 11.** *Let $\{x_t\}_{t \geq 0}$ denote the iterates generated by Algorithm 2 with $\alpha_t = \log(q)/q$ for $0 \leq t < q$ $\alpha_t = 2/(t+2)$ for any $t \geq q$, $S = q$, and let $C_{\delta,q}$ be defined as in (25c). If Assumption 1 and Condition 1, 2 and 4 hold, then with probability at least $1 - \delta$, it holds that*

$$G(x_t) - G_{\mathrm{opt}} \leq C_{\delta,q}\sigma_t, \qquad (26)$$

*for any $t \geq q$.*

*Proof of Lemma 11.* Combining (9) from Proposition 1 with the probabilistic bounds in Lemma 10 and Proposition 3, and after some straightforward calculations, with probability at least $1 - \delta$, we have

$$G(x_t) - G_{\mathrm{opt}}$$

$$\leq \frac{(q+1)\left(\max\{\Phi_0(x_0) - \Phi_0(x_{\mathrm{opt}}), 0\} + (L_f\sigma_0 + L_g)D^2\left(\frac{1}{2} + 16\sqrt{\log\left(\frac{16t^2}{\delta}\right)}\right)\log(q)\right)}{(t+1)t}$$

$$+ \frac{2(\sigma_0 L_f + L_g)D^2(t-q) + \left(8\sqrt{2}t - 4q\right)(\sigma_0 L_f + L_g)D^2\sqrt{\log\left(\frac{16t^2}{\delta}\right)}}{(t+1)t}$$

$$+ \left(F_{\mathrm{opt}} - \min_{x \in X}F(x)\right)\left(\sigma_t + \frac{\sum_{i \in [t]}(i+1)i(\sigma_{i-1} - \sigma_i)}{(t+1)t}\right).$$

Here, the right-hand side is at most $C_{\delta,q}$, and therefore, $G(x_t) - G_{\mathrm{opt}} \leq C_{\delta,q}\sigma_t$ for any $t \geq q$. $\qquad\square$

We next analyze the convergence of the sequence $\{z_t\}_{t \geq 0}$ in Algorithm 2 for both outer- and inner-level problems.

**Lemma 12.** *Let $\{x_t\}_{t\geq 0}$ denote the iterates generated by Algorithm 2 with $\alpha_t = \log(q)/q$ for $0 \leq t < q$ $\alpha_t = 2/(t+2)$ for any $t \geq q$, $S = q$, and let $C_{\delta,q}$ and $V_q$ defined as in (25). If Assumption 1 and Condition 1, 2 and 4 hold, then with probability at least $1 - \delta$, it (jointly) holds that*

$$F(z_t) - F_{\mathrm{opt}}$$
$$\leq \frac{(q+1)\left(\max\{\Phi_0(x_0) - \Phi_0(x_{\mathrm{opt}}), 0\} + (L_f\sigma_0 + L_g)D^2\left(\frac{1}{2} + 16\sqrt{\log\left(\frac{16t^2}{\delta}\right)}\right)\log(q)\right)}{(t+1)t\sigma_t}$$
$$+ \frac{2(\sigma_0 L_f + L_g)D^2(t-q) + \left(8\sqrt{2}t - 4q\right)(\sigma_0 L_f + L_g)D^2\sqrt{\log\left(\frac{16t^2}{\delta}\right)}}{(t+1)t\sigma_t},$$
$$G(z_t) - G_{\mathrm{opt}} \leq C_{\delta,q}(1 + V_q)\sigma_t,$$

*for any $t \geq q$.*

*Proof of Lemma 12.* Recall from Algorithm 1 that we have defined

$$\begin{cases} S_{t+1} := (t+2)(t+1)\sigma_{t+1} + \sum_{i\in[t+1]\setminus[q]}(i+1)i(\sigma_{i-1} - \sigma_i) \\ z_{t+1} := \frac{1}{S_{t+1}}\left((t+2)(t+1)\sigma_{t+1}x_{t+1} + \sum_{i\in[t+1]\setminus[q]}(i+1)i(\sigma_{i-1} - \sigma_i)x_i\right), \end{cases}$$

thus $z_t$ is simply a convex combination of $x_0, \ldots, x_t$ for every $t \geq q$. Therefore, as $F$ is convex, we can apply Jensen's inequality to the left-hand side of the inequality (10) with $k = q$, and after some tedious calculation, for any $t \geq q$, we arrive at

$$F(z_t) - F_{\mathrm{opt}} \leq \frac{2(\sigma_0 L_f + L_g)D^2 t}{S_t} + \frac{(q+1)q\left(\Phi_q(x_q) - \Phi_q(x_{\mathrm{opt}})\right)}{S_t}$$
$$+ \sum_{i\in[t]} \frac{4iD\|\widehat{\nabla\Phi}_{i-1} - \nabla\Phi_{i-1}(x_{i-1})\|}{S_t}.$$

Using the inequality $S_t \geq (t+1)t\sigma_t$, which holds thanks to Condition 1, and applying Lemma 10 and Proposition 2, with probability at least $1 - \delta$, we have

$$F(z_t) - F_{\mathrm{opt}}$$
$$\leq \frac{(q+1)\left(\max\{\Phi_0(x_0) - \Phi_0(x_{\mathrm{opt}}), 0\} + (L_f\sigma_0 + L_g)D^2\left(\frac{1}{2} + 16\sqrt{\log\left(\frac{16t^2}{\delta}\right)}\right)\log(q)\right)}{(t+1)t\sigma_t}$$
$$+ \frac{2(\sigma_0 L_f + L_g)D^2(t-q) + \left(8\sqrt{2}t - 4q\right)(\sigma_0 L_f + L_g)D^2\sqrt{\log\left(\frac{16t^2}{\delta}\right)}}{(t+1)t\sigma_t}$$

for every $t \geq q$. This completes the proof of the first claim.

For the second claim, we follow the same procedure. In particular, applying Jensen's inequality with respect to the convex function $G$ to the left-hand side of (26) and using the inequality $S_t \geq (t+1)t\sigma_t$, with probability at least $1 - \delta$, we have

$$G(z_t) - G_{\mathrm{opt}} \leq \frac{C_{\delta,q}}{t(t+1)\sigma_t}\left((t+1)t\sigma_t^2 + \sum_{i\in[t]/[q]}(i+1)i(\sigma_{i-1} - \sigma_i)\sigma_i\right).$$

The proof concludes by using the definition of $V_q$. $\qquad\square$

**Lemma 13.** *Consider the sequence $\sigma_t := \varsigma(t+1)^{-p}$ for $t \geq 0$ and the quantities defined in (25). If $p \in (0,1)$, then the sequence $\{\sigma_t\}_{t\geq 0}$ satisfies Condition 1, 2 and 4 with $L = p$. Furthermore,*

$$C_q \leq (1 + 2p)\left(F_{\mathrm{opt}} - \min_{x\in X} F(x)\right) + \frac{2(\varsigma L_f + L_g)D^2}{\varsigma}, \quad V_q \leq \frac{2p}{\min\{1, 2(1-p)\}}$$

*and*

$$\bar{C}_{\delta,q} \leq \max_{x \in X} F(x) - f(x_0) + \frac{\max_{x \in X} g(x) - g(x_0)}{\varsigma}$$

$$+ \frac{2}{1-p}\left(L_f + \frac{L_g}{\varsigma}\right)D^2\left(\frac{1}{2} + 16\sqrt{\frac{2}{e(1-p)} + \log(16/\delta)}\right)$$

$$+ 8\sqrt{2}\left(L_f + \frac{L_g}{\varsigma}\right)D^2\sqrt{\frac{1}{e(1-p)} + \log(16/\delta)}.$$

*Proof of Lemma 13.* We observe that $C_q, V_q$ can be bounded from above by $C, V$ defined in (14) under the conditions in Lemma 8. Therefore, we focus on $\bar{C}_{\delta,q}$ for the remainder of the proof. Using the basic inequality $\log(x)/x \leq 1/e$ for any $x > 0$, observe that for any $t \geq q$

$$\frac{(q+1)\left(\max\{\Phi_0(x_0) - \Phi_0(x_{\text{opt}}), 0\} + (L_f\sigma_0 + L_g)D^2\left(\frac{1}{2} + 16\sqrt{\log\left(\frac{16t^2}{\delta}\right)}\right)\log(q)\right)}{(t+1)t\sigma_t}$$

$$\leq \frac{\max\{\Phi_0(x_0) - \Phi_0(x_{\text{opt}}), 0\} + (L_f\varsigma + L_g)D^2\left(\frac{1}{2} + 16\sqrt{\log\left(\frac{16t^2}{\delta}\right)}\right)\log(t)}{\varsigma t^{1-p}}.$$

We have

$$\Phi_0(x_0) - \Phi_0(x_{\text{opt}}) = \sigma_0\left(f(x_0) - F_{\text{opt}}\right) + g(x_0) - G_{\text{opt}} \leq \varsigma\left(\max_{x \in X} F(x) - F_{\text{opt}}\right) + \max_{x \in X} g(x) - G_{\text{opt}},$$

which implies

$$\frac{\max\{\Phi_0(x_0) - \Phi_0(x_{\text{opt}}), 0\}}{\varsigma t^{1-p}} \leq \varsigma\left(\max_{x \in X} F(x) - F_{\text{opt}}\right) + \max_{x \in X} g(x) - G_{\text{opt}}.$$

We also have

$$\frac{\log(t)}{t^{(1-p)/2}} = \frac{2}{1-p}\frac{\log(t^{(1-p)/2})}{t^{(1-p)/2}} \leq \frac{2}{e(1-p)},$$

and

$$\frac{1}{t^{1-p}}\log\left(\frac{16t^2}{\delta}\right) = \frac{1}{t^{1-p}}\log(16/\delta) + \frac{2}{1-p}\frac{\log(t^{1-p})}{t^{1-p}} \leq \log(16/\delta) + \frac{2}{e(1-p)}.$$

Therefore,

$$\frac{(L_f\varsigma + L_g)D^2\left(\frac{1}{2} + 16\sqrt{\log\left(\frac{16t^2}{\delta}\right)}\right)\log(t)}{\varsigma t^{1-p}}$$

$$\leq \frac{2}{1-p}\left(L_f + \frac{L_g}{\varsigma}\right)D^2\left(\frac{1}{2} + 16\sqrt{\frac{2}{e(1-p)} + \log(16/\delta)}\right)$$

Similarly, we have

$$\frac{(8\sqrt{2}t - 4q)(\sigma_0 L_f + L_g)D^2\sqrt{\log\left(\frac{16t^2}{\delta}\right)}}{(t+1)t\sigma_t} \leq 8\sqrt{2}\left(L_f + \frac{L_g}{\varsigma}\right)D^2\sqrt{\frac{1}{e(1-p)} + \log(16/\delta)}$$

This concludes the proof. $\qquad\square$

*Proof of Theorem 3.* The first part is an immediate consequence of Lemmas 12 and 13. Thus, we devote this proof for the second part by proving that a slightly more general result that as long as Conditions 1–2 and Condition 4 hold, the asymptotic convergence remains.

Lemma 11 implies that $\lim_{t \to \infty} G(x_t) = G_{\text{opt}}$ with probability at least $1 - \delta$. As $\delta$ can be arbitrarily small, we conclude that $\lim_{t \to \infty} G(x_t) = G_{\text{opt}}$, almost surely. This implies that any limit point of

$\{x_t\}_{t\geq 0}$ is in $X_{\text{opt}}$ almost surely. Since $F$ is convex, hence lower semi-continuous over $X$, and by definition of $F_{\text{opt}}$, we have $\liminf_{t\to\infty} F(x_t) \geq F_{\text{opt}}$ almost surely.

Besides, by combining Proposition 1, Lemma 10 and Proposition 3, with probability at least $1-\delta$, we have

$$F(x_t) - F_{\text{opt}}$$
$$\leq \frac{(q+1)\left(\max\left\{\Phi_0(x_0) - \Phi_0(x_{\text{opt}}), 0\right\} + (L_f\sigma_0 + L_g)D^2\left(\frac{1}{2} + 16\sqrt{\log\left(\frac{16t^2}{\delta}\right)}\right)\log(q)\right)}{(t+1)t\sigma_t}$$
$$+ \frac{2(\sigma_0 L_f + L_g)D^2(t-q) + \left(8\sqrt{2}t - 4q\right)(\sigma_0 L_f + L_g)D^2\sqrt{\log\left(\frac{16t^2}{\delta}\right)}}{(t+1)t\sigma_t}$$
$$+ \frac{\sum_{i\in[t]}(i+1)i(\sigma_{i-1} - \sigma_i)\max\{F_{\text{opt}} - F(x_i), 0\}}{t(t+1)\sigma_t}.$$

Similar to the proof of Theorem 1, all terms on the right-hand side converge to $0$ as $t\to\infty$, implying that $\limsup_{t\to\infty}(F(x_t) - F_{\text{opt}}) \leq 0$ which holds with probability at least $1-\delta$ for any $\delta \in (0,1)$. Thus, $\limsup_{t\to\infty}(F(x_t) - F_{\text{opt}}) \leq 0$ almost surely. Combined with $\liminf_{t\to\infty}(F(x_t) - F_{\text{opt}}) \geq 0$, this yields $\lim_{t\to\infty} F(x_t) = F_{\text{opt}}$ almost surely, concluding the proof. $\square$

## C.2 Proof of Theorem 4

Here, we restate Theorem 4 with exact upper bounds on stationary gaps.

**Theorem 4.** *Let $\{x_t\}_{t\geq 0}$ denote the iterates generated by Algorithm 2 with stepsizes $\alpha_t = \log(q+1)/(q+1)$ for any $0 \leq t \leq q$, $\alpha_t = 1/(t+1)^\omega$ for any $t > q$ and $S = q$, and regularization parameters $\sigma_t = \varsigma(\max\{t, q+1\}+1)^{-p}$ with $p = \frac{1}{2}$ and $w = \frac{3}{4}$. If Assumption 1 holds with the exception that $F$ may be nonconvex and the sequence $\{\beta_t\}_{t\geq 0}$ is defined as in (4), then with probability at least $1-\delta$, for every $t \geq q$, it (jointly) holds that*

$$\frac{\sum_{i=0}^t \beta_i \mathcal{F}(x_i)}{\sum_{i=0}^t \beta_i} \leq \left(\frac{G(x_0) - G_{opt}}{(q+1)^{1-\omega}} + \left(2\varsigma(q+1)^{\omega-p} + \varsigma(q+2)^{-p}\right)\sup_{z\in X}|F(z)|\right)\left(a + bt^{1-p}\right)^{-1}$$
$$+ \frac{(\sigma_0 L_f + L_g)D^2\log(q+1)}{2(q+1)^{1-w}}\left(1 + 16\sqrt{\log\left(\frac{8(q+2)^2}{\delta}\right)}\right)\left(a + bt^{1-p}\right)^{-1}$$
$$+ \frac{8(\sigma_0 L_f + L_g)D(t-q)^{1-w}}{1-w}\sqrt{\log\left(\frac{8(t+2)^2}{\delta}\right)}\left(a + bt^{1-p}\right)^{-1}$$
$$+ 2\varsigma\sup_{z\in X}|F(z)|(t+1)^{\omega-p}\left(a + bt^{1-p}\right)^{-1}$$
$$+ \frac{(\varsigma L_f + L_g)D^2}{1-\omega}(t+1)^{1-\omega}\left(a + bt^{1-p}\right)^{-1},$$

$$\mathcal{G}(x_t) \leq \left(\frac{G(x_0) - G_{opt}}{(q+1)^{1-\omega}} + \left(2\varsigma(q+1)^{\omega-p} + \varsigma(q+2)^{-p}\right)\sup_{z\in X}|F(z)|\right)t^{-w}$$
$$+ \frac{(\varsigma L_f + L_g)D^2\log(q+1)}{2(q+1)^{1-w}}\left(1 + 16\sqrt{\log\left(\frac{8(q+2)^2}{\delta}\right)}\right)t^{-w}$$
$$+ \frac{8(\varsigma L_f + L_g)D^2(t-1-q)^{1-w}t^{-w}}{1-w}\sqrt{\log\left(\frac{8(t+1)^2}{\delta}\right)} + \frac{(\varsigma L_f + L_g)D^2}{1-\omega}t^{1-2\omega}$$
$$+ 2\varsigma\sup_{z\in X}|F(z)|t^{-p} + \left(\varsigma(q+1)^{\omega-p} + \varsigma\frac{(t+1)^{1-p} - q^{1-p}}{1-p}\right)t^{-w}\sup_{z\in X}|\mathcal{F}(z)|,$$

*where $\beta_i$'s are defined in (4) and $a, b$ are constants that satisfy*

$$\varsigma\frac{(q+2)^{w-p}}{3} + \varsigma\frac{t^{1-p} - (q+1)^{1-p}}{1-p} = a + bt^{1-p}.$$

The proof relies on several intermediate results. We begin with the following probabilistic bound.

**Lemma 14.** *Let $\{x_t\}_{t \geq 0}$ denote the iterates generated by Algorithm 2 with $\alpha_t = \alpha$ for any $0 \leq t \leq q$ and $\alpha_t = (t+1)^{-\omega}$ with $\omega > \frac{1}{2}$ for any $t > q$. If Assumption 1 holds with the exception that $F$ may be nonconvex, then for any $t \geq 0$, given $\delta \in (0,1)$, with probability at least $1 - \delta$, it (jointly) holds that*

$$
\left\| \widehat{\nabla F}_t - \nabla F(x_t) \right\| \leq 8 L_f D \sqrt{\frac{\alpha^2 q}{S}} \sqrt{\log(8/\delta)}
$$
$$
\left\| \widehat{\nabla G}_t - \nabla G(x_t) \right\| \leq 8 L_g D \sqrt{\frac{\alpha^2 q}{S}} \sqrt{\log(8/\delta)},
$$

(27)

*provided that $0 \leq t < q$, and*

$$
\left\| \widehat{\nabla F}_t - \nabla F(x_t) \right\| \leq 8 L_f D \sqrt{\frac{(t^{1-2w} - s_t^{1-2w})}{S(1-2w)}} \sqrt{\log(8/\delta)}
$$
$$
\left\| \widehat{\nabla G}_t - \nabla G(x_t) \right\| \leq 8 L_g D \sqrt{\frac{(t^{1-2w} - s_t^{1-2w})}{S(1-2w)}} \sqrt{\log(8/\delta)},
$$

(28)

*where $s_t := q\lfloor t/q \rfloor$, provided that $t > q$.*

*Proof of Lemma 14.* Recall from the proof of Lemma 9 that, employing the concentration inequality in [41, Theorem 3.5] yields

$$
P\left( \left\| \widehat{\nabla F}_t - \nabla F(x_t) \right\| \geq \lambda \right) \leq 4 \exp \left( -\frac{\lambda^2}{4 \sum_{s=s_t+1}^{t} \frac{4 L_f^2 D^2}{S} \alpha_{s-1}^2} \right).
$$

If $1 \leq t \leq q$, one can easily show that

$$
P\left( \left\| \widehat{\nabla F}_t - \nabla F(x_t) \right\| \geq \lambda \right) \leq 4 \exp \left( -\frac{\lambda^2 S}{16 L_f^2 D^2 q \alpha^2} \right).
$$

If $t > q$, we observe that

$$
\sum_{s=s_t+1}^{t} \alpha_{s-1}^2 = 4 \sum_{s=s_t+1}^{t} \frac{1}{s^{2w}} \leq 4 \int_{s_t}^{t} \frac{du}{s^{2w}} = \frac{4}{2w-1}(s_t^{1-2w} - t^{1-2w}).
$$

Thus, we have

$$
P\left( \left\| \widehat{\nabla F}_t - \nabla F(x_t) \right\| \geq \lambda \right) \leq 4 \exp \left( -\frac{\lambda^2 S(1-2w)}{64 L_f^2 D^2 (t^{1-2w} - s_t^{1-2w})} \right).
$$

Arguing similarly for $\widehat{\nabla G}_t$, we have

$$
P\left( \left\| \widehat{\nabla G}_t - \nabla G(x_t) \right\| \geq \lambda \right) \leq 4 \exp \left( -\frac{\lambda^2 S}{16 L_g^2 D^2 q \alpha^2} \right),
$$

if $0 \leq t \leq q$ and

$$
P\left( \left\| \widehat{\nabla G}_t - \nabla G(x_t) \right\| \geq \lambda \right) \leq 4 \exp \left( -\frac{\lambda^2 S(1-2w)}{64 L_g^2 D^2 (t^{1-2w} - s_t^{1-2w})} \right),
$$

if $t > q$. Given $\delta \in (0,1)$ and $1 \leq t \leq q$, setting the right hand side $= \delta/2$ and solving for $\lambda$ yields

$$
P\left( \left\| \widehat{\nabla F}_t - \nabla F(x_t) \right\| \geq 8 L_f D \sqrt{\frac{\alpha^2 q}{S}} \sqrt{\log(8/\delta)} \right) \leq \frac{\delta}{2}
$$

and

$$
P\left( \left\| \widehat{\nabla G}_t - \nabla G(x_t) \right\| \geq 8 L_g D \sqrt{\frac{\alpha^2 q}{S}} \sqrt{\log(8/\delta)} \right) \leq \frac{\delta}{2}.
$$

Applying union bound, we deduce that with probability at least $1 - \delta$, (19) holds. Arguing similarly for $t > q$, we also deduce that (20) holds with probability at least $1 - \delta$. $\qquad \square$

We next analyze the sequence $\{x_t\}_{t\geq 0}$ generated by Algorithm 2 over the first $q$ iterations.

**Lemma 15.** *Let $\{x_t\}_{t\geq 0}$ denote the iterates generated by Algorithm 2 with $\alpha_t = \alpha$ for any $0 \leq t \leq q$ and $S = q$. If Assumption 1 holds with the exception that $F$ may be nonconvex and Condition 4 is satisfied, then given $\delta \in (0,1)$, with probability at least $1 - \delta \sum_{0 \leq i \leq q} 1/(i+1)(i+2)$, for any $0 \leq t \leq q+1$, it holds that*

$$\mathcal{G}(x_t) \leq (1-\alpha)^t \mathcal{G}(x_0) + 2\sigma_0 \sup_{z \in X} |F(z)| - \sigma_0 \sum_{i=0}^{t-1} (1-\alpha)^{t-i} \alpha \mathcal{F}(x_i)$$

$$+ \frac{(\sigma_0 L_f + L_g)D^2\alpha}{2}\left(1 + 16\sqrt{\log\left(\frac{8(t+2)^2}{\delta}\right)}\right).$$

*Proof of Lemma 15.* Applying the bound (12) and after a straightforward re-arrangement, under Condition 4, for any $0 \leq i \leq t$, we obtain

$$\mathcal{G}(x_{i+1}) \leq (1-\alpha)\mathcal{G}(x_i) + \sigma_0(F(x_i) - \alpha\mathcal{F}(x_i) - F(x_{i+1}))$$

$$+ 2D\alpha\|\nabla\Phi_0(x_i) - \widehat{\nabla\Phi}_i\| + \frac{(L_f\sigma_0 + L_g)D^2\alpha^2}{2}.$$

Given $\delta \in (0,1)$, we apply Lemma 14 to deduce that with probability at least $1 - \delta/(i+1)(i+2)$, we have

$$\mathcal{G}(x_{i+1}) \leq (1-\alpha)\mathcal{G}(x_i) + \sigma_0(F(x_i) - \alpha\mathcal{F}(x_i) - F(x_{i+1}))$$

$$+ 8(\sigma_0 L_f + L_g)D^2\alpha^2\sqrt{\log\left(\frac{8(i+1)(i+2)}{\delta}\right)} + \frac{(L_f\sigma_0 + L_g)D^2\alpha^2}{2}.$$

Applying the union bound, for any $0 \leq t \leq q$, with probability at least $1 - \delta \sum_{0 \leq i \leq t} 1/(i+1)(i+2)$, we obtain

$$\mathcal{G}(x_{t+1}) \leq (1-\alpha)^{t+1}\mathcal{G}(x_0) + \sum_{i=0}^{t} \sigma_0(F(x_i) - \alpha\mathcal{F}(x_i) - F(x_{i+1}))(1-\alpha)^{t-i}$$

$$+ 8(\sigma_0 L_f + L_g)D^2\alpha^2 \sum_{i=0}^{t}(1-\alpha)^{t-i}\sqrt{\log\left(\frac{8(i+1)(i+2)}{\delta}\right)} + \frac{(L_f\sigma_0 + L_g)D^2\alpha^2}{2}\sum_{i=0}^{t}(1-\alpha)^{t-i}.$$

Using the geometric series bound $\sum_{i=0}^{t}(1-\alpha)^{t-i} \leq 1/\alpha$ and observing that

$$\sum_{i=0}^{t}(F(x_i) - F(x_{i+1}))(1-\alpha)^{t-i}$$

$$= \sum_{i\in[t]} F(x_i)\left((1-\alpha)^{t-i} - (1-\alpha)^{t+1-i}\right) + F(x_0)(1-\alpha)^t - F(x_{t+1})$$

$$= \sum_{i\in[t]} \alpha F(x_i)(1-\alpha)^{t-i} + F(x_0)(1-\alpha)^t - F(x_{t+1})$$

$$\leq \sup_{z\in X}|F(z)|\left(\sum_{i\in[t]}\alpha(1-\alpha)^{t-i} + (1-\alpha)^t + 1\right) \leq 2\sup_{z\in X}|F(z)|$$

conclude the proof. $\qquad\square$

We finally analyze the sequence $\{x_t\}_{t\geq 0}$ generated by Algorithm 2 after the first $q$ iterations.

**Proposition 4.** *Let $\{x_t\}_{t\geq 0}$ denote the iterates generated by Algorithm 2 with $\alpha_t = \alpha$ for any $0 \leq t < q$, $\alpha_t = (t+1)^{-w}$ with $\omega > \frac{1}{2}$ for any $t \geq q$ and $S = q$. If Assumption 1 holds with the exception that $F$ may be nonconvex and the parameters $\{\sigma_t\}_{t\geq 0}$ are non-increasing and positive, then given $\delta \in (0,1)$, with probability at least $1 - \delta \sum_{i\in[t]\setminus[q]} \frac{1}{(i+1)(i+2)}$, for any $t > q$, it holds that*

$$\sum_{i\in[t]\setminus[q]} \left\|\widehat{\nabla\Phi}_i - \nabla\Phi_i(x_i)\right\| \leq \frac{8(\sigma_0 L_f + L_g)D(t-q)^{1-w}}{1-w}\sqrt{\log\left(\frac{8(t+2)^2}{\delta}\right)}. \qquad (29)$$

*Proof of Proposition 4.* For any $i \geq 0$, we have

$$\|\widehat{\nabla\Phi}_i - \nabla\Phi_i(x_i)\| \leq (\sigma_i\|\widehat{\nabla F}_i - \nabla F(x_i)\| + \|\widehat{\nabla G}_i - \nabla G(x_i)\|)$$

$$\leq (\sigma_0\|\widehat{\nabla F}_i - \nabla F(x_i)\| + \|\widehat{\nabla G}_i - \nabla G(x_i)\|). \qquad (30)$$

Moreover, for any $i > q$, by the mean value theorem there exists $c_i \in [s_i, i]$ such that

$$\frac{(i^{1-2w} - s_i^{1-2w})}{(1-2w)} = c_i^{-2w}(i - s_i) \leq \frac{q}{(i-q)^{2w}}.$$

Given $\delta \in (0,1)$ and $i > q$, by Lemma 14 and the inequality (30), with probability at least $1 - \delta/(i+1)(i+2)$, we have

$$\|\widehat{\nabla\Phi}_i - \nabla\Phi_i(x_i)\| \leq \frac{8(\sigma_0 L_f + L_g)D}{(i-q)^w}\sqrt{\log\left(\frac{8(i+1)(i+2)}{\delta}\right)}.$$

Hence, if $t > q$, with probability at least $1 - \delta\sum_{q<i\leq t} 1/(i+1)(i+2)$, we have

$$\sum_{q<i\leq t}\|\widehat{\nabla\Phi}_i - \nabla\Phi_i(x_i)\| \leq \sum_{q<i\leq t}\frac{8(\sigma_0 L_f + L_g)D}{(i-q)^w}\sqrt{\log\left(\frac{8(i+1)(i+2)}{\delta}\right)}$$

$$\leq \frac{8(\sigma_0 L_f + L_g)D(t-q)^{1-w}}{1-w}\sqrt{\log\left(\frac{8(t+2)^2}{\delta}\right)}$$

where the last inequality follows from the Riemann sum approximation. $\qquad\square$

We now instantiate the regularization parameter and analyze Algorithm 2.

**Lemma 16.** *Let $\{x_t\}_{t\geq 0}$ denote the iterates generated by Algorithm 2 with $\alpha_t = \alpha := \log(q+1)/(q+1)$ for any $0 \leq t \leq q$, $\alpha_t = 1/(t+1)^\omega$ for any $t > q$ and $S = q$. If Assumption 1 holds with the exception that $F$ may be nonconvex and Condition 4 is satisfied, then given $\delta \in (0,1)$, with probability at least $1 - \delta\sum_{0\leq i<t-1} 1/(i+1)(i+2)$, for any $t > q+1$, it holds that*

$$t^\omega\mathcal{G}(x_t) \leq \frac{\mathcal{G}(x_0)}{(q+1)^{1-\omega}} + \left(2\varsigma(q+1)^{\omega-p} + \varsigma(q+1)^{-p}\right)\sup_{z\in X}|F(z)|$$

$$+ \frac{(\sigma_0 L_f + L_g)D^2\log(q+1)}{2(q+1)^{1-w}}\left(1 + 16\sqrt{\log\left(\frac{8(q+2)^2}{\delta}\right)}\right)$$

$$+ \frac{8(\varsigma L_f + L_g)D^2(t-1-q)^{1-w}}{1-w}\sqrt{\log\left(\frac{8(t+1)^2}{\delta}\right)} + \frac{(\varsigma L_f + L_g)D^2}{1-\omega}t^{1-\omega}$$

$$+ 2\varsigma\sup_{z\in X}|F(z)|t^{\omega-p} - \sigma_0(q+1)^\omega\sum_{i=0}^q\alpha\mathcal{F}(x_i)(1-\alpha)^{q-i} - \sum_{i=q+1}^{t-1}\sigma_i\mathcal{F}(x_i).$$

*Proof of Lemma 16.* By Lemma 3, for any $t \geq q+1$, we have

$$(t+1)^\omega\mathcal{G}(x_{t+1}) \leq (q+1)^\omega\mathcal{G}(x_{q+1}) + 2\varsigma\sup_{z\in X}|F(z)|(t+1)^{\omega-p} + \sigma_{q+1}F(x_{q+1})$$

$$- \sum_{i=q+1}^t\sigma_i\mathcal{F}(x_i) + \sum_{i=q+1}^t\left(2\|\widehat{\nabla\Phi}_i - \nabla\Phi_i(x_i)\|D + \frac{(\sigma_i L_f + L_g)D^2}{2(i+1)^\omega}\right).$$

Applying Lemma 15, with probability at least $1 - \delta\sum_{0\leq i\leq q} 1/(i+1)(i+2)$, we have

$$\mathcal{G}(x_{q+1}) \leq \frac{1}{q+1}\mathcal{G}(x_0) + 2\sigma_0\sup_{z\in X}|F(z)| - \sigma_0\sum_{i=0}^q\alpha\mathcal{F}(x_i)(1-\alpha)^{q-i}$$

$$+ \frac{(\sigma_0 L_f + L_g)D^2\log(q+1)}{2(q+1)}\left(1 + 16\sqrt{\log\left(\frac{8(q+2)^2}{\delta}\right)}\right).$$

Combining these two bounds with Proposition 4 then yields

$$(t+1)^\omega \mathcal{G}(x_{t+1}) \leq \frac{1}{(q+1)^{1-\omega}}\mathcal{G}(x_0) + 2\varsigma(q+1)^{\omega-p}\sup_{z\in X}|F(z)|$$

$$+ \frac{(\sigma_0 L_f + L_g)D^2 \log(q+1)}{2(q+1)^{1-w}}\left(1 + 16\sqrt{\log\left(\frac{8(q+2)^2}{\delta}\right)}\right)$$

$$+ \frac{8(\sigma_0 L_f + L_g)D(t-q)^{1-w}}{1-w}\sqrt{\log\left(\frac{8(t+2)^2}{\delta}\right)}$$

$$+ 2\varsigma\sup_{z\in X}|F(z)|(t+1)^{\omega-p} + \frac{(\varsigma L_f + L_g)D^2}{1-\omega}(t+1)^{1-\omega} + \sigma_{q+1}F(x_{q+1})$$

$$- \sigma_0(q+1)^\omega\sum_{i=0}^{q}\alpha\mathcal{F}(x_i)(1-\alpha)^{q-i} - \sum_{i=q+1}^{t}\sigma_i\mathcal{F}(x_i).$$

The proof concludes using the simple inequality $\sigma_{q+1}F(x_{q+1}) \leq \varsigma(q+2)^{-p}\sup_{z\in X}|F(z)|$. $\quad\square$

We are now well-equipped to prove Theorem 4.

*Proof of Theorem 4.* We first focus on the terms involving $\mathcal{F}(x_i)$ in Lemma 16. Note that the function $\mathcal{F}$ is defined with respect to $X_{\text{opt}}$, not $X$. Therefore, $\mathcal{F}(x_i)$ can take both positive and negative values since $x_i \in X$. However, we can derive the following bound

$$- \sigma_0(q+1)^\omega\sum_{i=0}^{q}\alpha\mathcal{F}(x_i)(1-\alpha)^{q-i} - \sum_{i=q+1}^{t-1}\sigma_i\mathcal{F}(x_i)$$

$$\leq \sup_{z\in X}|\mathcal{F}(z)|\left(\sigma_0(q+1)^\omega\sum_{i=0}^{q}\alpha(1-\alpha)^{q-i} + \sum_{i=q+1}^{t-1}\sigma_i\right)$$

$$= \sup_{z\in X}|\mathcal{F}(z)|\left(\sigma_0(q+1)^\omega\left(1-(1-\alpha)^{q+1}\right) + \sum_{i=q+1}^{t-1}\sigma_i\right)$$

$$\leq \sup_{z\in X}|\mathcal{F}(z)|\left(\varsigma(q+2)^{-p}(q+1)^\omega(1-(1-\alpha)^{q+1}) + \varsigma\frac{t^{1-p}-(q+1)^{1-p}}{1-p}\right)$$

$$\leq \varsigma\sup_{z\in X}|\mathcal{F}(z)|\left((q+1)^{\omega-p} + \frac{t^{1-p}-(q+1)^{1-p}}{1-p}\right),$$

where $\alpha = \log(q+1)/(q+1)$, and the second and third inequalities follow from the geometric series bound and the definition of $\sigma_t$. Using this bound together with Lemma 16, we arrive at

$$\mathcal{G}(x_t) \leq \left(\frac{\mathcal{G}(x_0)}{(q+1)^{1-\omega}} + \left(2\varsigma(q+1)^{\omega-p} + \varsigma(q+2)^{-p}\right)\sup_{z\in X}|F(z)|\right)t^{-w}$$

$$+ \frac{(\sigma_0 L_f + L_g)D^2 \log(q+1)}{2(q+1)^{1-w}}\left(1 + 16\sqrt{\log\left(\frac{8(q+2)^2}{\delta}\right)}\right)t^{-w}$$

$$+ \frac{8(\varsigma L_f + L_g)D^2(t-1-q)^{1-w}t^{-w}}{1-w}\sqrt{\log\left(\frac{8(t+1)^2}{\delta}\right)} + \frac{(\varsigma L_f + L_g)D^2}{1-\omega}t^{1-2\omega}$$

$$+ 2\varsigma\sup_{z\in X}|F(z)|t^{-p} + \left(\varsigma(q+1)^{\omega-p} + \varsigma\frac{(t+1)^{1-p}-q^{1-p}}{1-p}\right)t^{-w}\sup_{z\in X}|\mathcal{F}(z)|.$$

$$(31)$$

We next focus on the upper-level problem. Since $\mathcal{G}(x_{t+1}) \geq 0$, using Lemma 16 and the definition of the sequence $\{\beta_t\}_{t\geq 0}$, we obtain

$$\sum_{i=0}^{t}\beta_i\mathcal{F}(x_i) \leq \frac{1}{(q+1)^{1-\omega}}\mathcal{G}(x_0) + \left(2\varsigma(q+1)^{\omega-p}\sup_{z\in X} + \varsigma(q+2)^{-p}\right)\sup_{z\in X}|F(z)|$$

$$+ \frac{(\sigma_0 L_f + L_g)D^2 \log(q+1)}{2(q+1)^{1-w}}\left(1 + 16\sqrt{\log\left(\frac{8(q+2)^2}{\delta}\right)}\right)$$

$$+ \frac{8(\sigma_0 L_f + L_g)D(t-q)^{1-w}}{1-w}\sqrt{\log\left(\frac{8(t+2)^2}{\delta}\right)}$$

$$+ 2\varsigma \sup_{z \in X}|F(z)|(t+1)^{\omega-p} + \frac{(\varsigma L_f + L_g)D^2}{1-\omega}(t+1)^{1-\omega}.$$

Observe next that

$$\sum_{i=0}^{t}\beta_i \geq \varsigma(q+2)^{-p}(q+1)^{\omega}(1-(1-\alpha)^{q+1}) + \varsigma\frac{t^{1-p}-(q+1)^{1-p}}{1-p}$$

$$\geq \varsigma\frac{(q+2)^{w-p}}{3} + \varsigma\frac{t^{1-p}-(q+1)^{1-p}}{1-p} := a + bt^{1-p}.$$

where $a$ and $b$ are constants defined to make the equality hold. Combining these bounds, we arrive at

$$\frac{\sum_{i=0}^{t}\beta_i\mathcal{F}(x_i)}{\sum_{i=0}^{t}\beta_i} \leq \left(\frac{\mathcal{G}(x_0)}{(q+1)^{1-\omega}} + \left(2\varsigma(q+1)^{\omega-p} + \varsigma(q+2)^{-p}\right)\sup_{z\in X}|F(z)|\right)\left(a+bt^{1-p}\right)^{-1}$$

$$+ \frac{(\sigma_0 L_f + L_g)D^2\log(q+1)}{2(q+1)^{1-w}}\left(1 + 16\sqrt{\log\left(\frac{8(q+2)^2}{\delta}\right)}\right)\left(a+bt^{1-p}\right)^{-1}$$

$$+ \frac{8(\sigma_0 L_f + L_g)D(t-q)^{1-w}}{1-w}\sqrt{\log\left(\frac{8(t+2)^2}{\delta}\right)}\left(a+bt^{1-p}\right)^{-1}$$

$$+ 2\varsigma\sup_{z\in X}|F(z)|(t+1)^{\omega-p}\left(a+bt^{1-p}\right)^{-1}$$

$$+ \frac{(\varsigma L_f + L_g)D^2}{1-\omega}(t+1)^{1-\omega}\left(a+bt^{1-p}\right)^{-1}.$$

$$(32)$$

Both bounds involving $\mathcal{G}$ and $\mathcal{F}$ hold with probability at least $1-\delta$. We want the right hand side of both bounds $\to 0$. In order to guarantee this, we choose $p, \omega$ to minimize the slowest rate in terms of $t$:

$$\min_{p,w:0<p\leq\omega}\max\left\{p-1, w-1, p-w, -w, -p, 1-2w, 1-w-p\right\} = -\frac{1}{4},$$

which is realized by setting $p = 1/2, \omega = 3/4$ as required.

As for the second claim on asymptotic convergence, we prove a more general result: If

$$\max\left\{p-1, w-1, p-w, -w, -p, 1-2w, 1-w-p\right\} < 0,$$

then the asymptotic convergence holds. Under this additional assumption and the fact that $\mathcal{G}(x_t) \geq 0$, we deduce that $\lim_{t\to\infty}\mathcal{G}(x_t) = 0$ from taking the limit on both sides of (31). Recall that

$$\liminf_{t\to\infty}\mathcal{F}(x_t) = \lim_{t\to\infty}\inf_{k\geq t}\mathcal{F}(x_k),$$

and

$$\frac{\sum_{i=0}^{t}\beta_i\mathcal{F}(x_i)}{\sum_{i=0}^{t}\beta_i} \geq \frac{\sum_{i=0}^{t}\beta_i\inf_{k\geq i}\mathcal{F}(x_k)}{\sum_{i=0}^{t}\beta_i}.$$

Combining these observations and (32), we have

$$\frac{\sum_{i=0}^{t}\beta_i\inf_{k\geq i}\mathcal{F}(x_k)}{\sum_{i=0}^{t}\beta_i}$$

$$\leq \left(\frac{\mathcal{G}(x_0)}{(q+1)^{1-\omega}} + \left(2\varsigma(q+1)^{\omega-p} + \varsigma(q+2)^{-p}\right)\sup_{z\in X}|F(z)|\right)\left(a+bt^{1-p}\right)^{-1}$$

$$+ \frac{(\sigma_0 L_f + L_g)D^2\log(q+1)}{2(q+1)^{1-w}}\left(1 + 16\sqrt{\log\left(\frac{8(q+2)^2}{\delta}\right)}\right)\left(a+bt^{1-p}\right)^{-1}$$

$$+ \frac{8(\sigma_0 L_f + L_g)D(t-q)^{1-w}}{1-w}\sqrt{\log\left(\frac{8(t+2)^2}{\delta}\right)}\left(a + bt^{1-p}\right)^{-1}$$

$$+ 2\varsigma \sup_{z \in X}|F(z)|(t+1)^{\omega-p}\left(a + bt^{1-p}\right)^{-1} + \frac{(\varsigma L_f + L_g)D^2}{1-\omega}(t+1)^{1-\omega}\left(a + bt^{1-p}\right)^{-1}.$$

By the Stolz–Cesàro theorem (since $p \in (0,1)$, $\sum_{t \geq 0}\beta_t = \infty$) and taking the limit on both sides of the above inequality, we obtain $\liminf_{t \to \infty}\mathcal{F}(x_t) \leq 0$. Since this holds with probability at least $1 - \delta$ for any $\delta \in (0,1)$, it also holds almost surely. Due to the fact that any limit point of $\{x_t\}_t$ is in $X_{\text{opt}}$, the continuity of $\mathcal{F}$ from Lemma 4 and the fact that $\mathcal{F}(x) \geq 0$ for any $x \in X_{\text{opt}}$, we deduce that $\liminf_{t \to \infty}\mathcal{F}(x_t) \geq 0$. Thus, we conclude the proof. $\qquad\square$

# D  Additional Discussions

## D.1  Finiteness of constants

The following lemma establishes that the quantities introduced in (14) are finite.

**Lemma 17.** *If Assumption 1 and Conditions 1–3 hold, then $C, \bar{C}_\delta, C_\delta$ defined in (14) are finite. Furthermore, if $L$ defined in Condition 2 is strictly less than 1, then $V$ defined in (14d) is finite.*

*Proof of Lemma 17.* We first show that under Condition 1 and Condition 3, when the limit in Condition 2 exists, the parameter $L$ satisfies $0 \leq L \leq 1$. While it is trivial to see $L \geq 0$ thanks to Condition 1, $L \leq 1$ needs some justifications. For the sake of contradiction, suppose $L > 1$. Then for some sufficiently large $t$, we have

$$t\left(\frac{\sigma_t}{\sigma_{t+1}} - 1\right) > 1 \iff \frac{\sigma_t}{\sigma_{t+1}} > \frac{t+1}{t} \iff (t+1)\sigma_{t+1} < t\sigma_t.$$

This further implies

$$(t+1)\sigma_{t+1}^2 \leq (t+1)\sigma_{t+1}\sigma_t \leq t\sigma_t^2,$$

which always contradicts Condition 3. Thus, we have $0 \leq L \leq 1$ under our blanket assumptions.

We now show that $C$ is finite. Observe that

$$\lim_{t \to \infty}\frac{(t+1)t(\sigma_{t-1} - \sigma_t)}{(t+1)t\sigma_t - t(t-1)\sigma_{t-1}} = \lim_{t \to \infty}\frac{(1 + \frac{2}{t-1})(t-1)(\frac{\sigma_{t-1}}{\sigma_t} - 1)}{2 - (t-1)(\frac{\sigma_{t-1}}{\sigma_t} - 1)} = \frac{L}{2-L}$$

where the first equality is obtained by diving both the numerator and the denominator by $t\sigma_t$ and the second equality follows from Condition 2. Moreover, we have

$$\lim_{t \to \infty}\frac{(t+1)t(\sigma_{t-1} - \sigma_t)}{(t+1)t\sigma_t - t(t-1)\sigma_{t-1}} = \lim_{t \to \infty}\frac{\sum_{i \in [t]}(i+1)i(\sigma_{i-1} - \sigma_i) - \sum_{i \in [t-1]}(i+1)i(\sigma_{i-1} - \sigma_i)}{(t+1)t\sigma_t - t(t-1)\sigma_{t-1}}$$

$$= \lim_{t \to \infty}\frac{\sum_{i \in [t]}(i+1)i(\sigma_{i-1} - \sigma_i)}{(t+1)t\sigma_t}$$

where the second equality follows from the Stolz–Cesàro theorem. This yields

$$\lim_{t \to \infty}\frac{\sum_{i \in [t]}(i+1)i(\sigma_{i-1} - \sigma_i)}{(t+1)t\sigma_t} = \frac{L}{2-L} \implies \sup_{t \geq 1}\frac{\sum_{i \in [t]}(i+1)i(\sigma_{i-1} - \sigma_i)}{(t+1)t\sigma_t} \in (0, \infty). \quad (33)$$

Using Condition 2, we have $\min_{t \geq 0}(t+1)\sigma_t > 0$. Thus, $C$ is finite. It is also straightforward to verify that $\bar{C}_\delta$ is finite by Condition 3. Consequently, $C_\delta$ is finite as well.

To conclude, we show $V$ is finite. From Condition 3, $\{t\sigma_t\}_{t \geq 0}$ and $\{(t+1)t\sigma_t^2\}_{t \geq 0}$ are increasing and diverge to $\infty$. Note that

$$\lim_{t \to \infty}\frac{\sum_{i \in [t]}(i+1)i(\sigma_{i-1} - \sigma_i)\sigma_i}{(t+1)t\sigma_t^2} = \lim_{t \to \infty}\frac{(t+2)(t+1)(\sigma_t - \sigma_{t+1})\sigma_{t+1}}{(t+2)(t+1)\sigma_{t+1}^2 - (t+1)t\sigma_t^2}$$

$$= \lim_{t \to \infty}\frac{\left(\frac{t+2}{t}\right)t\left(\frac{\sigma_t}{\sigma_{t+1}} - 1\right)}{\left(2 - t\left(\frac{\sigma_t}{\sigma_{t+1}} - 1\right)\right)\left(\frac{\sigma_t}{\sigma_{t+1}} + 1\right) - 2\frac{\sigma_t}{\sigma_{t+1}}} = \frac{L}{2(1-L)}$$

where the first equality is implied by the Stolz-Cesàro theorem, the second equality is obtained from dividing both the denominator and the numerator by $(t+1)\sigma_{t+1}^2$, and the third equality comes from Condition 3 and

$$\lim_{t\to\infty} \frac{\sigma_t}{\sigma_{t+1}} = 1 + \lim_{t\to\infty} \frac{1}{t} \cdot t \left( \frac{\sigma_t}{\sigma_{t+1}} - 1 \right) = 1.$$

Note that $L/2(1-L) \in [0,\infty)$ since $L \in [0,1)$. Thus, $V$ is finite. This completes the proof. $\quad\square$

**Lemma 18.** *If Assumption 1, Condition 1, 2 and 4 hold, then $C_q, \bar{C}_{\delta,q}$ defined in (25b), and (25c) are finite. Furthermore, if $L$ defined in Condition 2 is less than 1, then $V_q$ defined in (25d) is finite.*

*Proof of Lemma 18.* The proof follows a similar argument to that of Lemma 17. Details are omitted for brevity. $\quad\square$

### D.2 Over-parameterized regression: implementation details

The implementation details are as follows. For IR-SCG, presented in Algorithm 1, we set $\sigma_t = \varsigma(t+1)^{-1/4}$, where $\varsigma = 10$, along with $\alpha_t = 2/(t+2)$. For IR-FSCG, presented in Algorithm 2, we set $S = q = \lfloor\sqrt{n}\rfloor$, $\sigma_t = \varsigma(\max\{t,q\}+1)^{-1/2}$, where $\varsigma = 10$, along with $\alpha_t = 2/(t+1)$ for every $t \geq q$ and $\alpha_t = \log(q)/q$ for every $t < q$. Our linear minimization oracle involves optimization over an $\ell_1$-norm ball, which admits an analytical solution; see [26, Section 4.1]. We compare our proposed methods against SBCGI with stepsize $\gamma_t = 0.01/(t+1)$, SBCGF with constant stepsize $\gamma_t = 10^{-5}$ (both using $K_t = 10^{-4}/\sqrt{t+1}, S = q = \lfloor\sqrt{n}\rfloor$), aR-IP-SeG with long ($\gamma_t = 10^{-2}/(t+1)^{3/4}$) and short stepsizes ($\gamma_t = 10^{-7}/(t+1)^{3/4}, \rho_t = 10^3(t+1)^{1/4}, r = 0.5$), and SDBGD with long ($\gamma_t = 10^{-2}$) and short ($\gamma_t = 10^{-6}$) stepsizes. We initialized all algorithms with randomized starting points. For SBCGI and SBCGF, we generated the required initial point $x_0'$ by running the SPIDER-FW algorithm [57, Algorithm 2] on the inner-level problem in (5), using a stepsize of $\gamma_t = 0.1/(t+1)$. The initialization process terminated either after $10^5$ stochastic oracle queries or when the computation time exceeded 100 seconds, with the resulting point serving as $x_0$ for both SBCGI and SBCGF. The implementation of these algorithms rely on certain linear optimization or projection oracles. For SBCGI and SBCGF, a linear optimization oracle over the $\ell_1$-norm ball intersecting with a half-space is required, which we employ CVXPY [11] to solve the problems, similar to the implementation in [4, Appendix F.1]. To compute the projection onto the feasible set required for aR-IP-SeG, we used [13, Algorithm 1]. To approximate the outer- and inner-level optimal values $F_{\text{opt}}, G_{\text{opt}}$, we again employ CVXPY to solve the inner-level problem and the reformulation (2) of the bilevel problem.

### D.3 Dictionary learning: implementation details

In our experiment, we obtain $A, A', \hat{X}$ from the code provided by Cao et al. [4] with $n = n' = 250$ and $p = 40, q = 50$. In addition, we also choose $\delta = 3$. For IR-SCG, presented in Algorithm 1, we set $\sigma_t = \varsigma(t+1)^{-2/7}$ and $\alpha_t = (t+1)^{-6/7}$, where $\varsigma = 0.1$. For IR-FSCG, presented in Algorithm 2, we set $S = q = \lfloor\sqrt{n}\rfloor$, $\sigma_t = \varsigma(\max\{t,q+1\}+1)^{-1/2}$, where $\varsigma = 0.1$, along with $\alpha_t = (t+1)^{-3/4}$ for every $t \geq q+1$ and $\alpha_t = \log(q+1)/(q+1)$ for every $t \leq q$. In both experiments, the linear optimization oracle over the $\ell_2$-norm ball admits an analytical solution; see [26, Section 4.1]. For performance comparison, we implement four algorithms: SBCGI [4, Algorithm 1] with stepsize $\gamma_t = 0.1(t+1)^{-2/3}$, SBCGF [4, Algorithm 2] with constant stepsize $\gamma_t = 10^{-3}$ (both using $K_t = 0.01(t+1)^{-1/3}, S = q = \lfloor\sqrt{n}\rfloor$), aR-IP-SeG [27] with long ($\gamma_t = 10^{-2}/(t+1)^{3/4}$) and short stepsizes ($\gamma_t = 10^{-4}/(t+1)^{3/4}, \rho_t = (t+1)^{1/4}, r = 0.5$), and the stochastic variant of the dynamic barrier gradient descent (SDBGD) [21] with long ($\gamma_t = 10^{-2}$) and short ($\gamma_t = 5 \times 10^{-3}$) stepsizes. We initialized all algorithms with randomized starting points. For SBCGI and SBCGF, we generated the required initial point $x_0'$ by running the SPIDER-FW algorithm [57, Algorithm 2] on the inner-level problem in (5), using a stepsize of $\gamma_t = 0.1/(t+1)$. The initialization process terminated either after $10^5$ stochastic oracle queries, with the resulting point serving as $x_0$ for both SBCGI and SBCGF. The implementation of these algorithms rely on certain linear optimization or projection oracles. For SBCGI and SBCGF, a linear optimization oracle over

the $\ell_2$-norm ball intersecting with a half-space is required, which we employ `CVXPY` [11] to solve the problems, similar to the implementation in [4, Appendix F.1]. To compute the projection onto the feasible set required for `aR-IP-SeG`, we used [13, Algorithm 1] to compute projection on $\ell_1$ norm ball and projection onto $\ell_2$ norm ball admits an analytical solution. To approximate inner-level optimal value $G_{\mathrm{opt}}$, we again employ `CVXPY` to solve the inner-level problem.

