# OpenReview forum: "Conditional Gradient Methods with Standard LMO for Stochastic Simple Bilevel Optimization"
_NeurIPS.cc/2025/Conference — NeurIPS 2025 poster_

### Official Review · Reviewer_ZPtF · 2025-06-23

**Clarity:** 3
**Significance:** 2
**Originality:** 2
**Rating:** 3
**Confidence:** 4

**Summary:**

In this paper, the authors propose projection-free methods for solving stochastic convex bilevel optimization problems. They establish non-asymptotic convergence rates of $\mathcal{O}(t^{-(\frac{1}{2} - p)})$ for the outer objective and $\mathcal{O}(t^{-p})$ for the inner objective in the one-sample stochastic setting, where $p \in (0, \frac{1}{2})$ controls the decay of regularization. In the finite-sum setting with a mini-batch scheme, they obtain convergence rates of $\mathcal{O}(t^{-(1 - p)})$ for the outer objective and $\mathcal{O}(t^{-p})$ for the inner objective, where $p \in (0, 1)$.

**Questions:**

1. The authors claim that the proposed algorithms eliminate the need for optimization over intersections of the base feasible set with halfspaces. However, for common feasible sets such as the $\ell_1$ or $\ell_2$ ball, linear minimization over their intersection with a halfspace admits a closed-form solution and does not introduce additional computational cost. Could the authors provide concrete examples or experimental evidence showing that such operations are indeed computationally expensive?

2. How should the parameter $\varsigma$ be chosen in both theory and practice? Does it depend on other problem-specific constants such as $D$, $L_f$, $L_g$, $\sigma_f$, or $\sigma_g$?

3. In Theorem 1, what is $d$? Is it intended to be $D$?

4. The appendix includes results for the case of a non-convex upper-level objective. Why are these results not presented in the main text?

5. The notation $\sigma_t$ is used for the regularization parameter, while $\sigma_f$ and $\sigma_g$ denote variance terms. This could be confusing. Consider using a different symbol for the regularization parameter to avoid ambiguity.

**Ethical Concerns:**

["NO or VERY MINOR ethics concerns only"]

**Final Justification:**

After the rebuttal period, my main concern remains: the proposed method has worse complexity than (Cao et al., 2023) in the convex setting. As the authors suggest moving the nonconvex setting from the appendix to the main text, I note that its complexity is even worse than (Cao et al., 2023). Hence, I maintain my original score.

**Limitations:**

Yes

**Quality:**

3

**Strengths And Weaknesses:**

Strengths:

1. The theoretical foundations of the paper are solid and largely correct.

2. The presentation is clear and easy to follow.

Weaknesses:

1. The main weakness is that the sample complexities of the proposed algorithms are worse than those in [1] under the same setting. There are also minor issues that could be addressed to improve the accuracy of the paper.

2. The paper focuses on simple bilevel problems, which differ from general bilevel problems by excluding an additional upper-level variable. It would be better to specify this in the title and abstract.

3. It would be preferable to present the exact upper bounds of the function gaps in the main theorems, rather than using big-𝒪 notation.

4. Since some log terms appear in the convergence rates, the rates stated in the abstract should be written using $\tilde{\mathcal{O}}$ instead of $\mathcal{O}$.

Reference:

[1] J. Cao, R. Jiang, N. Abolfazli, E. Yazdandoost Hamedani, and A. Mokhtari. Projection-free methods for stochastic simple bilevel optimization with convex lower-level problem. Advances in Neural Information Processing Systems, pages 6105–6131, 2023.

---

> ### Author Rebuttal · Authors · 2025-07-29
>
> We thank the reviewer for the thoughtful feedback. We address the concerns raised below.
>
> 1.  **Convergence rates.** Cao et al indeed has better iteration complexity. However, this is due to using a two stage algorithm that results in three limitations: (i) a requirement to initialize at a point that is $\epsilon_g/2$-optimal for the inner-level problem; (ii) a more expensive linear optimization oracle over $X$ intersected with a halfspace $H$ at each iteration; and (iii) lack of any-time guarantees. So, while it is true that our iteration complexity is worse, our overall algorithm aims to save on per-iteration costs, and also provide stronger any-time guarantees. Our numerical experiments show that the per-iteration savings are indeed valuable: our algorithm outperforms theirs on key examples, due directly to this more expensive per-iteration cost. Please see responses to other reviewers for more details.
>
> 2.  **"Simple bilevel" terminology.** This is a valid point and we will clarify the terminology accordingly. It seems that in the bilevel optimization literature, "simple bilevel" and "convex bilevel" are sometimes used synonymously; see, e.g., Doron and Shtern (2022), Sabach and Shtern (2017). That said, given that our paper addresses both convex and non-convex outer-level objectives, we agree that "simple bilevel" is more appropriate terminology.
>
> 3.  **Avoiding big-O notation:** The exact upper bounds for convex outer-level objective function with one-sample gradient estimator takes the form
> $$
> \begin{align*}
>     &F(z_t) - F_{\text{opt}}\\\\
>     &\leq \frac{2( L_f+L_g/\varsigma)D^2}{(t+1)^{1-p}} +
>     \frac{4c\left(4( L_f+L_g/\varsigma)D+3(\sigma_f+\sigma_g/\varsigma)\right)}{(t+1)^{1/2-p}}\sqrt{\log\left(\frac{8dt^2}{\delta}\right)},\\\\
>     &G(z_t)-G_{\text{opt}}\\\\
>     &\leq \left((1+2p) \left(F_{\text{opt}}-\min_{x \in X} F(x)\right)\varsigma + 2(\varsigma L_f +L_g)D^2\right.\\\\
>     &\quad \left.+4c\left( 8 \left(L_f \varsigma+L_g\right)D^2+3\left(\sigma_f \varsigma+\sigma_g\right)D\right) \sqrt{\frac{2}{(1-2p)e}+\log\left(\frac{8d}{\delta}\right)}\right)\left(1+\frac{2p}{\min\{1,2(1-p)\}}\right) (t+1)^{-p},
> \end{align*}
> $$
>     We note that this bound is also provided in the proof of Theorem 1 in the appendix. For other cases, particularly the nonconvex setting, the exact bounds are more involved. We will make every effort to include these exact rates in the final version. If space constraints prevent this, we will state the convergence results in big-O form in the main paper, while in the appendix we will restate each theorem explicitly with the exact bounds and then provide the corresponding proofs in full detail. We hope this addresses your concerns.
>
> 5.  **Hiding logarithmic terms.** Thank you for pointing this out; we will check all terms and ensure they are correctly expressed with either $O$ or $\tilde{O}$ notation.
>
> **Answers to Questions.**
>
> 1.  **Computational cost of linear minimization over $X$ intersected with a halfspace.** Our claim is that linear optimization over $X$ is often cheaper than over $X \cap H$, where $H$ is a halfspace. In many cases of interest, we believe that the difference is significant enough that our algorithm can outperform that of Cao et al, despite the worse iteration complexity guarantee.
>
>     This is evidenced by our experiments in Section 3, where $X$ was an $\ell_1$-ball. Linear optimization over $X$ can be computed in $O(d)$ operations. We used a LP solver to compute linear optimization over $X \cap H$, which is what was done in Cao et al. We see that our algorithm outperforms theirs, and the tables reported shows that this is because we are able to perform 2-3 orders of magnitude more iterations in the 4 minute time limit, a direct result of the cheaper per-iteration cost.
>
>     When $X$ is $\ell_1$- or $\ell_2$-norm ball, linear optimization over $X$ can indeed be expressed as a closed form and computed in $O(d)$ time. When we consider $X \cap H$ instead, we are unaware of any results that provide a closed form for this. If we analyze the linear optimization problem over $X \cap H$
>     $$ \min_x \\{ c^\top x : \lVert x \rVert \leq r, a^\top x \leq b \\}, $$
>     one can show that its dual problem takes the form:
>     $$ \max_{\mu} \\{ -r \lVert c + \mu a\rVert_* - \mu b : \mu \geq 0 \\}, $$
>     where $\lVert \cdot\rVert_*$ is the dual norm. Given an optimal dual solution $\mu^\star$, the optimal primal solution satisfies
>     $$ x^\star \in \arg\min_{x} \\{ c^\top x + \mu^\star (a^\top x - b): \lVert x \rVert \leq r \\}. \tag{1} $$
>
>     In the case of $\ell_2$-ball, the dual problem is a nonlinear one-dimensional convex optimization problem, which can be solved efficiently with a bisection algorithm. Each iteration of the bisection algorithm costs $O(d)$, so it is not as fast as the closed form for linear optimization over $X$ alone. Once the optimal dual multiplier $\mu^\star$ is found, the optimal primal solution $x^\star$ can be computed in $O(d)$ time since $\min_{x} \\{ c^\top x + \mu^\star (a^\top x - b): \lVert x \rVert_2 \leq r \\}$ has a unique optimum.
>
>     In the case of $\ell_1$-ball, we have $\lVert \cdot \rVert_* = \lVert \cdot \rVert_\infty$ thus the dual is a piecewise linear one-dimensional convex optimization problem. An exact solution can be obtained by computing the breakpoints, but there are $O(d^2)$ potential points, which is much more expensive than $O(d)$ complexity for optimizing over $X$ alone. However, obtaining the optimal dual variable $\mu^\star$ does not lead to easy recovery of the primal solution $x^\star$ for the $\ell_1$-norm ball. This difficulty arises because the primal recovery problem $\min_{x} \\{ c^\top x + \mu^\star (a^\top x - b): \lVert x \rVert \leq r \\}$ may have *multiple* solutions, and in this case, standard duality provides no mechanism for selecting the correct one. To solve the primal optimization problem, one must use a LP solver, or rely on an iterative averaging scheme (for example, Corollary 12.16 of [*]). The averaging approach, however, introduces two severe drawbacks. First, the averaging scheme itself converges slowly, at best with a rate of $1/N$ after $N$ iterations. Second, this iterative process completely negates the core benefit of the Frank-Wolfe method, as the recovered solution is typically dense and no longer a simple structured vertex.
>
>     Our numerical experiments in Section 3 directly support this analysis. We also include an additional result where $X$ is defined by the Nuclear norm, showing an even greater improvement as we are dealing with semidefinite programs. We refer the reviewer to our response to `Reviewer TL81`.
>
>     Besides, the linear minimization oracle in their algorithm is only guaranteed to be feasible with *high probability*, hence the algorithm can stall at any iteration due to oracle infeasibility. In contrast, our oracle is always feasible.
>     We would be grateful for any suggestions if the reviewer thinks we may have overlooked something.
>
>     [*] Orabona, Francesco. "A modern introduction to online learning." arXiv preprint arXiv:1912.13213 (2019).
>
> 2.  **Choice of $\varsigma$.** In theory, since we have two objective functions so there are many criteria to assess to which choice of $\varsigma$ is optimal. Here, let's say we choose $\varsigma$ such that the maximum of the coefficients of the dominant terms in the upper bounds for both level to be smallest. One can see that in all settings, we need to solve the problem of the forms
>     $$\inf_{\varsigma > 0} \max\\{a+b\varsigma, c+\frac{d}{\varsigma}\\},$$
>     where $a,b,c,d$ are positive constants depending on $p,q, \omega, D, L_f, L_g, \sigma_f,\sigma_g, \sup_{z \in X}|F(z)|, F_{\text{opt}}$, $\min_{x \in X} F(x), \max_{x \in X} F(x),F(x_0), G_{\text{opt}}, \min_{x \in X} G(x), \max_{x \in X} G(x), G(x_0)$. We observe that the equation
>     $$a+b\varsigma= c+\frac{d}{\varsigma} \iff b \varsigma^2 +(a-c)\varsigma -d= 0,$$
>     always has a unique positive solution, which is $\varsigma_{\text{opt}}=\frac{c-a+\sqrt{(a-c)^2+4bd}}{2b}$. We note the objective is a convex function with respect to $\varsigma > 0$ and
>     $$0=(\frac{d}{d+b\varsigma_{\text{opt}}^2})b + (\frac{b\varsigma_{\text{opt}}^2}{d+b\varsigma_{\text{opt}}^2})(-\frac{d}{\varsigma_{\text{opt}}^2}) \in \partial\max\\{a+b(\cdot), c+\frac{d}{(\cdot)}\\} (\varsigma_{\text{opt}}),$$
>     which implies $\varsigma_{\text{opt}}$ is indeed, the optimal choice. In practice, we observe that $\varsigma = 0.01, 0.1, 1, 10$ result in good performance.
>
> 3.  **Definition of $d$.** This is not a typo and $d$ is defined to be the dimension of variable $x$, which is mentioned in the introduction.
>
> 4.  **Moving the non-convex results.** This is a valid comment, we will move the non-convex results to the main text, replacing the proof sketches for the convex setting.
>
> 5.  **$\sigma_t$ notation.** We thank the reviewer for this excellent suggestion. To avoid any ambiguity with the variance terms, we will change the notation for the regularization parameter from $\sigma_t$ to $\gamma_t$ in the revised manuscript.
>
> We really appreciate the reviewer's detailed feedback and have aimed to address all points thoroughly. As suggested by `Reviewer 2wsw`, we will reorganize the main text to present the convex and nonconvex results side-by-side, first in Section 2 for the one-sample case (Theorem 1 and Theorem 5 from the appendix) and then in Section 3 for the finite-sum case (Theorem 2 and Theorem 6 from the appendix). To further emphasize the nonconvex setting, we will also replace the second numerical experiment with the nonconvex example from the appendix. Finally, we will revise the title and abstract to better reflect our focus on *simple* bilevel problems, including those with nonconvexity in the outer-level. We would welcome any further suggestions you may have that would help strengthen the manuscript and improve your final evaluation.

---

> > ### Comment · Reviewer_ZPtF · 2025-08-05
> >
> > I thank the authors for the rebuttal, which addressed some of my concerns. However, my main concern remains: the proposed method has worse complexity than (Cao et al., 2023) in the convex setting. As the authors suggest moving the nonconvex setting from the appendix to the main text, I note that its complexity is even worse than (Cao et al., 2023) in that case as well. Specifically, for the nonconvex setting, this work has a complexity of $\mathcal{O}(1/\epsilon^{7})$ in the stochastic setting and $\mathcal{O}(1/\epsilon^{4})$ in the finite-sum setting, whereas (Cao et al., 2023) achieves $\mathcal{O}(1/\epsilon^{3})$ and $\mathcal{O}(1/\epsilon^{2})$, respectively. Therefore, I maintain my original rating.

---

> ### Author Response · Authors · 2025-08-05
>
> Thank you once again for raising the point about our convergence rates. This was also noted by Reviewers `2wsw`, `TL81`, and `B9Ux`, and we appreciate the opportunity to clarify further (see also our responses to those reviewers). While we fully acknowledge that the convergence rate of Cao et al. [4] is theoretically better than ours, it is important to note that this improvement arises from the use of a **stronger linear oracle** over $X \cap H$, not due to a fundamentally tighter analysis. As we mentioned in our response to `Reviewer TL81`, there is compelling evidence that if one restricts to linear oracles over $X$ alone, our convergence rate is **not improvable**. More broadly, we emphasize that **convergence rate alone does not determine practical algorithmic performance**. Equally important is the **per-iteration computational cost**, which plays a critical role in determining overall runtime and scalability. As discussed in our response to your Question 1, linear optimization over $X$ is often significantly more efficient than over $X \cap H$—and in the worst case, certainly no more costly. This distinction is not merely theoretical; our experiments in Section 3 demonstrate tangible performance gains resulting from this difference. Therefore, we believe that evaluating our method solely on the basis of convergence rate provides an incomplete and potentially misleading assessment of its practical value.
>
> Beyond the convergence rate and per-iteration cost, we believe that a number of other aspects of our algorithm analysis warrant consideration in the comparison to Cao et al [4].
> 1. **No initialization requirement.** Their algorithm requires a specific *warm start*: it must be initialized that is already
> $\epsilon_g/2$-optimal for the lower-level problem. Finding such a point can be non-trivial especially in stochastic setting as it requires algorithms that have high probability guarantees rather than in expectation. Moreover, as observed in our numerical experiments (consistent with both theoretical and numerical results of Cao et al.), the convergence behavior of the inner-level problem degrades, going from $\epsilon_g/2$ to $\epsilon_g$, rather than improving. In contrast, our method is much more straightforward to initialize, requiring only a feasible point.
>
> 2. **Any-time guarantees and asymptotic correctness.** In addition to requiring a warm start with respect to the parameter $\epsilon_g$, the method of Cao et al. also uses this parameter to define the halfspace $H$ at each iteration. Thus, their algorithm is not *any-time*: running it indefinitely does not guarantee convergence to an optimal solution of the bilevel problem, but rather to a minimizer of $F$ over the set $\\{x : G(x) \leq G_{\text{opt}} + \epsilon_g\\}$, which is only an approximate solution. In contrast, our algorithm enjoys any-time behavior with correct asymptotic convergence guarantees (Theorems 3 and 4 in the appendix).
>
> 3. **Easy implementation and feasibility guarantee for linear oracle**. In addition to relying on the parameter $\epsilon_g$ for a careful initialization and defining $H$ at each iteration, the algorithm in Cao et al. also depends on an additional parameter $K_t$ to define $H$. This is defined in terms of noise variance, set diameter and objective function smoothness. Since these aspects are not readily available, more tuning may be required to optimally choose $K_t$. In addition, their linear minimization oracle over $X \cap H$ is only guaranteed to be feasible with high probability, hence the algorithm can stall at any iteration due to oracle infeasibility. In contrast, our algorithm requires only the parameter $\varsigma$, and our oracle is always feasible. While the optimal choice of $\varsigma$ depends on the same underlying parameters, this choice only changes the constant factor rather than the convergence rate. In practice, one can simply try $\varsigma \in \\{0.1,1,10\\}$ to achieve satisfactory empirical performance.
>
> 4. **Preserving structural properties.** A key advantage of projection-free algorithms is their ability to maintain low-complexity updates for structured sets. For instance, with the simplex (for sparsity) or the Nuclear norm ball (for low-rankness), the linear oracle over $X$ simply returns an atomic vertex. This crucial benefit is lost in their method, as the oracle over $X \cap H$ typically returns a dense solution. This negates one of the core strengths of projection-free algorithms.
>
> In summary, we believe that a fair and complete assessment of our algorithm’s practical value should consider not only its convergence rate, but also its per-iteration cost, simplicity, robustness, and ease of implementation, areas in which it compares favorably to Cao et al. [4], and which largely explain the superior practical performance in Section 3.

---

### Official Review · Reviewer_aKVj · 2025-06-29

**Clarity:** 4
**Significance:** 3
**Originality:** 4
**Rating:** 4
**Confidence:** 3

**Summary:**

This work studies stochastic bilevel optimization problem where the inner & the outer objectives are smooth convex functions. Additionally, the inner problem is constrained on a convex compact set. After a thourough literature review, the paper uses a regularized objective $\sigma_t F + G$ where the optimization algorithm focuses mainly on $F$ and gradually shifts toward minimization of G through decreasing $\sigma_t$. The gradient estimations were done by two distinct well-known variance reduction methods: STORM & SPIDER. The convergence rates of the proposed methods were studied * simple numerical results supported the superiority of the proposed methods.

**Questions:**

First of all, thanks for your efforts.

1- Why does the paper consider a 4-minute stopping criteria and not a stopping criteria based on suboptimality or Frank-Wolfe gap?

2- As the lower level problem is constrained, objective suboptimality is not a correct measure of stationarity. In such cases the Franke-Wolfe gap is a suitable measure instead. This applies to both your simulations and theorems 1 and 2. Could you please explain why didn't you use the FW gap?

3- Did you try other step-size strategies for your stochastic conditional gradient method?

Thanks!

**Ethical Concerns:**

["NO or VERY MINOR ethics concerns only"]

**Final Justification:**

After reading al the comments I decided to keep my score for two main reasons:

1- As raised by many reviewers, the theoretical results reported in this work are worse than the state of the art,

2- I was happy with the fact that the authors provided additional detail on the experiments, but as raised by Reviewer ZPtF, the practical performance is quite problem-based. Therefore, I keep my borderline positive rating.

**Limitations:**

Limitations were not discussed or at least I could not find them in the main body.

**Paper Formatting Concerns:**

no formatting issue

**Quality:**

3

**Strengths And Weaknesses:**

**Strengths**

1- Well-written paper

2- Good literature review

3- good contribution

**Weaknesses**

1- Numerics are either simple, or not even bilevel problems originally. My main issue is with section 3.2. This problem is a strongly convex constrained optimization. I guess one should consider other methods in this class of functions if we choose such a simulation.

---

> ### Author Rebuttal · Authors · 2025-07-29
>
> We thank the reviewer for the thoughtful feedback. We address the concerns and questions raised below.
>
> 1.  We thank the reviewer for this point. We agree with the reviewer that the experiment from Section 3.2 is adapted from an $\ell_2$-regularized learning problem. However, we wish to note that $\ell_2$-regularization is one of the early inspirations for bilevel optimization: Tikhonov and Arsenin showed that solutions of $\min_{x \in X} \\{\sigma F(x) + G(x)\\}$ converge to the bilevel solution as $\sigma \to 0$; see also Beck and Sabach (2014) for further discussions on minimum norm problems. Bilevel optimization methods then arose to enable us to converge to the bilevel solution without having to solve $\min_{x \in X} \\{\sigma F(x) + G(x)\\}$ completely at each iteration.
>
>     We agree that ideally, we would compare against methods that exploit the strong convexity and deterministic nature of the outer-level function $\frac{1}{2} \lVert x\rVert_2^2$. To that end, Doron and Shtern (2023) provide a class of *deterministic* algorithms that do so for coercive outer-level functions. However, our paper focuses on the *stochastic* setting. We are not aware of any methods that both handle this stochasticity and exploit the specific structure of the outer-level problem in the same way. This is why our comparison focuses on general-purpose stochastic bilevel methods. While our experiment uses a classic $\ell_2$-regularized problem, this serves as a fundamental benchmark for the class of stochastic algorithms we are developing. We welcome any suggestions for work we may have overlooked.
>
> 2. A through comparison between our proposed method against the one in Cao et al. (2023) is available in our responses to `Reviewer 2wsw`. We also run additional experiment on a bilevel matrix completion problem discussed in the reponse for `Reviewer TL81` to further demonstrate our algorithm’s performance relative to Cao et al.
>
> **Answers to questions**
>
> 1.  **Stopping criterion.** Our choice of stopping criterion based on running time rather than an error threshold is chosen because, in practice in the stochastic setting, one would not be able to compute the Frank-Wolfe or suboptimality gap without a full gradient evaluation. Therefore, using running time as a stopping criterion seems more natural. That said, we evaluate the progress at the time of stopping via the suboptimality gap. Also, using running time allows us to have more control over how much we need to use the limited computational resources at our disposal. That said, we do not expect that our comparisons will change if we use a gap-based stopping criterion.
>
>     *Table 1: Time Elapsed to Reach Specific Inner Sub-optimal Levels on Over-parameterized Regression Problem*
>     | Algorithm | 0.1 | 0.01 | 0.001 | 0.0001 |
>     | :--- | :--- | :--- | :--- | :--- |
>     | **`IR-FSCG`** | 1.00e0 | 1.00e0 | 1.00e0 | 1.00e1 |
>     | **`IR-SCG`** | 1.04e0 | 2.01e0 | 4.90e1 | N/A |
>     | **`SBCGF`** | 3.19e1 | 3.19e1 | 3.19e1 | 3.19e1 |
>     | **`SBCGI`** | 3.08e1 | 3.08e1 | 3.08e1 | 3.08e1 |
>     | **`aR-IP-SeG` (long step)** | 1.00e0 | 1.00e0 | 1.00e0 | 1.50e1 |
>     | **`aR-IP-SeG` (short step)**| 1.00e0 | 1.00e0 | 5.00e1 | N/A |
>     | **`SDBGD` (long step)** | 1.00e0 | 1.00e0 | 1.40e1 | 3.40e1 |
>     | **`SDBGD` (short step)** | 1.00e0 | 1.00e0 | 1.40e1 | 3.30e1 |
>
>
>     *Table 2: Time Elapsed to Reach Specific Outer Absolute Sub-optimality on Over-parameterized Regression Problem*
>     | Algorithm | 0.1 | 0.01 | 0.001 | 0.0001 |
>     | :--- | :--- | :--- | :--- | :--- |
>     | **`IR-FSCG`** | 1.00e0 | 1.00e0 | 1.00e0 | 2.00e0 |
>     | **`IR-SCG`** | 1.03e0 | 4.00e0 | 9.01e0 | 1.12e2 |
>     | **`SBCGF`** | 3.19e1 | 3.19e1 | 8.20e1 | N/A |
>     | **`SBCGI`** | 3.08e1 | 3.08e1 | 3.40e1 | N/A |
>     | **`aR-IP-SeG` (long step)** | 1.00e0 | 1.00e0 | N/A | N/A |
>     | **`aR-IP-SeG` (short step)**| 1.00e0 | 1.00e0 | N/A | N/A |
>     | **`SDBGD` (long step)** | 1.00e0 | 1.00e0 | N/A | N/A |
>     | **`SDBGD` (short step)** | 1.00e0 | 1.00e0 | N/A | N/A |
>
> 2.  **Suboptimality vs. Frank-Wolfe gap.** When the inner-level function is convex, which is the case for our experiments in Section 3, the suboptimality gap is upper-bounded by the Frank-Wolfe gap: $0 \leq G(x) - G_{\text{opt}} \leq \max_{v \in X} \nabla G(x)^\top (x - v)$. Therefore the suboptimality gap gives a more accurate measure of progress. Note also that in practice the optimizer does not have access to $G_{\text{opt}}$, so if one were to use a gap measure as a stopping criterion we could use the Frank-Wolfe gap, but we are using the suboptimality gap as an "ideal" measure to assess the performance of the algorithms. Besides, we are not aware of any existing algorithms, particularly in the stochastic setting, that use the Frank-Wolfe gap as a stationarity measure. To the best of our knowledge, both projection-based and projection-free methods typically use suboptimality to quantify stationarity. That said, we welcome any suggestions if there is relevant work we may have overlooked
>
> 3.  **Other step-size strategies.** We have considered exact line search and closed-loop strategies, which are commonly used in Frank-Wolfe–type algorithms for *deterministic* problems. However, we are not aware of *stochastic* variants of these strategies, and implementing them in practice appears to be nontrivial. For this reason, we adopt the open-loop strategy.
>
> We thank the reviewers for their constructive feedback on the paper's presentation. To improve clarity and flow in the final version, we will implement several key revisions. As suggested by `Reviewer 2wsw`, we will reorganize the main text to present the convex and nonconvex results side-by-side, first in Section 2 for the one-sample case (Theorem 1 and Theorem 5 from the appendix) and then in Section 3 for the finite-sum case (Theorem 2 and Theorem 6 from the appendix). To further emphasize the nonconvex setting, we will also replace the second numerical experiment with the nonconvex example from the appendix. Finally, as suggested by `Reviewer ZPtF`, we will revise the title and abstract to better reflect our focus on *simple* bilevel problems, including those with nonconvexity in the outer-level. We welcome any further questions that may arise as you finalize your review, and we thank you again for your time and guidance.

---

> > ### Comment · Reviewer_aKVj · 2025-08-02
> >
> > I'd like to thank the authors for their time and clarifications.
> >
> >
> > Since they clarified my major concerns regarding simulations and added more experiments in reponse to other reviewers, I might raise my score in the final phase.

---

> > > ### Author Response · Authors · 2025-08-02
> > >
> > > We thank the reviewer for their positive evaluation.

---

### Official Review · Reviewer_B9Ux · 2025-06-30

**Clarity:** 3
**Significance:** 2
**Originality:** 2
**Rating:** 4
**Confidence:** 4

**Summary:**

This paper studies the constrained convex simple bilevel optimization via penalty methods with Frank Wolfe algorithm. The proposed algorithm is projection-free and eliminates the need for optimization over intersections of the base feasible set with halfspaces. Convergence rate is given in both stochastic and finite-sum setting. Numerical results on over-parameterized regression and minimal norm logistic regression validate the effectiveness of the proposed method.

**Questions:**

1. How to implement $v_t \in \underset{v \in X}{\arg \min }\left\{\left(\sigma_t \widehat{\nabla F}_t+\widehat{\nabla G}_t\right)^{\top} v\right\}$ in practice? Is there any closed-form solution or we need iterative algorithm?

**Ethical Concerns:**

["NO or VERY MINOR ethics concerns only"]

**Final Justification:**

This paper proposed a simple but efficient algorithm tailored for stochastic simple bilevel optimization problem. It has both theoretical guarantee and strong empirical performance. The merit of the proposed algorithm is its simple implementation and empirical performance, so my previous concerns are mainly around its theoretical guarantee compared with existing works.

The authors solve my concerns during rebuttal including:
1) technical novelty of proving finite-time convergence in the stochastic setting without using linear optimization oracle of the intersection of a hyperplane.
2) although the convergence rate provided is loose than the existing works (the only weakness after the rebuttal), considering its simple implementation, it is still acceptable.
3) the authors have included the results for nonconvex objectives in Appendix which they will highlight in the main paper in the revision.
4) For the lower bound of $f(z_t)-f^*$, the author provided asymptotic guarantee in Appendix B.1.
5) Implementation details and the complexity of linear optimization oracle are provided.

Based on the above justifications, I find this paper to be technically solid, presenting a simple yet practically efficient algorithm with sufficient theoretical guarantees. Therefore, I have changed my score to positive.

**Limitations:**

Same as weakness.

**Quality:**

2

**Strengths And Weaknesses:**

Strength: 1) The proposed method is easy-to-implement and theoretical sound. 2) The method has strong empirical performance.

Weakness: 1) The novelty of this work is limited, as both penalty methods and the Frank-Wolfe algorithm have been extensively studied in the context of simple and general bilevel optimization. 2) The convergence rate of the proposed method appears to be worse than that of Cao et al. [4, Algorithm 2], even when $p$ is chosen to optimally balance the upper- and lower-level objective value gaps. 3) This paper only considers the convex setting rather than the nonconvex case, which restricts the practical applicability of the algorithm. 4) The theoretical guarantee for the upper-level objective is weak. Specifically, $f^$ is not defined as the global minimum over the entire domain, so the actual objective value $f$ can be smaller than $f^$. As a result, the theory provides only an upper bound on $f - f^*$, without a corresponding lower bound.

---

> ### Author Rebuttal · Authors · 2025-07-29
>
> We thank the reviewer for their thoughtful feedback. We address the concerns raised below.
> *   **Novelty of the work.** We agree that penalty methods are well-studied in the bilevel optimization literature. However, for simple bilevel optimization, to our knowledge only Jiang et al, Giang-Tran et al, Doron and Shtern (2022) and Cao et al. explored Frank-Wolfe-based methods. Amongst these, the stochastic setting was only considered by Cao et al.. Our work improves upon theirs in two ways: we allow for an oracle over only $X$ rather than $X$ intersected with a hyperplane, and we obtain *any-time* convergence guarantees. (That said, the reviewer is correct that our iteration complexity is worse than theirs; we address this in the next point.) We take a completely different approach to their paper by using an iterative regularization (penalty) method, which combined with Frank-Wolfe type updates has not been considered in the stochastic setting. Our algorithm design and analysis are non-trivial, requiring a carefully constructed averaging scheme to obtain convergence rates, and also particular attention to the initialization of the gradient estimates in the finite-sum setting. Therefore, we believe that our paper is novel compared to current bilevel optimization literature. If we have overlooked any related work, we would be grateful for suggestions and would be glad to consider them carefully.
>
> *   **Convergence rates.** The reviewer is correct in stating that our iteration complexity (and hence also sample complexity) is worse than that of Cao et al.. However, we believe that our algorithm is still of value despite this, for four key reasons.
>     1.  *Type of linear oracle.* As mentioned above, our algorithm uses a linear oracle over $X$ only, while theirs requires one for $X \cap H_t$, where $H_t$ is a dynamically changing halfspace. Generally, linear optimization over $X$ only is often much cheaper than linear optimization over the intersection $X \cap H$, where $H$ is a halfspace. This is true even in cases where $X \cap H$ has an efficiently implementable (e.g., polynomial time) linear oracle. An example from the paper is when $X$ is an $\ell_1$-ball. Since $X$ is a polytope, $X \cap H$ is also a polytope, hence linear optimization can be computed using any off-the-shelf LP solver. (This is also how Cao et al. implement this example in their paper.) While LP solvers are generally efficient, linear optimization over $X$ alone can be computed in $O(d)$ time, which is much more efficient than using a solver. (See our response to `Reviewer 2wsw` for shortcomings of another approach besides LP.)
>
>         As the reviewer has noted, the empirical performance of our method is strong compared to others, including Cao et al., and we directly attribute this to the cheaper linear optimization oracle required. Therefore, we have shown that trading off iteration complexity for a cheaper per-iteration cost can bring computational benefits.
>     2.  *Easy initialization.* The algorithm of Cao et al. requires initialization with a point $x_0$ that is already $\epsilon_g/2$-optimal for the lower-level problem. Finding such a point can be non-trivial especially in stochastic setting as it requires algorithms that provide such a point with high probability rather than in expectation. In contrast, our algorithm does not require any complicated initialization.
>     3.  *Any-time guarantees.* The warm-start parameter $\epsilon_g$ of Cao et al. also appears in the definition of the halfspaces $H_t$ at each iteration. In essence, this means that the sublevel set $\\{x : G(x) \leq G_{\text{opt}} + \epsilon_g\\}$ is contained in $H_t$. However, this means that if their algorithm is run indefinitely, there is no assurance that their algorithm converges to an optimal solution of the bilevel problem, but rather it may instead converge to a minimizer of $f$ over the sublevel set $\\{x : G(x) \leq G_{\text{opt}} + \epsilon_g\\}$, which is an *approximate* solution. In contrast, if our algorithm is run indefinitely, we proved in Appendix B.1 that it will converge to an optimal solution of the bilevel problem. In other words, our algorithm is *any-time* whereas theirs is not.
>     4.  *Feasibility of the linear oracle.* The linear minimization oracle in their algorithm is only guaranteed to be feasible with *high probability*, hence the algorithm can stall at any iteration due to oracle infeasibility. In contrast, our algorithm requires only the parameter $\varsigma$, and our oracle is always feasible.
>
> *   **Non-convex objectives.** We have already provided results for the non-convex outer level setting in Appendix B.2. That said, to highlight this further, we intend to reorganize the paper slightly by placing these results in the main text, and moving the proof outlines to the appendix.
>
> *   **Guarantee for outer-level objective.** We agree with the reviewer that since each iteration $x_t$ may not satisfy $G(x_t) \leq G_{\text{opt}}$, they may in fact be *super*optimal and we have $F(x_t) < F_{\text{opt}}$. In fact, in Appendix B.1. we show that $F(x_t) \to F_{\text{opt}}$, which gives reassurance on the behavior of the outer-level objective. The reviewer is correct that we do not provide a lower *bound* on this gap. We are aware of such lower bounds in the deterministic setting under an additional error bound assumption for the inner level objective; see, e.g., Jiang et al. (2023) and Merchav et al. (2024). However, we are unaware of such results in the stochastic bilevel setting, and we think it is indeed a very interesting direction. Given the length of our current manuscript, we decided to defer this investigation for future work.
>
> *   **Answer to Question.** The implementation of the linear optimization oracle for $X$ is highly dependent on the structure of $X$. In cases such as $\ell_1$- or $\ell_2$-norm ball, there is a closed form which can be computed in $O(d)$ time. When $X$ is a Nuclear norm ball, the solution involves computing a leading singular vector, for which algorithms exist. In our paper we considered the $\ell_1$-ball, and we implemented the linear optimization oracle via this closed form expression:
> $$ \arg\min_{v \in X} z^\top v = \text{sign}(z_i) e_i, \quad i \in \arg\max_{j \in [d]} |z_j|. $$
>
> We thank the reviewers for their constructive feedback on the paper's presentation. To improve clarity and flow in the final version, we will implement several key revisions. As suggested by `Reviewer 2wsw`, we will reorganize the main text to present the convex and nonconvex results side-by-side, first in Section 2 for the one-sample case (Theorem 1 and Theorem 5 from the appendix) and then in Section 3 for the finite-sum case (Theorem 2 and Theorem 6 from the appendix). To further emphasize the nonconvex setting, we will also replace the second numerical experiment with the nonconvex example from the appendix. Finally, as suggested by `Reviewer ZPtF`, we will revise the title and abstract to better reflect our focus on *simple* bilevel problems, including those with nonconvexity in the outer-level. Please let us know if there are any remaining points we can address that would be helpful in your final evaluation.

---

> > ### Comment · Reviewer_B9Ux · 2025-08-01
> >
> > I thank the authors for their detailed response. As for the complexity of linear optimization oracle, I recommend adding a remark comparing it with that of the projection operator for popular constraint sets. This would help clarify why Frank-Wolfe is preferable over projected gradient descent in this setting. For example, refer to the results in [R1].
> >
> > Also for the lower bound for $f-f^*$, I recommend to highlight the results in Appendix B.1 in the main paper as they are essential to understand whether the guarantee rules out the case where $f(z_t) < f(z_{\text{opt}})$.
> >
> > Overall, the response solves all of my concerns. I will increase my score to 4.
> >
> > [R1] Complexity of linear minimization and projection on some sets. Cyrille W. Combettes, et. al. 2021.

---

> ### Author Response · Authors · 2025-08-01
>
> We thank the reviewer for their kind response and the revision of their score.

---

### Official Review · Reviewer_TL81 · 2025-07-02

**Clarity:** 4
**Significance:** 3
**Originality:** 2
**Rating:** 4
**Confidence:** 4

**Summary:**

This paper tackles the challenge of solving stochastic convex bilevel optimization problems, where the goal is to minimize a stochastic outer function over the solution set of a stochastic inner function. Such problems are difficult because the inner problem's solution set is not explicitly known, which makes standard optimization operations like projections computationally expensive or intractable.  To address this, the authors propose an efficient framework based on the conditional gradient method (Frank-Wolfe-style algorithm) combined with iterative regularization. This approach sidesteps costly projections by relying exclusively on simple linear optimization oracles over the base feasible set, which is a significant advantage over prior methods.

The authors present two specific algorithms: IR-SCG for the general one-sample stochastic setting (the stochastic objective is averaged over a continuous distribution that only has sample access) and IR-FSCG for the finite-sum setting (the stochastic objective is averaged over finite samples). Both algorithms employ variance reduction techniques (STORM and SPIDER from previous literature) to handle the noise from stochastic gradient estimates. The paper establishes non-asymptotic convergence rates for both algorithms, showing rates of $O(t^{-(1/2−p)})$ for the outer problem in the one-sample case and $O(t^{-(1−p)})$ in the finite-sum case. Experiments on over-parametrized and logistic regression tasks demonstrate that these methods significantly outperform existing approaches in terms of wall-clock time, a practical advantage attributed to the computational efficiency of their simpler linear optimization oracles.

Regarding the major claim of the contributions of the use of linear optimization oracles over the original feasible region $X$ as opposed to the intersection of the feasible region and a hyperplane, this part I don't completely get the computation benefit. When the feasible region $X$ is specified as a polygon, adding a hyperplane to get its intersection doesn't add additional difficulty. When the feasible region $X$ is not a polygon, adding a hyperplane can indeed lead to more difficulty but the original linear optimization in the non-polygon feasible region is also non-trivial if the feasible region is non-trivial. Since this is the major difference compared to prior work by Cao et al. 2023, could you please explain in what cases the linear optimization oracle has a clear advantage over the one by Cao that uses hyperplane? Other than that, I checked the proof in the main paper and Lemma 1, 2. I didn't check the rest of the proof in the appendix but the proof sketch looks correct to me.

**Questions:**

Please refer to the strengths and weaknesses section.

**Ethical Concerns:**

["NO or VERY MINOR ethics concerns only"]

**Final Justification:**

I have read the authors' response and the response helps address my concerns. I agree that the convergence analysis is slightly different from prior work, and further analysis of the lower bound and impossibility result would be an interesting follow up. Therefore, I update my assessment accordingly.

**Limitations:**

Yes

**Quality:**

3

**Strengths And Weaknesses:**

Strengths:
- Clear presentation and clear difference compared to prior work (especially Cao et al.)
- One difference compared to Cao et al. is the use of the weight $\rho$ and the dynamic weight sequence $\rho_t$. This choice of sequence gives an additional dependency of $p \in (0,1/2)$ or $p \in (0,1)$ for two cases that the paper considers. From the theory perspective, this is an interesting observation but not very significantly different from prior analysis though.
- Very detailed summary of the literature! This is very helpful for readers.

Weaknesses:
- Very similar results compared to Cao et al. 2023. Please clarify the advantage of not requiring intersection over the feasible set in linear optimization. Why does linear optimization with one additional hyperplane cause significant computational issues?
- Algorithm designs are also directly followed by Cao et al. I didn't check the proof in Cao et al. but I believe it follows a similar strategy.
- Theoretical results are slightly worse than Cao et al. Could you explain why the use of dynamic weight sequence gives you worse result?

Neutral questions:
- Experimental results look good, but I wonder how many iterations does each algorithm run? How much worse is the linear optimization oracle in terms of the computational cost? How did you implement the one with additional hyperplane and is its computation cost optimized? I think it is hard to judge the performance by the computation cost merely because I can't tell if the oracles are all implemented in the optimal way.

---

> ### Author Rebuttal · Authors · 2025-07-29
>
> Thank you for your thoughtful review. Below, we address the other specific points and questions from your review.
>
> *   **Linear oracles over $X \cap H$ versus $X$.** Our central argument is that the lower per-iteration cost of our algorithm's oracle over $X$ often more than compensates for its worse iteration complexity guarantee compared to the oracle over $X \cap H$. For many important structured sets, this trade-off makes our method superior in terms of execution time. While we acknowledge that for a general polytope $X$, the intersection $X \cap H$ is also a polytope and both oracles would rely on an LP solver, the situation is different for many structured sets of interest. For sets like the $\ell_1$-ball, $\ell_2$-ball, Nuclear norm ball or the unit simplex, the linear oracle over $X$ can be computed exactly and efficiently. In contrast, the oracle over $X \cap H$ is significantly more complex.
>
>     For the sake of illustration, assume that $X$ is a norm ball. Consider the linear minimization oracle problem over $X \cap H$:
>     $$ \min_x \\{ c^\top x : \lVert x \rVert \leq r, a^\top x \leq b \\}. $$
>     One can show that its dual problem is
>     $$ \max_{\mu} \\{ -r \lVert c + \mu a\rVert_* - \mu b : \mu \geq 0 \\}, $$
>     where $\lVert \cdot\rVert_*$ is the dual norm. Given an optimal dual solution $\mu^\star$, the optimal primal solution satisfies
>     $$ x^\star \in \arg\min_{x} \\{ c^\top x + \mu^\star (a^\top x - b): \lVert x \rVert \leq r \\}. \tag{1}$$
>
>     In the case of $\ell_2$-ball, the dual problem is a nonlinear one-dimensional convex optimization problem, which can be solved efficiently with a bisection algorithm. Each iteration of the bisection algorithm costs $O(d)$, so it is not as fast as the closed form for linear optimization over $X$ alone. Once the optimal dual multiplier $\mu^\star$ is found, the optimal primal solution $x^\star$ can be computed in $O(d)$ time since (1) has a unique optimum.
>
>     In the case of $\ell_1$-ball, we have $\lVert \cdot \rVert_* = \lVert \cdot \rVert_\infty$ thus the dual is a piecewise linear one-dimensional convex optimization problem. An exact solution can be obtained by computing the breakpoints, but there are $O(d^2)$ potential points, which is much more expensive than $O(d)$ complexity for optimizing over $X$ alone. However, obtaining the optimal dual variable $\mu^\star$ does not lead to easy recovery of the primal solution $x^\star$ for the $\ell_1$-norm ball. This difficulty arises because the minimization problem in (1) may have *multiple* solutions, and in this case, standard duality provides no mechanism for selecting the correct one.
>     To solve the primal optimization problem, one must use a LP solver, or rely on an iterative averaging scheme (for example, Corollary 12.16 of [*]). The averaging approach, however, has two severe drawbacks. First, the averaging scheme converges slowly, at best with a rate of $1/N$ after $N$ iterations. Second, this iterative process completely negates the benefit of the Frank-Wolfe method, as the recovered solution is typically dense and no longer a simple structured vertex.
>
>     Our numerical experiments in Section 3 support this analysis. We also include an additional result where $X$ is defined by the Nuclear norm, showing an even greater improvement as we are dealing with semidefinite programs.
>
>     [*] Orabona, Francesco. A modern introduction to online learning. arXiv preprint arXiv:1912.13213 (2019).
>
> *   **Algorithm design and analysis.**
>     We provide a detailed comparison with Cao et al. in our response to `Reviewer 2wsw`. While our gradient estimators are the same as those of Cao et al. , our overall algorithm design, and consequently its analysis, is fundamentally different. The key differences are outlined below.
>     1.  *Type of linear oracle.* As discussed above, our algorithm uses a linear oracle over $X$ only, while theirs requires one for $X \cap H_t$, where $H_t$ is a dynamically changing halfspace. This results in a different analysis with worse iteration complexity, but better overall running time. Additionally, the linear minimization oracle in their algorithm is only guaranteed to be feasible with *high probability*, hence the algorithm can stall at any iteration due to oracle infeasibility.
>
>     2.  *Regularization versus sublevel set approximation.* The key difference between the two algorithm designs is their approach. Cao et al. approximates the lower level problem $G(x) \leq G_{\text{opt}}$ with an outer halfspace approximation $H_t$ at each step. On the other hand, we use a regularization approach of adding $\sigma_t f$ to guide the iterates towards the bilevel solution. This is the reason behind the different oracles, and also explains why the analysis is so different.
>
>     3.  *Single stage versus two-stage algorithm.* The algorithm of Cao et al. requires initialization with a point $x_0$ that is $\epsilon_g/2$-optimal for the lower-level problem. Finding such a point can be non-trivial especially in stochastic setting as it requires algorithms that provide such a point with high probability rather than in expectation. In contrast, our algorithm does not require any such initialization.
>
>     4.  *Any-time versus approximate guarantees.* The warm-start parameter $\epsilon_g$ of Cao et al. also appears in the definition of the halfspaces $H_t$ at each iteration. In essence, this means that the sublevel set $\\{x : G(x) \leq G_{\text{opt}} + \epsilon_g\\}$ is contained in $H_t$. However, this means that if their algorithm is run indefinitely, there is no assurance that their algorithm converges to an optimal solution of the bilevel problem, but rather it may instead converge to a minimizer of $f$ over the sublevel set $\\{x : G(x) \leq G_{\text{opt}} + \epsilon_g\\}$, which is an *approximate* solution. In contrast, if our algorithm is run indefinitely, we proved in Appendix B.1 that it will converge to an optimal solution of the bilevel problem. In other words, our algorithm is *any-time* whereas theirs is not.
>
> *   **Worst iteration complexity.** The reviewer is correct in saying that the iteration complexity of Cao et al. is better than ours. This is a direct consequence of them utilizing a more powerful linear oracle over $X \cap H_t$ each iteration, where $H_t$ is a dynamically varying halfspace. On the other hand, we use a much cheaper oracle over $X$.
>     Our experiments in Section 3 demonstrate that our algorithms achieve lower gaps in the same amount of running time, thus there is immense computational benefit to trading off iteration complexity for a lower per-iteration cost.
>
> *   **Answer to Neutral Question.** In Section 3, we included tables reporting the average number of iterations, which equals the number of linear optimization oracle calls. We observe that our algorithms require 2–3 orders of magnitude more linear optimizations compared to. Following Cao et al., we implemented the minimization oracle over $X \cap H$ using CVXPY. To our knowledge, there is no more efficient method for linear optimization over a norm ball with an additional hyperplane constraint; however, if the reviewer is aware of one, we would be happy to explore it.
>
>     To further demonstrate our algorithm’s performance relative to Cao et al., we conduct a new synthetic experiment on the following bilevel matrix completion problem
>     $$ \begin{array}{cl}
>         \min\limits_{X \in \mathbb{R}^{m \times p}} & F(X):= \frac{1}{2mp} \lVert X\rVert_F^2 = \frac{1}{2mp}\sum_{(i,j) \in [m] \times [p]} |X_{i,j}|^2 \\\\
>          \text{s.t.} & X \in \arg\min_{\lVert Z \rVert_{nuc} \leq r} G(Z):= \frac{1}{|\Omega|} \sum_{(i,j) \in \Omega} H_{\xi}(M_{i,j}-Z_{i,j}).
>     \end{array} $$
>     where $X \in \mathbb{R}^{m \times p}$, $H_\xi$ is the Huber loss function with parameter $\xi$ and $M$ is a given matrix with only entries $\{M_{i,j}: (i,j) \in \Omega\}$ are observed.
>     We generate $M$ as follows: we first generate a random matrix $M \in \mathbb{R}^{m \times p}$ where $m=p=100$ and then randomly choose $20\\%$ of the entries as being unobserved. We also choose $\xi = 1, r = 10$. The linear minimization problem $\text{Tr}(C^\top X)$ over the Nuclear norm ball is solved analytically by $-\delta u_1 v_1^\top$ where $u_1, v_1$ are the left and right singular vectors corresponding to the largest singular value of $C$, respectively. Meanwhile, the linear minimization problem over the Nuclear norm ball intersecting a half-space is solved using CVXPY. We note that although our approach can handle larger-scale problems, the CVXPY solver fails on these instances due to the out of memory error. Below, a summary of our experiment after running each algorithm for $10$ minutes is reported. This further shows that our algorithm has better empirical performance against Cao et al..
>
> | Method | Oracle Time | Inner Sub-optimality | Absolute Outer Sub-optimality | # Iteration Counts |
> | :--- | :---: | :---: | :---: | :---: |
> | `IR-FSCG` | 8.91e-3 | 8.11e-7 | 2.60e-7 | 57778 |
> | `IR-SCG` | 8.33e-3 | 1.71e-3 | 1.61e-4 | 64295 |
> | `SBCGF` | 1.96e2 | 8.67e-3 | 7.94e-4 | 3 |
> | `SBCGI` | 1.47e1 | 8.94e-3 | 8.12e-4 | 34 |
>
> We thank the reviewers for their constructive feedback on the paper's presentation. As suggested by `Reviewer 2wsw`, we will reorganize the main text to present the convex and nonconvex results side-by-side, first in Section 2 for the one-sample case (Theorem 1 and Theorem 5 from the appendix) and then in Section 3 for the finite-sum case (Theorem 2 and Theorem 6 from the appendix). To emphasize the nonconvex setting, we will replace the second numerical experiment with the nonconvex example from the appendix. Finally, as suggested by `Reviewer ZPtF`, we will revise the title and abstract to reflect our focus on *simple* bilevel problems.
>
> Please let us know if there are any remaining points we can address or clarifications we can provide that would be helpful in your final evaluation.

---

> > ### Comment · Reviewer_TL81 · 2025-08-03
> >
> > Thank you for the clarification. I get the difference between optimizing over $X \cap H$ v.s. $X$. As you explained in your response, this difference is most significant when there are some simple algorithms that can optimize over $X$, but for $X \cap H$ it may require standard projection-based method.
> >
> > * One follow-up question: is the slightly worse non-asymptotic convergence rate inevitable for the projection-free method? That is said, is there any lower bound or impossibility example that you can show for the projection-free method (only using $X$)?
> >
> > Could you please try to provide any preliminary guess or answer to my question above? This will strengthen the contribution on the projection-free algorithms and their convergence rates. I will update my assessment accordingly.
> >
> > I also went through Cao et al.'s algorithm and analysis. Although the algorithms are similar, Cao et al. use standard momentum method and keep the solution within the half space with near optimal lower level value, while this paper uses Frank-Wolfe style algorithm instead. The convergence analyses are significantly different despite the algorithm similarity in my opinion.

---

> ### Author Response · Authors · 2025-08-04
>
> We thank the reviewer for their insightful question. Throughout our response, we reference the bibliography from our submission. We address it from four distinct perspectives:
>
> **1. Justification for the Convergence Rate**
> Our proposed algorithms achieve rates of $O(1/T^{1/4})$ and $O(1/T^{1/2})$ in the one-sample and finite-sum settings, respectively. These are indeed slower than the $O(1/T^{1/2})$ and $O(1/T)$ rates achieved by state-of-the-art stochastic Frank-Wolfe algorithms like STORM and SPIDER in standard (non-bilevel) settings.
>
> We conjecture that this slower rate is a fundamental consequence of the iterative regularization approach. In essence, the inverse of the regularization parameter, $1/\sigma_t$, plays the role of the dual variable for the constraint $\\{x: G(x) \leq G_{\text{opt}}\\}$. As shown by Friedlander & Tseng [17] for a special case, there exists a critical parameter $\sigma^\star$ for which the optimal value of the regularized problem with any $\sigma_t \leq \sigma^\star$ coincides with the optimal value of the bilevel problem. Unfortunately, as this critical value is unknown, the iterative regularization scheme requires $\sigma_t$ to decrease at each iteration, asymptotically approaching zero.
>
> Since our method is *essentially* a primal-dual method, the rates in our paper are expected. To see this, in the deterministic case, Lan [*] (Chapter 7) and Lan et al. [**] (discussion after Corollary 2.5) establish that the convergence rate of primal-dual methods that rely on linear minimization oracles over the base set $X$ is lower-bounded by $O(1/T^{1/2})$ in the *deterministic* setting. Given this result, our rate of $O(1/T^{1/4})$ in the one-sample stochastic setting is consistent with expectations, as stochastic rates are often slower than their deterministic counterparts by a square-root factor.
>
> **2. Consistency with Other Iterative Regularization Approaches**
> This trade-off between a slower rate and the efficient implementation offered by iterative regularization is not unique to our work and is documented in the literature. For instance, using the iterative regularization approach in the deterministic setting:
> - The proximal gradient-based algorithm of Merchav and Sabach [35] achieves $O(1/T^{1/2})$ under smoothness conditions for both levels, as opposed to the standard $O(1/T)$ rate.
> - The fast proximal gradient-based algorithm of Merchav et al. [36] achieves $O(1/T)$ under a standard composite structure for both levels, in contrast to the standard $O(1/T^2)$ rate for accelerated methods.
>
> We believe that the slower rates of the iterative regularization-based approach are unavoidable when aiming to provide convergence rates for *both* the inner and upper levels simultaneously.
>
> **3. On Achieving Faster Rates and the Oracle Complexity Trade-off**
> To the best of our knowledge, existing methods that achieve faster (i.e., standard non-bilevel) convergence rates do so by employing a **stronger oracle**. For example:
> - Doron and Shtern [11] achieve faster rates for projection-free methods by employing a linear oracle over **dynamically changing** sublevel sets of the outer-level objective function in addition to the linear oracle over $X$.
> - Similarly, Jiang et al. [28] and Cao et al. [4] achieve faster rates by intersecting $X$ with a **dynamic linear** constraint.
>
> These approaches, however, require a more complex oracle capable of solving an optimization problem over an evolving, more constrained set. Our approach, by design, avoids this complexity by operating solely on $X$. This design choice trades the need for a more powerful oracle for a slower convergence rate, which is governed by the necessary tuning of the regularization parameter.
>
> **4. Discussion on Theoretical Lower Bounds**
> To our knowledge, no specific lower bound has been established for stochastic projection-free methods, even in the standard (non-bilevel) setting. For the deterministic case, Lan [*] provides such a lower bound. For further discussion on how deterministic bounds relate to our stochastic rates, we refer the reviewer to our argument in point 1.
>
> In summary, we conjecture that our derived rates may be optimal for the class of algorithms that rely only on a standard linear minimization oracle over the **fixed** base set $X$. While established lower bounds do not yet exist even for simple stochastic projection-free methods, we acknowledge that proving this conjecture for the bilevel problem remains an important and open question requiring more thorough analysis. If the reviewer wishes, we would be happy to include this discussion as a remark in the final version of the paper.
>
> **Additional References**
>
> [*] Lan, G. (2020). First-Order and Stochastic Optimization Methods for Machine Learning. Springer.
>
> [**] Lan, G., Romeijn, E., & Zhou, Z. (2021). Conditional gradient methods for convex optimization with general affine and nonlinear constraints. SIAM Journal on Optimization, 31(3), 2307-2339.

---

### Official Review · Reviewer_2wsw · 2025-07-03

**Clarity:** 1
**Significance:** 3
**Originality:** 2
**Rating:** 2
**Confidence:** 3

**Summary:**

This paper studies *simple* (stochastic) Bilevel optimization, where the target variable $x$ is minimized over the solution set of a convex lower-level problem. The core contribution is an Iteratively Regularized Conditional Gradient (IR-CG) framework that avoids projection oracles by relying solely on linear optimization over the base feasible set. The authors present two algorithms (IR-SCG and IR-FSCG) for one-sample and finite-sum settings, respectively, and derive non-asymptotic convergence guarantees. Empirical comparisons on two benchmark problems demonstrate improved practical performance relative to baseline methods.

**Questions:**

-	Line 89-91: “However, their method requires linear optimization over intersections of X with halfspaces.” – could you clarify what *with halfspaces* means and why this is an issue? Is it computationally prohibitive?

-	On vanishing penalty parameters: assume we fix the max iteration $T$ or target accuracy. What happens if we set $\sigma_t = \sigma_T$ for all $t$? Does it degrade the convergence result? (if so, how worse would it be?)

**Ethical Concerns:**

["NO or VERY MINOR ethics concerns only"]

**Final Justification:**

I think the paper needs more polish at least.

**Limitations:**

Overall, the current presentation needs to be significantly improved. I recommend rejection this round. It is difficult to comprehend the precise novelty, the trade-offs relative to prior work, or the design choices behind the algorithms.

**Quality:**

2

**Strengths And Weaknesses:**

My main concern is the poor technical writing that makes it difficult to assess their core contributions. With the current draft, it is very difficult to apprehend what the major improvement over the prior work is, and what is particularly interesting in their algorithm design. Detailed examples are below:

-	The paper does not clearly explain how the proposed rates improve existing results. For instance, Cao et al. [4], achieves $O(t^{-1/2})$ convergence rates for both levels, while this paper obtains $O(t^{-1/2+p})$ and $O(t^{-p})$ for the upper and lower level, respectively, which seems worse anyway. It is not very clear either how significant the claimed computational gain is quantitatively.

-	Algorithm 1 appears to be merely a combination of STORM gradient estimation and Frank-Wolfe updates on a penalized objective (except the last two lines, which is ad-hoc to varying penalty parameter). Aside from the weighting scheme and averaging, the novelty in algorithm design is not highlighted clearly.

-	While it's reasonable to analyze both one-sample and finite-sum stochastic settings, presenting them in parallel obscures the logical flow. It would be more natural to treat the one-sample case as the more general scenario, and then specialize to finite-sum with in-depth intuition and analysis. This would also help clarify why the rates differ -- something that is rarely the case in related work.

---

> ### Author Rebuttal · Authors · 2025-07-29
>
> We thank the reviewer for their thoughtful feedback. We address the concerns raised below:
> *   **Convergence rates, contribution and differentiation with Cao et al..** We thank the reviewer for this important question. The reviewer is correct that our convergence rates are not as strong as those in Cao et al.. However, our work's primary goal is not to improve iteration complexity but to propose a method that is more computationally efficient. We improve upon their approach in several fundamental ways, which we will clarify in the revised manuscript.
>
>     1.  *Improved per-iteration costs.* Cao et al. requires at each iteration solving a linear optimization over $X \cap H$, where $H = \\{x : a^\top x \leq b\\}$ is some halfspace. In contrast, our algorithm only requires linear optimization over $X$, which in general is cheaper. As Cao et al. use a more powerful oracle, they achieve a better iteration complexity. However, for many examples, optimizing over $X \cap H$ is expensive compared to just $X$. For example, if $X$ is an $\ell_1$-ball, then optimizing over $X$ can be done exactly in $O(d)$. In contrast, optimization over $X \cap H$ has no closed-form expression and thus requires a linear solver. Cao et al. also use a linear solver to implement this oracle. In such situations, we believe our method is beneficial compared to Cao et al. despite the worse iteration complexity, since the *overall* computational cost is better.
>
>         For the sake of illustration, assume that $X$ is a norm ball. Consider the linear minimization oracle problem over $X \cap H$:
>         $$ \min_x \\{ c^\top x : \lVert x \rVert \leq r, a^\top x \leq b \\}. $$
>         One can show that its dual problem takes the form:
>         $$ \max_{\mu} \\{ -r \lVert c + \mu a\rVert_* - \mu b : \mu \geq 0 \\}, $$
>         where $\lVert \cdot\rVert_*$ is the dual norm. Given an optimal dual solution $\mu^\star$, the optimal primal solution satisfies
>         $$ x^\star \in \arg\min_{x} \\{ c^\top x + \mu^\star (a^\top x - b): \lVert x \rVert \leq r \\}. \tag{1} $$
>
>         In the case of $\ell_2$-ball, the dual problem is a nonlinear one-dimensional convex optimization problem, which can be solved efficiently with a bisection algorithm. Each iteration of the bisection algorithm costs $O(d)$, so it is not as fast as the closed form for linear optimization over $X$ alone. Once the optimal dual multiplier $\mu^\star$ is found, the optimal primal solution $x^\star$ can be computed in $O(d)$ time since (1) has a unique optimum.
>
>         In the case of $\ell_1$-ball, we have $\lVert \cdot \rVert_* = \lVert \cdot \rVert_\infty$ thus the dual is a piecewise linear one-dimensional convex optimization problem. An exact solution can be obtained by computing the breakpoints, but there are $O(d^2)$ potential points, which is much more expensive than $O(d)$. However, obtaining the optimal dual variable $\mu^\star$ does not lead to easy recovery of the primal solution $x^\star$ as the minimization in (1) may have *multiple* solutions, and in this case, standard duality provides no mechanism for selecting the correct one.
>         In this case, one must use a LP solver or rely on an iterative averaging scheme (e.g., Corollary 12.16 of [*]). The averaging approach, however, converges slowly, at best with a rate of $1/N$ after $N$ iterations. Besides, it negates the core benefit of the Frank-Wolfe method, as the recovered solution is no longer a simple structured vertex.
>
>         Our numerical experiments in Section 3 directly support this analysis. We also include an additional result where $X$ is defined by the Nuclear norm, showing an even greater improvement as we are dealing with semidefinite programs. We refer the reviewer to our response to `Reviewer TL81`.
>
>         [*] Orabona, Francesco. A modern introduction to online learning. arXiv:1912.13213 (2019)
>     2.  *No initialization requirement.* Their algorithm requires a specific *warm start*: it must be initialized that is already $\epsilon_g/2$-optimal for the lower-level problem. Finding such a point can be non-trivial especially in stochastic setting as it requires algorithms that have high probability guarantees rather than in expectation. Moreover, as observed in our numerical experiments (consistent with both theoretical and numerical results of Cao et al.), the convergence behavior of the inner-level problem degrades, going from $\epsilon_g/2$ to $\epsilon_g$, rather than improving. In contrast, our method is initialization-free and does not requires such pre-computation.
>
>     3.  *Any-time guarantees and asymptotic correctness.* In addition to requiring a warm start with respect to the parameter $\epsilon_g$, the method of Cao et al. also uses this parameter to define $H$ at each iteration. Thus, their algorithm is *not any-time*: running it indefinitely does not guarantee convergence to an optimal solution of the bilevel problem, but rather to a minimizer of $f$ over the set $\\{x : G(x) \leq G_{\text{opt}} + \epsilon_g\\}$, which is only an *approximate* solution. In contrast, our algorithm enjoys *any-time* behavior with asymptotic convergence guarantees (Theorems 3 and 4 in the appendix).
>
>     4.  *Preserving structural properties.* A key advantage of projection-free algorithms is their ability to maintain low-complexity updates for structured sets. For instance, with the simplex (for sparsity) or the Nuclear norm ball (for low-rankness), the linear oracle over $X$ simply returns an atomic vertex. This crucial benefit is lost in their method, as the oracle over $X \cap H$ typically returns a dense solution. This negates one of the core strengths of projection-free algorithms.
>
>     5.  *Easy implementation.* In addition to relying on the parameter $\epsilon_g$ for a careful initialization and defining $H$ at each iteration, the algorithm in Cao et al. also depends on an additional parameter $K_t$ to define $H$, which itself is defined in terms of noise variance, set diameter and objective function smoothness. Besides, the linear minimization oracle in their algorithm is only guaranteed to be feasible with *high probability*, hence the algorithm can stall at any iteration due to oracle infeasibility. In contrast, our algorithm requires only the parameter $\varsigma$, and our oracle is always feasible. While the optimal choice of $\varsigma$ depends on the same underlying parameters, this choice only changes the constant factor rather than the convergence rate. In practice, one can simply try $\varsigma \in \\{0.1, 1, 10\\}$ to achieve satisfactory empirical performance.
>
> *   **Novelty of the algorithm design and analysis.** We thank the reviewer for this observation. While our framework indeed integrates established components like stochastic gradient estimators (STORM/SPIDER) and the Frank-Wolfe method, our core contribution lies in the specific algorithmic innovations required to address the unique bilevel structure. Specifically, we introduce a dynamic regularization term, $\sigma_t f$, which is crucial for managing the bilevel objective. This new component, combined with any-time step-size rules, require a non-trivial convergence analysis. A key feature of this analysis is a novel *averaging scheme*, which was specifically devised to guarantee *simultaneous* convergence for both inner- and outer-level objectives while preserving the algorithm's any-time properties.
>
> *   **Presentation:** We thank the reviewers for their constructive feedback on the paper's presentation. In the final version, we will implement several key revisions. Following `Reviewer B9Ux`'s suggestion, we will restructure the paper to better highlight our contributions to the nonconvex case. We will reorganize the main text to present the convex and nonconvex results side-by-side, first in Section 2 for the one-sample case (Theorem 1 and Theorem 5 from the appendix) and then in Section 3 for the finite-sum case (Theorem 2 and Theorem 6 from the appendix). We will also replace the second numerical experiment with the nonconvex example from the appendix. To accommodate these changes within the page limit, the proof sketches will be removed from the paper. Finally, as suggested by `Reviewer ZPtF`, we will revise the title and abstract to better reflect our focus on *simple* bilevel problems, including those with nonconvexity in the outer-level.
>
> **Answers to Questions:**
>
> *   We thank the reviewer for pointing out this ambiguity. We will revise the text to clarify that our comparison refers to intersecting with a *single* halfspace, not multiple. We thus change the text from *halfspaces* to *some halfspace*. Note that the method in requires solving a linear oracle over the set $X \cap H_t$, where the halfspace $H_t = \\{x : a^\top x \leq b\\}$ is a single, but *dynamically changing* halfspace that approximates the functional constraint set $\\{x : G(x) \leq G_{\text{opt}} + \epsilon_g/2\\}$ at each iteration. As discussed in *Improved per-iteration costs.*, even a single additional constraint can make the subproblem expensive.
>
> *   Fixing $\sigma_t = \varsigma T^{-p}$ based on a predetermined total iteration count $T$ would indeed preserve the theoretical convergence rate. However, this approach has two significant drawbacks. First, fixed penalty schedule makes the algorithm's final accuracy dependent on the choice of $T$, whereas our approach guarantees convergence to the true optimum as $t \to \infty$. Second, we consistently observe that the dynamic, time-varying regularization scheme achieves better practical results. This, indeed, motivates us to prove the convergence rate for the anytime update rule.
>
> We thank the reviewer again for their insightful feedback. We hope our response and planned revisions have addressed the main concerns. We were wondering if there are any other points we can clarify that could further improve the manuscript and their assessment of the work?

---

### Note · Authors · 2025-08-11

We thank the reviewers for their thoughtful comments, which have given us the opportunity to clarify the distinct advantages of our work.

1. **Comparison with Cao et al. [4]** While Cao et al. [4] achieve a theoretically *faster* convergence rate, this stems from their use of a stronger linear oracle over the intersection of the base set $X$ and a *dynamic* halfspace $H_t$, rather than from a fundamentally tighter analysis. Our method, in contrast, uses a *cheaper oracle* over only $X$. This advantage is seen numerically in Section 3, where our methods show notable improvement. Besides this, our method offers several other distinct theoretical and practical advantages. First, our method does not require a *warm-start initialization* which is often non-trivial in stochastic settings, and consequently our method has *asymptotic correctness guarantees*. Furthermore, ours also has *any-time guarantees*. Second, the $H_t$ are *not always feasible*, and even ensuring feasibility with high probability requires problem-specific parameters such as Lipschitz constants. We circumvent this issue by avoiding $H_t$ altogether, which also *simplifies the implementation* by not requiring knowing the parameters. Finally, projection-free methods are valued for their low-complexity updates on structured sets. This benefit is *lost* in their approach, as the oracle over $X \cap H_t$ typically yields dense solutions, undermining one of the main advantages of projection-free algorithms.

4. **Tightness of the convergence analysis** While we do not have a formal proof, we conjecture that our rates are optimal for any algorithm that *only* relies on a linear minimization oracle over the base domain $X$. In the deterministic case, Lan [*] (Chapter 7) and Lan et al. [**] (see discussion after Corollary 2.5; references in the last comment to `Reviewer TL81`) show that primal–dual methods using such oracles are lower-bounded by a rate of $O(1/T^{1/2})$. Our $O(1/T^{1/2})$ rate in the finite-sum setting is thus potentially tight, as our method is *essentially* a primal–dual algorithm, with $\sigma_t$ playing the role of the inverse dual variable. Similarly, our $O(1/T^{1/4})$ rate in the one-sample stochastic setting is also consistent with expectations, as stochastic rates are often slower than their deterministic counterparts by a square-root factor.

We once again thank the reviewers for their constructive feedback and the engaging discussion during the rebuttal phase.

---

### Decision · Program_Chairs · 2025-09-17

**Decision:**

Accept (poster)

**Comment:**

This work proposes projection-free algorithms for stochastic convex bilevel problems (with nonconvex extensions in the appendix). The problem is $\min_{x\in X_{opt}} f(x)$ where $X_{opt}=\arg\min_{z\in X} g(x)$. In particular, the authors provide an algorithm that uses linear oracles over the set $X$ instead of the prior work [Cao et al 2023] that used linear oracles over the intersection of $X$ and dynamic selected half spaces $H_t$.

This work had diverging reviews with some reviewers suggesting acceptance and some reviewers suggesting rejection.

The main weakness of this work pointed out by some of the reviewers was that the resulting sample complexities in this work are not as good as [Cao et al. 2023]. In particular, the new rate is $O(1/T^{1/4})$, whereas the state-of-the-art rate was $O(1/T^{1/2})$ in the prior work in the stochastic case. Similarly, the rate in this paper is $O(1/T^{1/2})$ in the finite sum case whereas the state-of-the-art is $O(1/T)$. The authors argued that their oracle is simpler than the previous work to justify the worse sample complexity and that they suspect their complexities under their oracles to be tight. However, no formal guarantee is provided. Empirical results seem promising.

In the previous work, the linear minimization problem over the intersection of two sets had to be solved by off-the-shelf solvers such as cvxpy whereas the oracle in this work is the linear minimization problem over the compact set $X$ for which there are good examples where lmo over $X$ is more preferable than projection, especially with matrix problems. The authors provided a new numerical experiment on this line.

After reading the reviews and the discussions as well as the paper with the relevant literature, I do agree with the reviewers who suggest acceptance. In particular, I think that having a simpler oracle (just lmo over a compact set $X$) compared to the lmo over the intersection of sets for which one needs a subsolver (which was used in the previous work) is interesting. In particular, in such a case with different oracles, comparing convergence rates is not always fair. Therefore, I value and find compelling the motivation of the paper as well as its theoretical and empirical contribution. I also appreciate that the paper and the appendix are well-written.

I recommend the authors to clearly describe the differences between their work and [Cao et al., 2023] involving their informal explanations justifying their worse convergence rate. Another small recommendation for the authors: you may consider slightly changing your title to clarify that you are focusing on an algorithm with a "simpler oracle". This can be more descriptive than using "efficient" to describe your method.

[Cao et al., 2023] J. Cao, R. Jiang, N. Abolfazli, E. Yazdandoost Hamedani, and A. Mokhtari. Projection-free methods for stochastic simple bilevel optimization with convex lower-level problem. In Advances in Neural Information Processing Systems, pages 6105–6131, 2023.